# Transcription factor NFYA directs male meiotic entry by regulating accessible chromatin at meiotic promoters in mice

Martin Säflund [iD] [1,3], Masomeh Askari[1,3], Atiyeh Eghbali[1,3], Mukhtar Mohamed Abdi [iD] [1], Dilay Deren Er [iD] [1], Ann-Kristin Östlund Farrants [iD] [1], Tianxiong Yu [iD] [2✉] & Deniz M Özata [iD] [1✉]

## Abstract

**Meiotic prophase I, characterized by homologous recombination and synapsis, is a critical step in spermatogenesis. This process entails extensive changes to chromatin and transcription. Prior to prophase I, accessible chromatin bound by paused Pol II at meiotic gene promoters is essential for their timely activation later during meiosis. However, the factors responsible for establishing accessible chromatin at meiotic gene promoters before entry into prophase I are unknown. Here, we discovered that NFYA, expressed in pre-meiotic germ cells, regulates accessible chromatin at meiotic gene promoters, including those activated by the STRA8/MEISON axis. Concordantly, conditional germline deletion of Nfya in male mice blocks meiotic entry. Single-cell ATAC-seq analysis shows that loss of NFYA in pre-meiotic cells disrupts accessible chromatin at poised meiotic gene promoters. These findings establish NFYA as a regulator of accessible chromatin at meiotic gene promoters and of the timely activation of the meiotic genetic program.**

**Keywords** Spermatogenesis; Transcription Factor NFYA; Meiotic Initiation; Meiosis; Single-cell Multi-omics
**Subject Categories** Cell Cycle; Chromatin, Transcription & Genomics

## Introduction

Mammalian spermatogenesis is a step-wise developmental process to produce spermatozoa from spermatogonial stem cells (SSC) (Handel and Schimenti, 2010; Säflund and Özata, 2023). This intricate and highly conserved developmental program involves three major steps. SSC mitotically divide, differentiate, and enter meiosis (Dirk and Grootegoed, 1998); upon meiotic entry, primary spermatocytes (SpI) undergo two rounds of meiotic division, resulting in haploid round spermatids (RS) (Handel and Schimenti, 2010; Säflund and Özata, 2023); RS then

finally differentiate into spermatozoa, i.e., spermiogenesis (O'Donnell, 2014). Meiosis is a critical step during spermatogenesis. Upon entry into prophase I of meiosis, chromosomes are reorganized into proteinaceous structures, axial elements (AE), to facilitate homologous chromosome pairing and synapsis, as well as crossover (Handel and Schimenti, 2010; Säflund and Özata, 2023; Ishiguro, 2024).

Both entry into meiosis and progression through prophase I entail extensive chromatin and transcriptional changes (Soumillon et al, 2013; Hammoud et al, 2014; Green et al, 2018; Hermann et al, 2018; Maezawa et al, 2018; Geisinger et al, 2021). The switch from mitosis to meiosis is regulated by the transcription factors (TFs) STRA8 and MEIOSIN, which activate the meiotic genetic program in response to retinoic acid (Anderson et al, 2008; Ishiguro, 2024). During this switch, Znhit1 replaces canonical histone H2A with H2A.Z (Sun et al, 2022). At the pachytene stage of prophase I, the TFs A-MYB and TCFL5 initiate the transcriptional burst of thousands of meiotic genes, including genes producing pachytene piRNAs, as well as genes required during spermiogenesis (Li et al, 2013; Özata et al, 2020; Yu et al, 2021; Cecchini et al, 2023; Yu et al, 2023).

Strikingly, in pre-meiotic spermatogonia (SpG), the promoters of genes expressed during meiosis already have accessible chromatin (Maezawa et al, 2018), and are occupied by paused RNA polymerase II (Pol II) (Kaye et al, 2024). The establishment of paused Pol II at gene promoters in SpG is thought to be a key regulatory step that primes genes for timely activation later during meiosis. In fact, at the pachytene stage of prophase I, A-MYB recruits testis-specific bromodomain protein, BRDT, to release paused Pol II into elongation (Alexander et al, 2023). Nevertheless, the factor that regulates accessible chromatin, and thus promotes the binding of Pol II at meiotic gene promoters before entry into prophase I, remain largely unknown.

Here, we demonstrate that prior to the initiation of meiosis, NFYA regulates accessible chromatin and thus facilitates paused Pol II occupancy at the promoters of genes expressed during meiotic entry and progression. Germline-specific deletion of *Nfya* results in spermatogenic arrest at meiotic entry. Together, our findings identify NFYA as a key factor regulating chromatin

[1]Department of Molecular Biosciences, The Wenner-Gren Institute, Stockholm University, Stockholm, Sweden. [2]Department of Genomics and Computational Biology, University of Massachusetts Medical School, Worcester, MA, USA. [3]These authors contributed equally: Martin Säflund, Masomeh Askari, Atiyeh Eghbali.
✉E-mail: tianxiong.yu@umassmed.edu; deniz.ozata@su.se

accessibility during spermatogenesis and provide new insights into the molecular mechanisms of meiotic initiation in mice.

# Results

## Each major step of spermatogenesis retains a distinct gene expression profile

To define the expression profiles of genes across spermatogenesis, we used our published RNA sequencing (RNA-seq) data from Fluorescence-Activated Cell Sorting (FACS)-purified germ cells with spike-in sequences (Gainetdinov et al, 2021; Cecchini et al, 2023; Yu et al, 2023), and classified genes according to their absolute steady-state transcript abundance: i.e., molecules per FACS-purified germ cell (Appendix Fig. S1A; Dataset EV1A). Those genes, with transcript abundance ≥fourfold higher in SpG than in pachytene/diplotene spermatocytes (P/D), secondary spermatocytes (SpII), and round spermatids (RS), were classified as mitosis genes (1181 genes). Whereas, the transcript abundance of meiosis-I genes in P/D was ≥fourfold higher than in other germ cells (3062 genes). As RNAs required for spermiogenesis are first expressed during prophase I (Kierszenbaum and Tres, 1975; Grunewald et al, 2005; da Cruz et al, 2016; Naro et al, 2017), we classified the third category as spermiogenesis genes whose transcript abundance in P/D was ≥fourfold higher than in SpG and changed ≤twofold in SpII and RS (2345 genes). Exemplified genes confirmed the differential expression of previously described marker genes across spermatogenesis (Appendix Fig. S1B–D). The mRNA abundance of *Sall4* and *Dmrt1* genes was highest in SpG consistent with their crucial function in the maintenance of SpG (Matson et al, 2010; Hobbs et al, 2012) (Appendix Fig. S1B). In human and mouse testis, at the pachytene stage of male prophase I, ~100 well-annotated genes produce long transcripts that are processed into pachytene piRNAs, i.e., pachytene piRNA genes (Li et al, 2013; Özata et al, 2020; Yu et al, 2021). Consistently, the piRNA precursor transcript of *pi9* was highest in P/D (Appendix Fig. S1C). Of note, the category for meiosis-I genes included 99 of the 100 mouse pachytene piRNA genes (Dataset EV1A,B). mRNA of *sycp2* gene, whose protein product is an essential component of the synaptonemal complex formed during prophase I of meiosis (Yang et al, 2006), was higher in P/D compared to other germ cells (Appendix Fig. S1C). *Crem* and *Spata6* genes are essential for spermiogenesis (Blendy et al, 1996; Yuan et al, 2015). Consistently, they were first expressed in P/D and their RNA abundance increased in RS (Appendix Fig. S1D).

We further performed Gene Ontology analysis (GO; ≥twofold change; Fisher's exact test, FDR < 0.01; Dataset EV1C–E) to characterize the biological function of genes in each category. We found that mitotic genes were enriched for mitotic DNA replication, housekeeping functions such as cell proliferation and morphogenesis, and macromolecule biosynthesis (Dataset EV1C). GO for meiosis-I genes revealed enriched gene categories related to meiosis, including positive regulation of DNA repair, male meiotic nuclear division, and piRNA processing (Appendix Fig. S1E; Dataset EV1D). Notably, spermiogenesis genes were mainly enriched for GO related to spermatid differentiation, sperm function, and fertilization, such as acrosome assembly, flagellated sperm movement, and sperm-egg recognition (Appendix Fig. S1E; Dataset EV1E).

## Promoter-proximal regions of poised genes expressed during meiosis accumulate paused Pol II and have accessible chromatin in spermatogonia

Recent studies reported that SpG and early prophase I cells accumulate paused Pol II around the transcription start sites (TSSs) of genes expressed during meiosis (Alexander et al, 2023; Kaye et al, 2024). Pol II pausing at TSSs of such genes is indeed required for the timely transcriptional activation during the pachynema stage of prophase I (Alexander et al, 2023; Kaye et al, 2024). A-MYB, expressed in pachytene spermatocytes, recruits BRDT to release paused Pol II into elongation (Bolcun-Filas et al, 2011; Li et al, 2013). Using publicly available Precision Run-On sequencing data (PRO-seq (Kaye et al, 2024)) from purified SpG, SpI, and RS, we quantified paused and elongating Pol II across meiosis-I and spermiogenesis genes. Spike-in sequences in each sample enabled us to quantify the absolute PRO-seq signal. Note that PRO-seq provides the genome-wide mapping of transcriptionally engaged RNA polymerases (Core et al, 2008). However, the transcription units we study are transcribed by Pol II.

Engaged Pol II peak around TSS is the characteristic of promoter-proximal pausing of Pol II (Core et al, 2008), wherein the promoters of many metazoan developmental genes exhibit paused Pol II that is later released into active elongation in response to secondary signals (Core and Adelman, 2019). We thus first assessed whether the TSSs of meiosis-I and spermiogenesis genes retain significantly engaged Pol II peaks in SpG prior to their expression in SpI or RS. Using MACS3 (FDR < 0.01), we found that ~79% of meiosis-I genes exhibited significantly engaged Pol II peaks within the ±2 kb of their TSSs in SpG (2415 of 3062 genes; median distance from the TSS to nearest engaged Pol II peak = 158 bp) (Fig. EV1A; Dataset EV1B). Likewise, the promoters of ~63% of spermiogenesis genes retained engaged Pol II peak in SpG (1488 of 2345 genes; median distance from the TSS to nearest engaged Pol II peak = 166 bp) (Fig. EV1A; Dataset EV1B). This suggests that Pol II is loaded on chromatin and remains paused at the promoters of the majority of both meiosis-I and spermiogenesis genes prior to their expression in SpI or RS. Therefore, we named those genes with engaged Pol II peak near their TSSs as poised genes, while genes without a significant engaged Pol II peak within the ±2 kb of their TSSs were named non-poised genes. We next measured the relative promoter and gene body PRO-seq signal for poised and non-poised genes across SpG, SpI, and RS (Figs. 1A and EV1B,C). For both poised and non-poised meiosis-I genes, active RNA synthesis at gene bodies was highest in SpI and declined ~threefold in RS (Figs. 1A and EV1C). Whereas, nascently transcribed RNA signal at gene bodies from poised and non-poised spermiogenesis genes remained almost unchanged between SpI and RS (decrease <twofold; Figs. 1A and EV1C). Mitosis genes, by contrast, revealed a ~threefold higher promoter and gene body PRO-seq signal in SpG than in SpI and RS, suggesting that their transcription is switched off after entry into meiosis-I (Fig. EV1B,C). Importantly, however, although both poised and non-poised genes exhibited maximal gene body PRO-seq signal in SpI or RS compared with SpG, promoter PRO-seq signal of poised genes was >twofold higher than that of non-poised genes in SpG

(Figs. 1A and EV1C), consistent with the engaged Pol II peak near the TSSs of poised genes in SpG (Fig. EV1A).

To quantitatively assess pause-release across each cell stage, we calculated the polymerase pausing index (i.e., the density of PRO-seq signal at promoter relative to the PRO-seq density at gene body) for poised and non-poised genes. Intriguingly, pausing index for poised genes in SpG was ~twofold higher than in SpI and RS (two-sided wilcoxon matched-pairs signed-ranked sum test; ****$P < 0.0001$), whereas we did not observe a higher pausing index for non-poised genes in SpG when compared to SpI and RS (Fig. 1B). Metagene plots corroborated computed pausing indexes for poised and non-poised genes: In SpG, poised genes displayed high Pol II signal at promoters, but not at gene bodies, whereas proximal-promoter PRO-seq signal for non-poised genes in SpG was at the background level (Fig. 1C).

Pol II pausing at developmentally regulated genes requires accessible chromatin (Vihervaara et al, 2018). To test the idea that accumulation of paused Pol II at poised gene promoters is accompanied by accessible chromatin in SpG, we purified populations of SpG, leptonene/zygotene spermatocytes (L/Z), and pachytene/diplotene spermatocytes (P/D) from wild-type C57BL/6 mice with high purity using FACS (Appendix Fig. S2) and performed Assay for Transposase-Accessible Chromatin using sequencing (ATAC-seq). Intriguingly, in SpG, the chromatin around promoters of poised genes showed a strong ATAC-seq signal, which is sustained throughout prophase I, whereas the promoters of non-poised genes lacked ATAC-seq signal for accessible chromatin in this stage (Fig. 1D; Appendix Fig. S3A). However, the promoters of non-poised genes displayed ATAC-seq signal in only P/D, consistent with their expression at this stage (Fig. 1D; Appendix Fig. S3A). To ensure a robust conclusion that poised meiosis-I and spermiogenesis genes have accessible chromatin around their promoters in SpG cells, we located significant ATAC-seq peaks genome-wide using MACS3 (FDR < 0.01; Dataset EV2) and computed the distance from the TSSs of poised and non-poised genes to the nearest ATAC-seq peak in three biological replicates from each cell stage (Appendix Fig. S3B; agreement between replicates; SpG Spearman's $\rho > 0.95$, L/Z Spearman's $\rho > 0.95$, P/D Spearman's $\rho > 0.96$). Genes whose promoters exhibit ATAC-seq peaks within the ±2 kb of their TSSs in at least two replicates were considered as genes with an accessible promoter (Fig. 1E). Strikingly, in SpG, >90% of poised genes had ATAC-seq signal near their TSSs (poised meiosis-I genes, median distance from TSS to nearest ATAC-seq peak = 186 bp; poised spermiogenesis genes, median distance = 182 bp), whereas, the TSSs of ≤40% of non-poised genes displayed ATAC-seq peak (non-poised meiosis-I genes, median distance = 6456 bp; non-poised spermiogenesis median distance = 10,233 bp) (Fig. 1E; Dataset EV2). Finally, we asked whether, in both SpG and SpI, chromatin accessibility at promoters predicts the engaged Pol II peak around the TSSs of genes we categorized. We found moderate to strong correlation between the accumulation of Pol II signal and chromatin accessibility at meiosis-I and spermiogenesis gene promoters (Fig. 1F; Spearman's $\rho = 0.38–0.72$). Exemplified poised and non-poised genes corroborated these findings: poised gene promoters had both Pol II and ATAC-seq peaks in SpG, while non-poised promoters did not (Appendix Fig. S3C). We conclude that the majority of meiosis-I and spermiogenesis genes are marked by paused Pol II and accessible chromatin at their promoters in SpG

prior to the onset of their transcription during prophase I of meiosis.

## Accessible poised gene promoters retain a binding motif for NFYA that is highly expressed in pre-meiotic cells

Many poised genes retain accessible promoters that accumulate paused Pol II in SpG prior to their expression later during meiosis-I. This finding prompted us to decipher TFs that potentially regulate chromatin accessibility in SpG. We thus searched for enriched TF binding motifs under ATAC-seq peaks located around the promoters of poised genes in SpG (HOMER; Benjamini–Hochberg multiple test correction, FDR < 0.05). NFYA binding motif, RRCCAATSRS, was significantly enriched as one of the top-five scoring motifs beneath the ATAC-seq peaks around the promoters of poised genes in SpG (Fig. 2A; Appendix Fig. S4). The CCAAT pentanucleotide located near TSSs is bound by several TFs, including NFYA, CTF (CCAAT Transcription Factor), and C/EBP (CCAAT Enhancer Binding Protein) (Dolfini et al, 2025). Because an accumulated line of evidence has revealed the regulatory role of NFYA in accessible chromatin in other cellular contexts (Coustry et al, 2001; Fleming et al, 2013; Nardini et al, 2013; Oldfield et al, 2014; Sherwood et al, 2014; Oldfield et al, 2019), we sought to examine the molecular function of NFYA during spermatogenesis. Notably, our analysis identified other motifs that are bound by KLF1/3/5, SP1/2, DMRT1/6, and ELK3/4 (Appendix Fig. S4). Resolving the possible regulatory role of these TFs in chromatin accessibility and their relation with NFYA during spermatogenesis will be a promising research direction.

Consistent with the ubiquitous expression of NFYA (Maity and de Crombrugghe, 1998), the mouse Encyclopedia of DNA Elements (ENCODE) revealed broad expression of *Nfya* mRNA across many mouse tissues (Appendix Fig. S5A). To study the developmental expression of NFYA during spermatogenesis, we first measured protein abundance of *Nfya* across the testes of staged mice (Fig. 2B; Appendix Fig. S5B). By 9 to 11 days post-partum (dpp), the first cohort of spermatogenic cells enter prophase I and progress no further than Z. Hence, at 9 to 11 dpp, the majority of seminiferous tubules contain SpG and L/Z (Nebel et al, 1961). Intriguingly, NFYA abundance peaked at 11 dpp and was reduced in the testes of later stages (Fig. 2B; Appendix Fig. S5B), suggesting that NFYA is expressed higher in SpG or in L/Z compared to germ cells at later developmental stages. To better assess the timing of NFYA expression during spermatogenesis, we immunostained the tubules with antibodies against NFYA (Fig. 2C; Appendix Fig. S5C). To ensure robust conclusion from immunostaining, we used antibodies from two different companies (one is used for immunohistochemistry, the other is for immunofluorescence). Note that the specificity of antibodies used in this study was validated in *Nfya* mutant mice (Appendix Fig. S8D). The cycle of mouse seminiferous epithelium is divided into 12 stages (Oakberg, 1956; Ahmed and de Rooij, 2009), which provides an important map to understand the expression of a protein throughout germ cell development. In stage II tubules, type A SpG moderately expressed NFYA, while tubules at stages III and VI exhibited a high abundance of nuclear NFYA protein in differentiating SpG (i.e., intermediate and type B) and preleptotene spermatocytes (pL) that are the immediate descendants of SpG appearing just before the entry into prophase I (Fig. 2C). However, in the tubules at stages from VIII to XI, NFYA

expression was gradually decreased from the leptotene stage to diplotene stages of prophase I (Fig. 2C). Notably, even though we observed reduced expression of NFYA in primary meiocytes, RS showed moderate to strong nuclear NFYA in the stages II, III, and VI tubules (Fig. 2C). Consistently, immunofluorescence staining revealed that those germ cells residing near basal membrane—intermediate and type B SpG, as well as pL—had the highest NFYA signal in their nuclei when compared to those localized more towards the lumen—L, Z, P, and D (Appendix Fig. S5C). Moreover, immunostaining of NFYA in the testes sections from staged mice corroborated these findings: we detected a strong NFYA signal localized in the nuclei of pre-meiotic cells and spermatids, whilst the signal from meiocytes was either low or below the limit of detection (Appendix Fig. S5D). We conclude that NFYA, whose binding motif enriched in the promoters of poised genes, is expressed abundantly in SpG and pL.

## NFYA binds the promoters of poised genes in spermatogonia

To characterize the binding sites of NFYA genome-wide and to determine whether NFYA directly binds the promoters of poised genes in SpG, we performed CUT&RUN from FACS-purified SpG, L/Z, and P/D using antibodies against NFYA and IgG. We sequenced three biological replicates for each cell population (Appendix Fig. S6; agreement between replicates; Pearson correlation coefficient ($r$); SpG $r > 0.95$, L/Z Spearman's $r > 0.93$, P/D $r > 0.89$). We identified significant NFYA peaks, reflecting NFYA binding sites, genome-wide using MACS3 (FDR < 0.01; Dataset EV3). IgG CUT&RUN served as a control to map significant NFYA peaks. In SpG, 70% of all NFYA peaks were within the ±2 kb of annotated TSSs, whereas only 2.2% of genome-wide NFYA peaks resided in enhancer regions marked by histone modifications, H3K4me1 (Appendix Fig. S7A). Similar observations were obtained in L/Z and P/D, suggesting that NFYA primarily functions at the promoter-proximal regions of genes (Appendix Fig. S7A). Given that NFYA is expressed abundantly in pre-meiotic cells and that its expression reduces after meiotic entry (Fig. 2C; Appendix Figs. S5C,D), we next examined the change in NFYA occupancy throughout pre-meiotic and meiotic stages. We observed that NFYA occupancy at TSSs was higher in pre-meiotic SpG than in L/Z and P/D (Appendix Fig. S7B). Consistently, our genome-wide analysis identified 4020 protein-coding and 360 noncoding genes with an NFYA peak within ±2 kb of their TSSs in at least two replicates of SpG, whereas we found a lesser number of genes bound by NFYA in L/Z and P/D (Appendix Fig. S7C; Dataset EV3).

We next sought to directly test that the promoters of poised genes expressed after meiotic entry are bound by NFYA in SpG. Remarkably, we detected strong NFYA occupancy around the TSSs of poised genes in SpG, whereas non-poised genes had no NFYA signal around their TSSs (Figs. 2D and EV2A). NFYA occupancy at poised gene promoters in SpG (median = 9.1 and 7.9 rpm; meiosis-I and spermiogenesis genes, respectively) was indeed significantly higher than in L/Z (median = 4.7 and 4.5 rpm; meiosis-I and spermiogenesis genes, respectively) and P/D (median = 5.9 and 5.6 rpm; meiosis-I and spermiogenesis genes, respectively), consistent with the higher expression of NFYA in SpG than in L/Z and P/D (Fig. EV2B; Mann–Whitney–Wilcoxon $U$ test). We next examined the correlation between NFYA occupancy and chromatin

accessibility at the promoters of poised and non-poised genes in SpG and P/D. Strikingly, we found a strong correlation between NFYA occupancy and accessible chromatin at poised gene promoters (Fig. 2E; Spearman's $\rho = 0.78$–0.86). In fact, in SpG, the signal for both ATAC-seq and NFYA CUT&RUN at poised gene promoters was higher compared to non-poised gene promoters (Fig. 2E). Note that although non-poised gene promoters displayed accessible chromatin in P/D, NFYA occupancy was not prominent, suggesting that the regulation of permissive chromatin at non-poised gene promoters is likely through another factor that may be expressed at the pachytene stage of prophase I (Fig. 2E).

To determine the number of unambiguous NFYA-bound genes for each gene category, we computed the distance from the TSSs of poised and non-poised genes to the nearest NFYA peak in three biological replicates. Those genes with NFYA peak within the ±2 kb of their TSSs in at least two replicates were considered NFYA-bound genes. Notably, the promoters of ~40% of poised genes were bound by NFYA in SpG (Fig. 2F; 1001 of 2415 poised meiosis-I genes with median distance from TSS to the nearest NFYA peak = 302 bp; 575 of 1488 poised spermiogenesis genes with median distance = 251 bp). On the contrary, only ≤5% of non-poised gene promoters were occupied by NFYA, consistent with their closed chromatin state at SpG (Fig. 2F; 35 of 647 non-poised meiosis-I genes; 19 of 856 non-poised spermiogenesis genes). Corroborating the reduced expression of NFYA after meiotic entry, the promoters of <25% of poised genes were occupied by NFYA in L/Z or P/D (Fig. 2F). Among the 1181 mitosis genes, NFYA bound to the promoters of 28% in SpG (Fig. 2F; median distance = 295 bp). Finally, examination of PRO-seq, ATAC-seq, and NFYA CUT&RUN data on exemplified poised genes revealed an accumulation of paused Pol II at accessible promoters, which is occupied by NFYA in SpG (Fig. EV2C). Together, our data suggest that prior to meiosis, NFYA may regulate accessible chromatin at the promoters of poised genes that are expressed after meiotic entry.

## Male germ cell-specific deletion of *Nfya* severely impairs spermatogenesis

Prior to meiotic entry, NFYA—expressed highly in differentiating SpG and pL—binds the promoters of poised genes whose expression is in fact turned on at the pachytene stage of prophase I. To understand the biological role of NFYA in meiotic entry and progression in a C57BL/6J background, we generated *Nfya*<sup>fl/fl</sup>; *Stra8-Cre* mice in which exons 5–6 of *Nfya* is deleted specifically in germ cells by Cre recombinase (Appendix Fig. S8A–C; henceforth *Nfya-CKO*). *Stra8* expression is active in undifferentiated and differentiating SpG, and pL (Zhou et al, 2008), *Stra8-Cre* mice generated in C57BL/6J background by Cyagen thus express *P2A-ZsGreen1-T2A-Cre* cassette upstream of the stop codon of *Stra8* in pre-meiotic germ cells (Appendix Fig. S8A–C; "Methods"). Western blotting and immunohistochemistry analysis of staged mice testes confirmed the depletion of NFYA in germ cells, but not somatic cells (Appendix Fig. S8D,E). Consistently, our immunofluorescence staining for SOX9, a marker for Sertoli cells (Li et al, 2014), revealed a comparable number of SOX9-positive Sertoli cells between *Nfya-CKO* and control *Stra8-Cre*<sup>Cre/+</sup> males (henceforth *Cre/+*) (Appendix Fig. S8F).

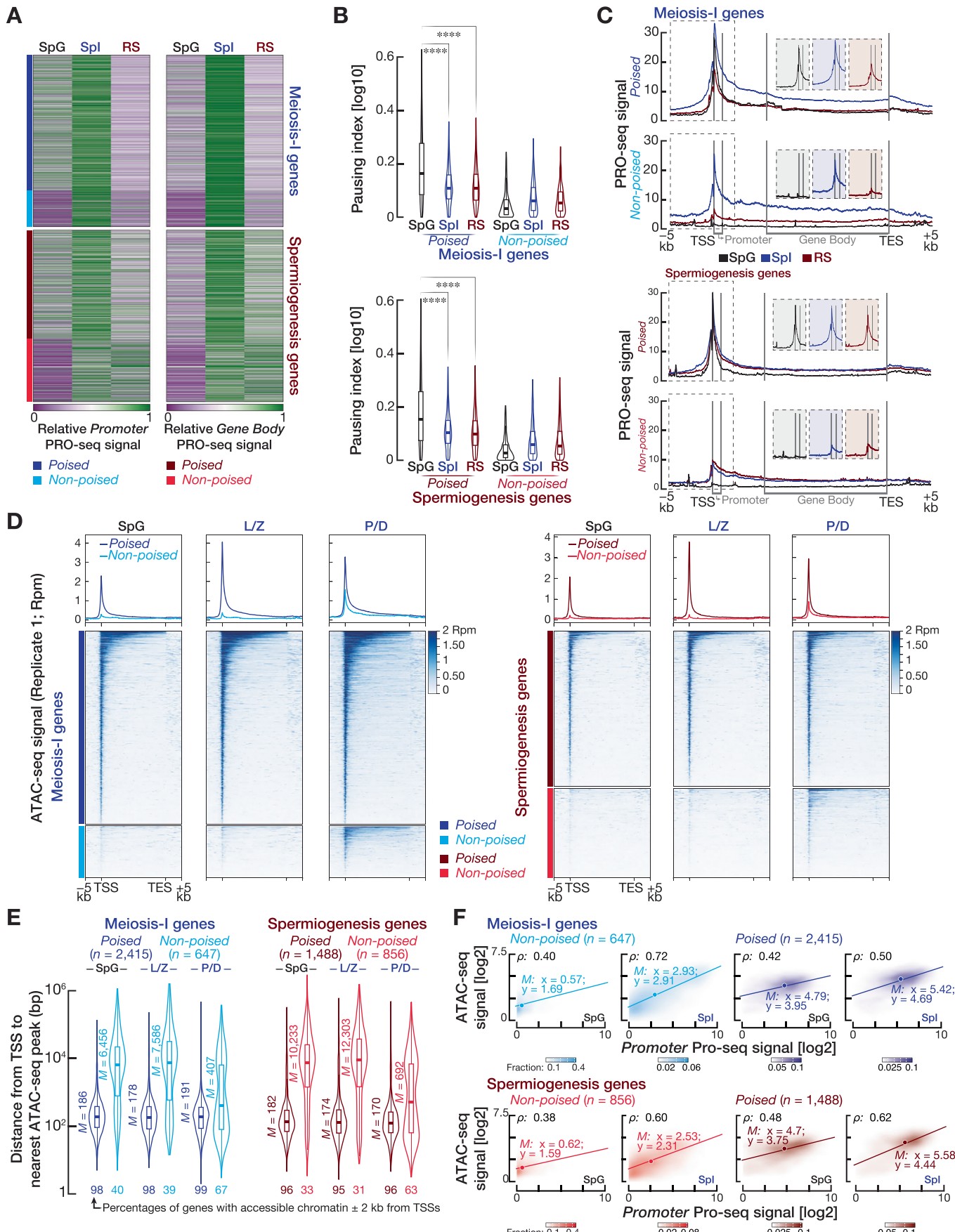

Figure 1.  Promoter-proximal regions of poised genes expressed during meiosis accumulate paused Pol II and have accessible chromatin in spermatogonia.

(A) Heatmaps show the relative promoter PRO-seq signal (left; from transcription start site [TSS] to first 5% of gene body) and relative gene body PRO-seq signal (right; from the first 30% of the gene body to transcription end site [TES]) across spermatogonia (SpG), primary spermatocytes (SpI), and round spermatids (RS) cells. (B) Violin plots (top, meiosis-I genes; bottom, spermiogenesis genes) of pausing index calculated from the ratio of promoter PRO-seq density to gene body PRO-seq density on poised and non-poised genes. ****$P < 0.000$; Paused meiosis-I genes in SpG vs SPI $P < 2.2 \times 10^{-16}$, SpG vs RS $P < 2.2 \times 10^{-16}$; Paused spermiogenesis genes in SpG vs SPI $P = 2.2 \times 10^{-16}$, SpG vs RS $P < 2.2 \times 10^{-16}$; two-sided wilcoxon matched-pairs signed-ranked sum test. Horizontal lines represent the median. Whiskers show maximum and minimum values. IQR is represented by boxplots. Plotted pausing indexes are the average of three biological replicates from SpG, SpI, and RS cells. (C) Metagene plots (left two plots, meiosis-I genes; right two plots, spermiogenesis genes) of average PRO-seq signals of three biological replicates from SpG, SpI, and RS cells at annotated gene boundaries for poised and non-poised genes. Insets show PRO-seq density for SpG, SpI, and RS at the promoters (from TSS to first 5% of gene length). (D) Metagene plots (top) and heatmaps (bottom) show reads per million (Rpm)-normalized ATAC-seq signals in the −5 kb to +5 kb window flanking TSSs and TESs of poised and non-poised genes for spermatogonia (SpG), leptotene/zygotene (L/Z), and pachytene/diplotene (P/D) cells. (E) Distance (average of three biological replicates) from the TSS to the nearest ATAC-seq peak for poised and non-poised meiosis-I and spermiogenesis genes in SpG, L/Z, and P/D cells. Genes were classified as genes with accessible chromatin at promoters if they had significant ATAC-seq peak ± 2 kb from their TSSs in at least two biological replicates of the ATAC-seq experiment. Vertical lines represent the median. Whiskers show maximum and minimum values. IQR is represented by boxplots. (F) Scatter plots show the Spearman's correlation ($\rho$) between ATAC-seq density and PRO-seq density at the promoters of poised and non-poised meiosis-I genes and spermiogenesis genes for SpG and SpI cells. Each data point represents the average of three biological replicates.

In *Nfya-CKO* male mice, we observed no abnormalities in organs other than the testis. *Nfya-CKO* males had significantly smaller testes compared with *Cre/+* (Appendix Fig. S9A,B). Histological analysis of the epididymis from 8-week-old mice additionally revealed a lack of sperm in *Nfya-CKO* males (Appendix Fig. S9C). We thus conclude that *Nfya-CKO* males are infertile.

## NFYA-deficient germ cells arrest at meiotic entry

To gain understanding of the defective spermatogenesis in *Nfya-CKO* males, we histologically examined the testes at 9, 14, and 30 dpp (Fig. 3A). In *Cre/+* mice, by 9 dpp, the first cohort of spermatogenic cells reached L; at 14 dpp, early P appeared; while in 30 dpp, spermatogenesis reached to spermatozoa (Fig. 3A). Notably however, in *Nfya-CKO* males, spermatogenesis progressed no further than spermatocytes resembling preleptotene spermatocytes (Fig. 3A). We next immunostained the tubules for DDX4, a marker for both male and female germ cells (Fujiwara et al, 1994). Here, by 9 dpp, spermatogenesis reaches no later than L—accounting for the smallest fraction of all germ cells (Nebel et al, 1961). Hence, we observed a comparable number of DDX4-positive germ cells between *Nfya-CKO* and *Cre/+* mice at 9 dpp (Appendix Fig. S10A). However, at 14 and 30 dpp, DDX4-positive germ cells were markedly reduced in *Nfya-CKO* males when compared with *Cre/+* mice (Appendix Fig. S10A). In the testis sections from *Nfya-CKO* males, we observed germ cells displaying morphological characteristics of dying cells (Fig. 3A). Subsequently, we sought to examine the fate of defective germ cells in *Nfya-CKO* by performing the terminal dUTP nick-end labeling (TUNEL) staining. We found that the seminiferous tubules of *Nfya-CKO* testis had a significantly higher number of TUNEL-positive germ cells compared to *Cre/+* testis, suggesting that defective germ cells in *Nfya-CKO* testis may be eliminated in part by apoptosis (Appendix Fig. S10B).

Spermatogenic defect appears to occur at meiotic entry rather than during meiotic progression, as judged from the morphology of germ cells which resemble preleptotene spermatocytes (Fig. 3A). To substantiate this observation, we immunostained the tubules with an antibody against HSPA2, a testis-specific member of HSP70 family that is expressed from pachytene spermatocytes and onward (Scieglinska and Krawczyk, 2015). We found that HSPA2-positive germ cells were absent in *Nfya-CKO* testis (Appendix Fig. S10C).

Moreover, immunostaining for peanut agglutinin (PNA) further confirmed the absence of round and elongating spermatids in *Nfya-CKO* testis (Appendix Fig. S10D). Together, our data demonstrate that NFYA-deficient spermatocytes—that resemble pL—fail to progress through prophase I.

Formation of DNA double-strand break (DSB) and waves of expression of components of axial elements (AE) are hallmarks of meiotic initiation (Ishiguro, 2024). We thus performed immunostaining for γH2AX, a marker for DSB, and SYCP3, a component of AE, to directly test that NFYA-deficient germ cells reach pL stage but fail to progress through prophase I. γH2AX staining demonstrated the presence of DSBs in a greater number of spermatocytes from *Cre/+* testis when compared to those from *Nfya-CKO* testis. Nonetheless, we detected a γH2AX signal in the nuclei of spermatocytes resembling pL (Fig. 3B; Appendix Fig. S10E). Moreover, the nuclei of spermatocytes resembling preleptotene cells showed a SYCP3 signal (Fig. 3C). To further substantiate these observations, we performed double staining for γH2AX and SYCP3 on chromosome spreads from the cells of *Nfya-CKO* and *Cre/+* testis. While germ cells successfully progressed through meiosis—as evidenced by the presence of L, Z, P and D—and developed into spermatozoa in *Cre/+* testis, spermatocytes later than the pL stage did not appear in *Nfya-CKO* testis (Fig. 3D). Notably, spermatocytes resembling pL showed an aggregated SYCP3 signal, suggesting incomplete AE formation (Fig. 3D). We next performed immunostaining for STRA8—expressed highly in pL (Zhou et al, 2008)—along with γH2AX to evaluate whether NFYA is required for meiotic entry. Intriguingly, some γH2AX-positive spermatocytes from *Nfya-CKO* testis showed nuclear STRA8 signal even though those cells were low in number (Fig. 3E). Both *Stra8* KO and *Meiosin* KO spermatocytes revealed precocious mitotic status (Mark et al, 2008; Ishiguro et al, 2020). To test whether NFYA-deficient spermatocytes resembling preleptotene cells underwent mitotic chromosome condensation, we performed double staining for H3S10P—H3Ser10 phosphorylation, a marker for mitotic chromosome condensation (Hendzel et al, 1997)—and γH2AX. Notably, some γH2AX-positive spermatocytes from *Nfya-CKO* testis revealed H3S10P staining, whereas strong H3S10P signal on centromeres was confined only in spermatocytes at Metaphase I in *Cre/+* mice, suggesting that NFYA-deficient spermatocytes resembling pL fail to properly transit from mitosis to meiosis-I, but rather retain mitotic characteristic (Fig. 3F).

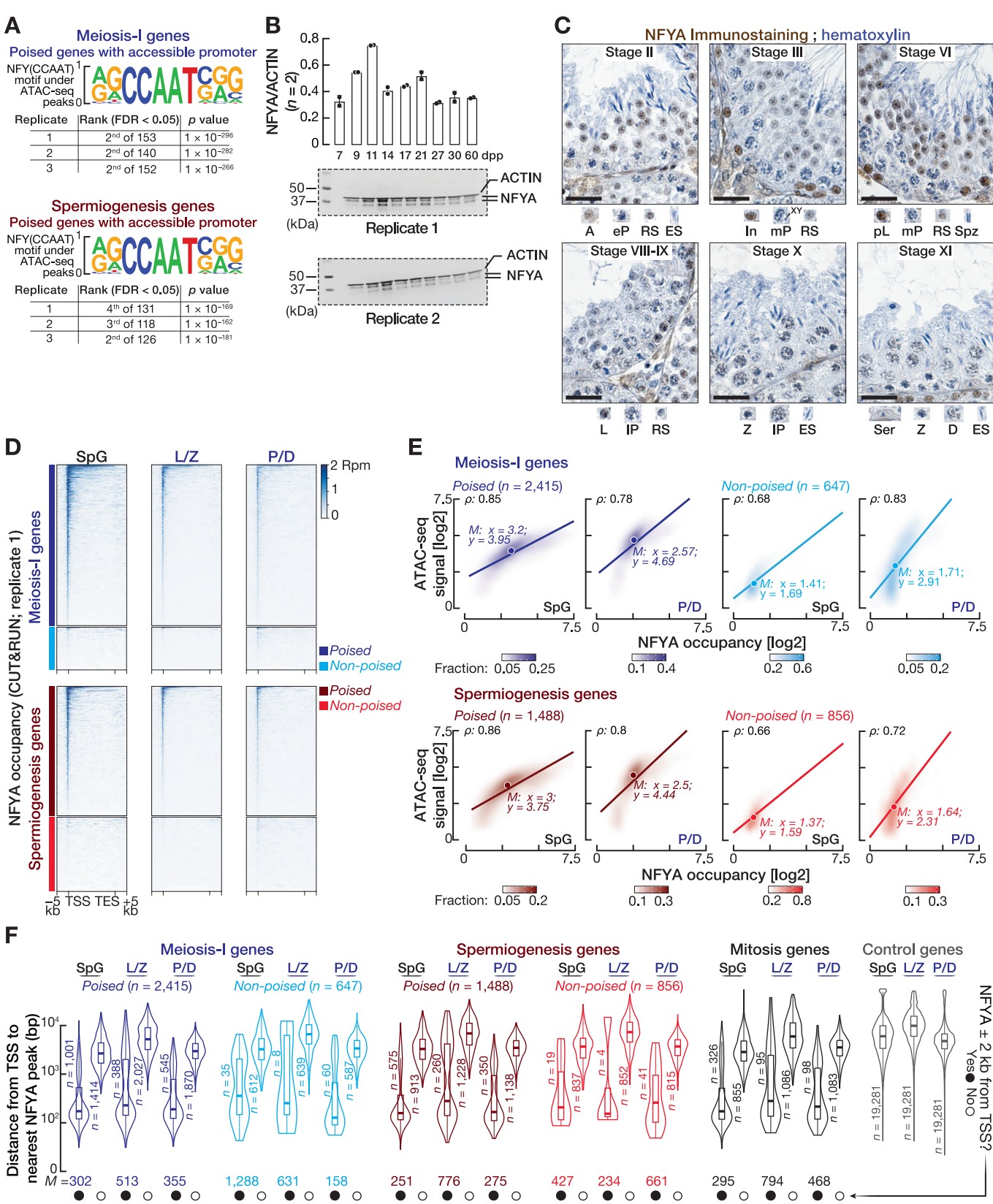

**Figure 2. NFYA expressed highly in pre-meiotic cells binds the accessible promoters of poised genes in spermatogonia.**

(A) HOMER (Heinz et al, 2010) (Benjamini–Hochberg multiple test correction, FDR < 0.05) identified a sequence motif for NFYA under ATAC-seq peaks located around the promoters of poised genes (top: meiosis-I genes; bottom: spermiogenesis genes) in SpG. Promoters of control genes ($n = 19,281$), whose transcript abundance remained constant across germ cells, served as background. (B) Abundance of NFYA protein across the testes of staged mice. ACTIN served as a loading control. Each lane contained 75 µg testis protein. Bars represent the mean protein abundance of *Nfya* from two independent replicates ($n = 2$). Whiskers show standard deviation. Quantification of NFYA and ACTIN bands was performed using ImageJ2 v2.14. See Appendix Fig. S6B for uncropped western blot images. (C) Immunohistochemical detection of NFYA in a testis section from a 3-month-old mouse. Seminiferous tubules at different epithelial stages are presented. A type A SpG, In intermediate SpG, pL preleptotene, L leptotene, Z zygotene, eP early pachytene, mP middle pachytene, lP late pachytene, D diplotene, RS round spermatid, ES elongating spermatid, Spz spermatozoa, Ser sertoli. Scale bars: 25 µm. (D) Heatmaps show reads per million (Rpm)-normalized NFYA CUT&RUN signals in the −5 kb to +5 kb window flanking transcription start sites (TSSs) and transcription end sites (TESs) of poised and non-poised genes for spermatogonia (SpG), leptotene/zygotene (L/Z), and pachytene/diplotene (P/D) cells. Heatmaps for the second and third replicates are in Fig. EV2A. (E) Scatter plots show the Spearman's correlation ($\rho$) between ATAC-seq density and NFYA occupancy at the promoters of poised and non-poised meiosis-I and spermiogenesis genes. Each data point represents the average of three biological replicates. (F) Distance (average of three biological replicates) from the TSS to the nearest NFYA peak for poised and non-poised meiosis-I and spermiogenesis genes, mitosis genes, and control genes in SpG, L/Z, and P/D cells. Genes were classified as NFYA-bound genes if they had a significant NFYA peak ±2 kb from their TSSs in at least two biological replicates of CUT&RUN experiment. Vertical lines represent the median. Whiskers show maximum and minimum values. IQR is represented by boxplots. Source data are available online for this figure.

Together, we conclude that NFYA-deficient germ cells developed into pL, but stalled there to progress through meiosis.

## *Nfya* deletion impairs spermatogonial differentiation

NFYA is expressed abundantly in SpG and pL (Fig. 2; Appendix Fig. S5) and the promoters of ~one-third of genes required for the mitotic stage are bound by NFYA (Fig. 2F). Yet, spermatogenesis arrests at the pL stage in *Nfya-CKO* testis. We thus examined whether spermatogonial maintenance and differentiation occur normally in *Nfya-CKO* males. We performed triple staining for SALL4, c-Kit, and GATA4 on the testis sections from *Nfya-CKO* and *Cre/+* mice. SALL4 is a marker for undifferentiated SpG (Hobbs et al, 2012), while c-Kit is for differentiating SpG (Yoshinaga et al, 1991). GATA4 is essential for the function of Sertoli cells (Kyrönlahti et al, 2011). Given that Sertoli cells were unaffected in *Nfya-CKO* compared to *Cre/+* males (Appendix Fig. S8F), we normalized the number of SALL4-positive undifferentiated SpG or c-Kit-positive differentiating SpG to the number of GATA4-positive Sertoli cells. We found a comparable number of SALL4-positive undifferentiated SpG normalized to Sertoli cells between *Nfya-CKO* and *Cre/+* mice, whereas the number of c-Kit-positive differentiating SpG was reduced ~threefold in *Nfya-CKO* compared to *Cre/+* (Fig. 3G). These findings suggest that while the maintenance of undifferentiated SpG occurs normally within *Nfya-CKO* testis, deletion of *Nfya* results in compromised differentiation of SpG.

## Simultaneous scRNA-seq and scATAC-seq reveals lack of meiocytes in *Nfya-CKO*

To directly test whether the arrest at meiotic entry in *Nfya-CKO* mice is due to the disruption of chromatin accessibility at meiotic promoters in SpG, we performed simultaneous single-cell RNA-seq (scRNA-seq) and single-cell assay for transposase accessible chromatin sequencing (scATAC-seq) on the nuclei from the testes of 8-week-old *Cre/+* and *Nfya-CKO* mice, as well as 11, 17, and 25 dpp wild-type mice using BD Rhapsody Single-Cell Analysis System. For scRNA-seq, a total of 45,250 individual cells were captured. We detected an average of 1,977 genes per cell and an average of 5,850 copies of transcripts (unique molecular indices, UMIs) per cell (Dataset EV4A). We performed uniform manifold

approximation and projection (UMAP) using Seurat package (Hao et al, 2021) for dimension reduction analysis on the combined datasets. Independent of their origin of dataset, cells formed 32 clusters, indicating minimal batch effect (Appendix Fig. S11A). Using previously described cell-type specific markers (Appendix Fig. S11B; Dataset EV4B), we further resolved 32 clusters into 13 cell populations: nine major germ cell populations—undifferentiated spermatogonia (undiff. SpG), differentiating spermatogonia (diff. SpG), preleptotene spermatocytes (pL), early leptotene, leptotene, zygotene spermatocytes (eL/L/Z), leptotene, zygotene, early pachytene spermatocytes (L/Z/eP), middle and late pachytene spermatocytes (mP/lP), diplotene spermatocytes (D), round spermatids (RS), and elongating spermatids (ES)—; two cell populations from *Nfya-CKO* mice—SpG and somatic cells—; Sertoli cells; and unknown cells (Fig. 4A–C).

Differences for cell-type composition across the testes from staged and *Cre/+* and *Nfya-CKO* mice recapitulated both the timeline of germ cell development throughout stages and our observed phenotype for *Nfya-CKO* mice (Fig. 3). Developing germ cell populations agreed well with the expected developmental stage (Nebel et al, 1961). At 11 dpp, the most developed germ cell population was eL/L/Z, whereas 17 dpp reached L/Z/eP. By 25 dpp, we detected all kinds of spermatocytes, as well as haploid spermatids. Testis of 8-week-old *Cre/+* contained entire germ cell populations, while cell populations from *Nfya-CKO* mice occupied distinct locations in UMAP space, reflecting the absence of spermatocyte populations. In fact, the major cell populations were somatic cells and SpG, consistent with the spermatogenic arrest at meiotic entry observed in *Nfya-CKO* mice (Figs. 3 and 4A–C).

We next analyzed the scATAC-seq dataset of our simultaneous sc-multiomics experiment. After filtering low-quality cells and doublets, we captured 343,866 cells with high-quality accessible chromatin for downstream analysis (average TSS enrichment score per cell = 11.3; Dataset EV4C). Through clustering based on a genome-wide tile matrix of 500 bp using ArchR (Granja et al, 2021) on the combined cells from the testes of both 8-week-old *Cre/+* and *Nfya-CKO* mice, as well as 11, 17, and 25 dpp wild-type mice, our analysis identified 25 clusters (Appendix Fig. S11C). To annotate the cell types from 25 clusters, we used our scRNA-seq-based cell-type annotation for cells with both accessible chromatin and captured transcriptome (Appendix Fig. S11D). Here, our analysis resolved 25 clusters into nine cell populations eL/L/Z, L/Z/eP,

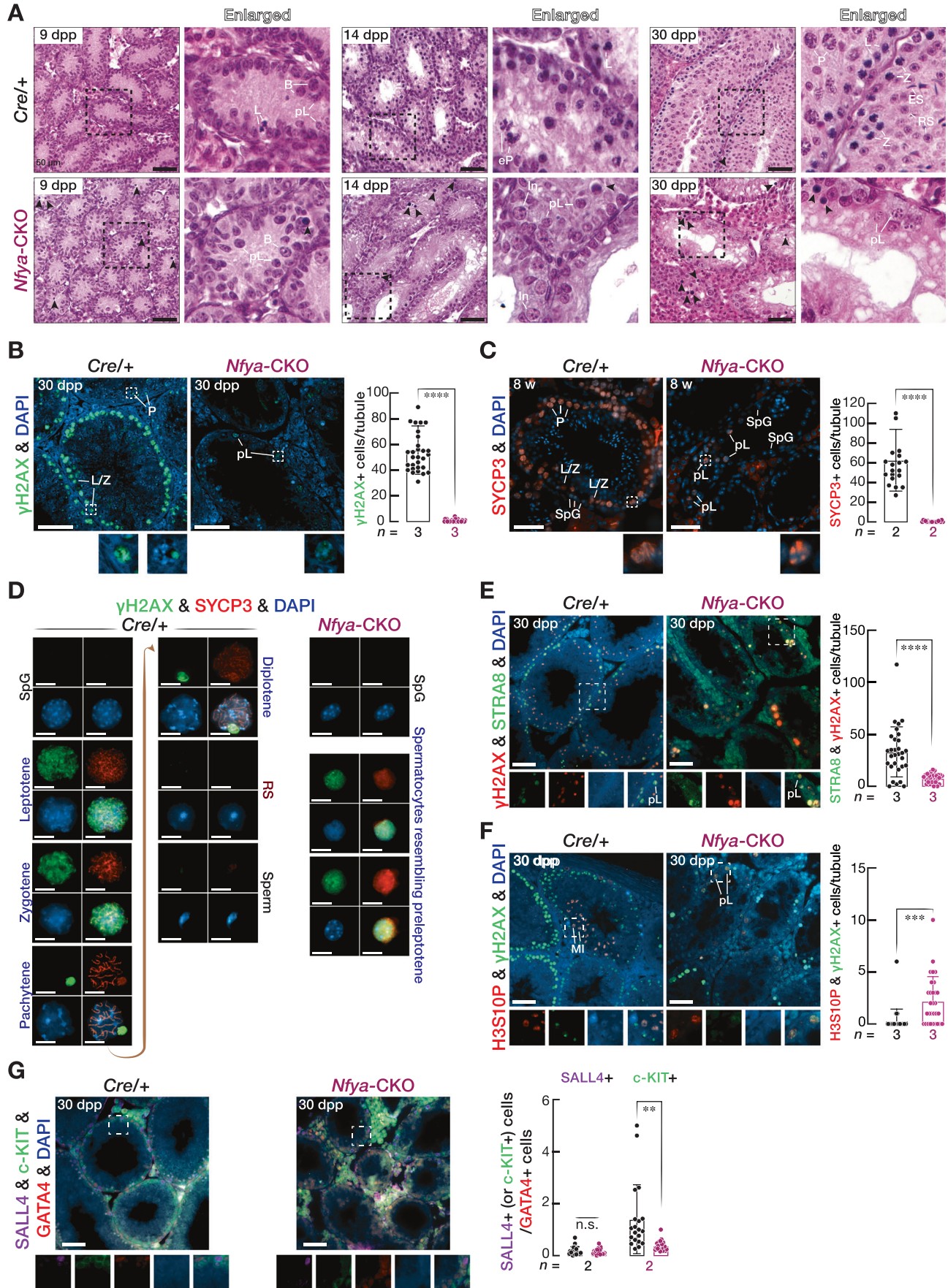

**Figure 3.  Germline-conditional deletion of *Nfya* blocks meiotic entry.**

(A) Hematoxylin and eosin staining of the sections from *Cre/+* and *Nfya-CKO* mice at 9 dpp, 14 dpp, and 30 dpp. Right: Enlarged images. Black arrowheads depict dying cells. B type B SpG, In intermediate SpG, pL preleptotene, L leptotene, Z zygotene, eP early pachytene, P pachytene, RS round spermatid, ES elongating spermatid. Scale bars: 50 μm. (B, C) Immunofluorescence staining of the sections from 30 dpp and 8-week-old *Cre/+* and *Nfya-CKO* mice for (B) γH2AX ($n = 3$; $P = 6.34 \times 10^{-23}$) and (C) SYCP3 ($n = 2$; $P = 8.52 \times 10^{-11}$). All images are representative of two or three biological replicates. Bars represent quantification for positive cells per tubule from two or three independent biological replicates for each staining. Whiskers show the standard deviation, and the center of the whiskers represents the mean. Significance is measured using two-sided unpaired Student $t$ test. ****$P < 0.0001$. Scale bars: 50 μm. (D) Immunofluorescence staining for γH2AX and SYCP3 from the chromosome spreads of *Cre/+* and *Nfya-CKO* germ cells. Scale bars: 10 μm. (E–G) Immunofluorescence staining of the sections from 30 dpp and 8-week-old *Cre/+* and *Nfya-CKO* mice for (E) γH2AX and STRA8 ($n = 3$; $P = 4.72 \times 10^{-7}$), (F) H3S10P and γH2AX ($n = 3$; $P = 1.87 \times 10^{-4}$; MI, first meiotic division; pL, preleptotene) and (G) SALL4, c-KIT and GATA4 ($n = 3$; $P = 1.48 \times 10^{-3}$). All images are representative of three biological replicates. Bars represent quantification for positive cells per tubule from two or three independent biological replicates for each staining. Whiskers show the standard deviation, and the center of the whiskers represents the mean. Significance is measured using two-sided unpaired Student $t$ test. **$P < 0.01$, ***$P < 0.001$, ****$P < 0.0001$. Scale bars: 50 μm.

mP/lP, RS, somatic, Sertoli, undiff. SpG, undiff. SpG from *Nfya-CKO*, and unknown (Fig. 4D). Cell populations defined by scATAC-seq agreed well with the cell populations obtained from scRNA-seq (Appendix Fig. S11D,E). Determination of chromatin accessibility around the TSSs of key cell-type-specific marker genes further corroborated the annotation of major cell populations (Fig. 4E). Finally, cell-type composition differences in each dataset recapitulated the developmental trajectory of germ cell and the spermatogenic arrest in *Nfya-CKO* mice (Appendix Fig. S11F,G). In the testes of wild-type staged and 8-week-old *Cre/+* mice, we observed step-wise development of male germ cells. Whereas, in 8-week-old *Nfya-CKO* testes, the major cell populations we captured were somatic cells and SpG (Appendix Fig. S11F,G). Even though 8 weeks possibly limits the depth for detection of spermatogonia in *Cre/+* mice, our systematic analysis of simultaneous scRNA-seq and scATAC-seq demonstrated continuous development of germ cells in wild-type and *Cre/+* testes, and provided further supporting evidence that spermatogenesis arrests at meiotic entry in *Nfya-CKO* mice.

### *Nfya* deletion impairs chromatin accessibility and paused Pol II occupancy at the promoters of poised genes in spermatogonia

We defined three gene categories based on the change in absolute transcript abundance of genes across four major FACS-purified bulk germ cell populations (Appendix Fig. S1A). Supporting our classification, unsupervised hierarchical clustering based on the expression of mitosis, meiosis-I, and spermiogenesis genes from nine cell populations captured by scRNA-seq revealed their temporal expression across spermatogenesis (Fig. 5A). The expression of mitosis genes was highest in undif. SpG and sharply reduced in early meiocytes (i.e., L/Z/eP). In contrast, transcripts of meiosis-I genes first accumulated in early meiocytes (i.e., eL/L/Z); peaked in mP/lP; and substantially declined in RS. The expression of spermiogenesis genes coincided with meiosis-I genes yet persisted in spermatids (Fig. 5A,B).

Prior to the initiation of meiosis, in pre-meiotic cells, NFYA occupies the promoters of genes that remain poised for activation later during meiosis. To directly test the idea that NFYA regulates chromatin accessibility in pre-meiotic cells, we examined whether *Nfya* deletion impairs chromatin accessibility at meiotic promoters in SpG. We analyzed changes in chromatin accessibility around the promoters of NFYA-bound and -unbound gene categories in undiff. SpG from *Cre/+* and *Nfya-CKO* mice (Fig. 5C). We found

that the relative scATAC-seq signal at the promoters of NFYA-bound poised meiosis-I and spermiogenesis genes was reduced in NFYA-deficient undifferentiated SpG compared to *Cre/+* mice (Fig. 5C; Dataset EV4D). In contrast, ATAC-seq signal at the promoters of NFYA-unbound poised genes remained almost unchanged between undifferentiated SpG from *Cre/+* and *Nfya-CKO* mice (Fig. 5C; Dataset EV4D). In fact, chromatin accessibility at the promoters of NFYA-bound poised genes was significantly reduced in undiff. SpG from *Nfya-CKO* (*Nfya-CKO* ÷ *Cre/+* = ~0.75) (Fig. 5D; Two-sided Wilcoxon matched-pairs signed-ranked sum test; ****$P < 0.0001$), whereas we observed a minor change in chromatin accessibility at the promoters of NFYA-unbound poised genes (*Nfya-CKO* ÷ *Cre/+* = ~0.93). We found a strong correlation between NFYA occupancy and the degree of chromatin accessibility at poised promoters in wild-type SpG (Fig. 2E). We thus further examined the change in chromatin accessibility at NFYA-bound promoters between *Nfya-CKO* and *Cre/+* relative to their NFYA occupancy. After rank ordering the NFYA-bound promoters into four quartiles based on their NFYA occupancy (Fig. EV3A), we found a more pronounced reduction in accessible chromatin at promoters, which displayed the highest NFYA occupancy, than those promoters with the least NFYA occupancy (Figs. EV3B–D; fourth quartile, *Nfya-CKO* ÷ *Cre/+* = 0.66–0.69; first quartile, *Nfya-CKO* ÷ *Cre/+* = 0.81–0.85; Two-sided Wilcoxon matched-pairs signed-ranked sum test; ****$P < 0.0001$).

After the unphosphorylated Pol II is recruited to accessible promoters (Chen et al, 2018a), phosphorylation of the C-terminal domain of the largest subunit of Pol II at serine-5 (Pol II Ser5P) allows Pol II to escape pre-initiation complex and synthesizes 20–60 nt of RNAs, and finally Pol II Ser5P undergoes promoter-proximal pausing by the DRB Sensitivity-Inducing Factor (DSIF) and the Negative Elongation Factor (NELF) (Core et al, 2008; Kwak et al, 2013; Chen et al, 2018a; Vihervaara et al, 2023). We therefore performed CUT&Tag (Henikoff et al, 2020) for Pol II Ser5P from FACS-purified SpG of *Nfya-CKO* and *Cre/+* mice to test whether reduced chromatin accessibility impairs paused Pol II occupancy at poised promoters in NFYA-deficient SpG. We performed three biological replicates from *Nfya-CKO* and *Cre/+* (Fig. EV4A; agreement between replicates; SpG from *Nfya-CKO* Spearman's $\rho > 0.92$, SpG from *Cre/+* Spearman's $\rho > 0.95$). We first quantified the density of Pol II Ser5P around the TSS relative to the Pol II Ser5P density at the gene body for poised and non-poised meiosis-I and spermiogenesis genes, and mitosis genes in SpG from *Cre/+* (Fig. EV4B). Both poised meiosis-I and spermiogenesis genes displayed higher Pol II Ser5P accumulation at promoters compared

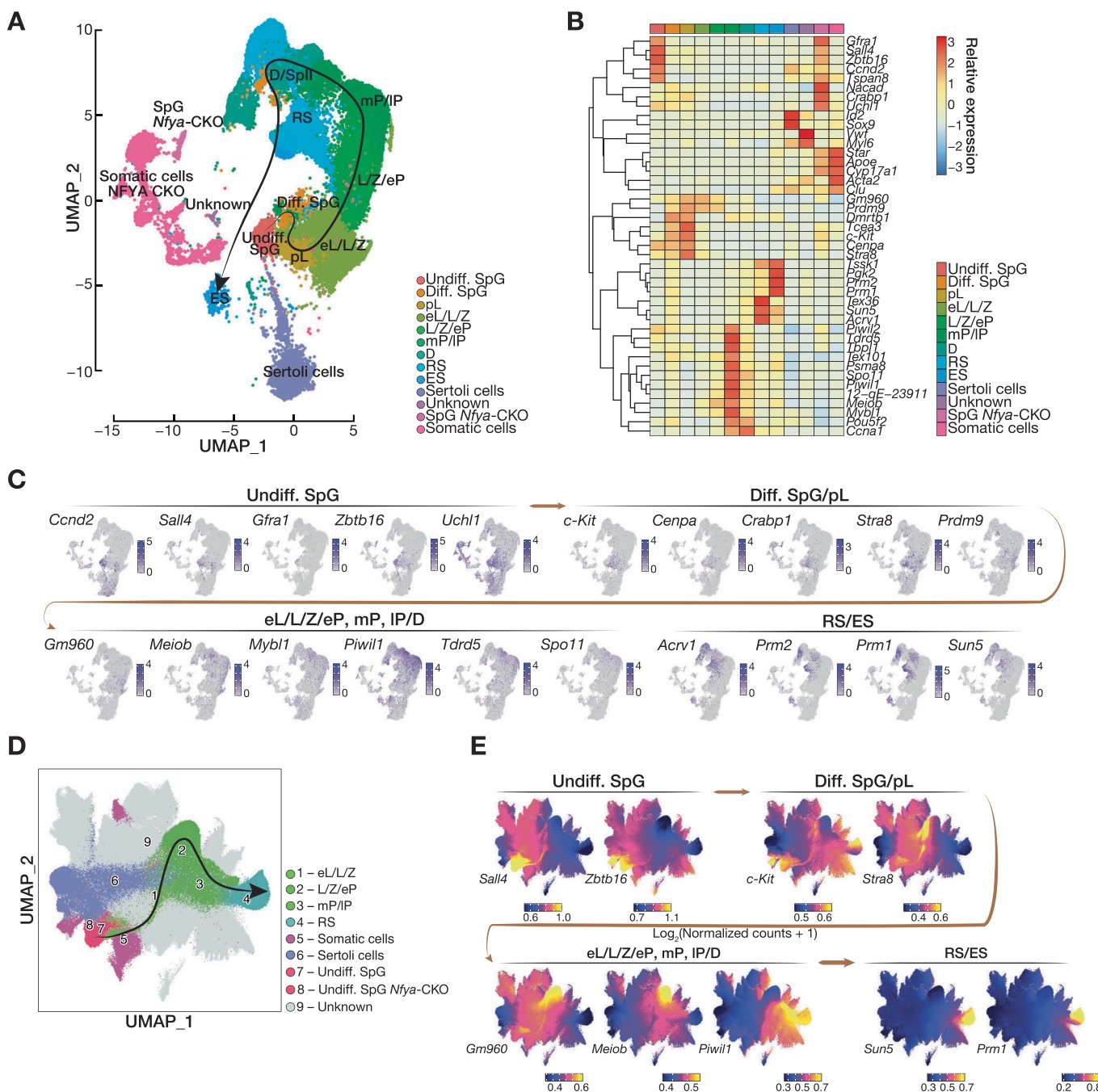

**Figure 4. Identifying cellular diversity of testes from 8-week-old *Cre/+* and *Nfya*-CKO mice along with wild-type staged mice using simultaneous single-cell (sc) RNA-seq and ATAC-seq.**

(A) Dimension reduction analysis by uniform manifold approximation and projection (UMAP) on 45,250 individual cells captured from all datasets illustrates 13 major cell populations in different colors. Major cell populations are defined from 32 clusters using the spatiotemporal expression of previously described cell-type-specific marker genes. See Appendix Fig. S11A for the full set of 32 clusters. Undiff. SpG undifferentiated SpG, Diff. SpG differentiating SpG, pL preleptotene, eL/L/Z early leptotene, leptotene, and zygotene, L/Z/eP leptotene, zygotene, early pachytene, mP/lP middle and late pachytene, D diplotene, RS round spermatid, ES elongating spermatid, Sertoli cells; Unknown cells; SpG *Nfya*-CKO, SpG from *Nfya*-CKO; other somatic cells. (B) Heatmap illustrates the relative expression levels of all marker genes included in the study across 13 annotated cell populations. (C) Per-cell gene expression patterns for exemplified known cell-type-specific marker genes are visualized in UMAP space. Transitions between major cellular states are highlighted in brown color. (D) UMAP space visualizes nine cell populations as defined by the chromatin accessibility state of 343,866 individual cells obtained from scATAC-seq. Major cell populations are defined from the 25 clusters by the guidance from scRNA-seq-based cell-type annotation, as well as the chromatin state of marker genes. See Appendix Fig. S11C for the full set of 25 clusters. (E) Dynamic change in permissive chromatin state around the promoters of exemplified known cell-type specific marker genes in individual cells is illustrated in UMAP space. Transitions between major cellular states are highlighted in brown color.

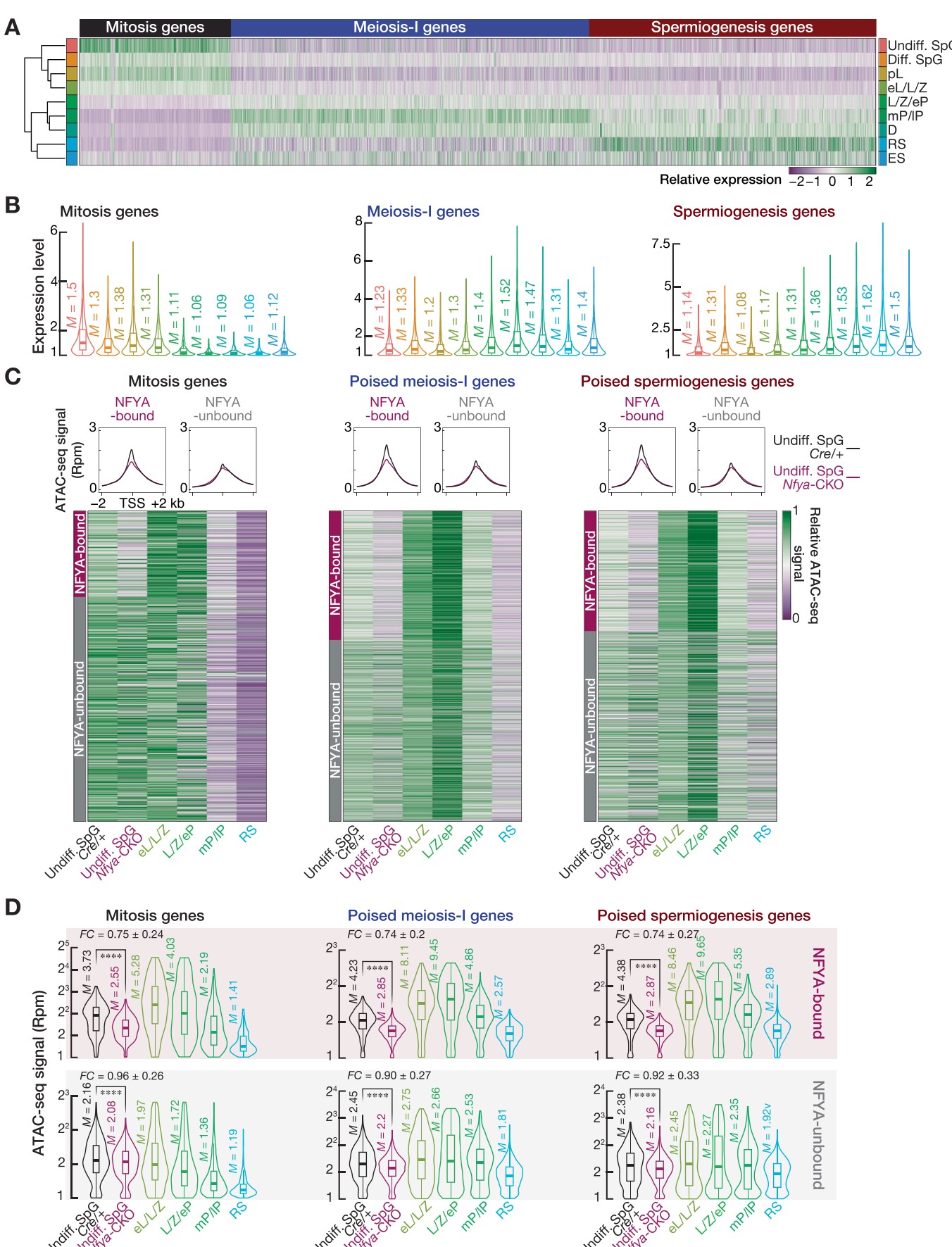

**Figure 5. *Nfya* deletion impairs chromatin accessibility at meiotic gene promoters in spermatogonia.**

(A) Unsupervised hierarchical clustering of nine germ cell populations captured by sc-RNA using the transcript abundance of mitosis, meiosis-I, and spermiogenesis genes. Undiff. SpG undifferentiated SpG, Diff. SpG differentiating SpG, pL preleptotene, eL/L/Z early leptotene, leptotene, and zygotene, L/Z/eP leptotene, zygotene, early pachytene, mP/lP middle and late pachytene, D diplotene, RS round spermatid, ES elongating spermatid. (B) RNA expression levels of mitosis ($n = 1152$), meiosis-I ($n = 2733$), and spermiogenesis ($n = 2182$) genes across nine germ cell populations defined by scRNA-seq. Vertical lines represent the median. Whiskers show maximum and minimum values. IQR is represented by boxplots. (C) Metaplot (top) shows ATAC-seq signal in the $-2$ kb to $+2$ kb window flanking transcription start sites (TSSs) of NFYA-bound and -unbound mitosis, poised meiosis-I and poised spermiogenesis genes. Heatmap (bottom) represents relative ATAC-seq signal at the promoters of mitosis, poised meiosis-I, and poised spermiogenesis genes that are either NFYA-bound or -unbound. (D) Reads per million (Rpm)-normalized ATAC-seq signals at the promoters of mitosis, poised meiosis-I, and poised spermiogenesis genes, which are either NFYA-bound (Mitosis, $n = 326$; poised meiosis-I, $n = 1001$; poised spermiogenesis, $n = 574$) or -unbound (Mitosis, $n = 854$; poised meiosis-I, $n = 1414$; poised spermiogenesis, $n = 913$). Vertical lines represent the median. Whiskers show maximum and minimum values. IQR is represented by boxplots. Two-sided Wilcoxon matched-pairs signed-ranked sum test. ****$P < 0.0001$. Mitosis NFYA-bound genes in Undiff. SpG from *Cre/+* vs *Nfya-CKO* $P = 2.817 \times 10^{-46}$, NFYA-unbound $P = 2.145 \times 10^{-22}$; Poised meiosis-I NFYA-bound genes in Undiff. SpG from *Cre/+* vs *Nfya-CKO* $P = 8.658 \times 10^{-153}$, NFYA-unbound $P = 6.273 \times 10^{-80}$; Poised spermiogenesis NFYA-bound genes in Undiff. SpG from *Cre/+* vs *Nfya-CKO* $P = 1.008 \times 10^{-85}$, NFYA-unbound $3.725 \times 10^{-27}$.

with mitosis genes in SpG from *Cre/+* (Fig. EV4B). This finding corroborates with the temporal expression of these gene categories: Mitosis genes are actively transcribed in SpG, whereas poised meiosis-I and spermiogenesis genes accumulate paused Pol II that is later released into transcriptional elongation in the pachytene stage of meiosis-I (Appendix Fig. S1A; Figs. EV1B,C). We next quantified the normalized Pol II Ser5P coverage around the TSSs of NFYA-bound and -unbound poised genes in SpG of *Nfya-CKO* and *Cre/+* (Fig. EV4C,D). *E. coli* spike-in sequences in each sample enabled us to calculate normalized coverage for each sample. We found reduced Pol II Ser5P occupancy around the TSSs of NFYA-bound poised genes in SpG from *Nfya-CKO* compared to that from *Cre/+* mice (Fig. EV4C,D). We also observed a reduction in Pol II Ser5P occupancy around the TSSs of NFYA-unbound poised genes between *Nfya-CKO* and *Cre/+* mice. Yet, such reduction appeared to be less than the reduction we observed for NFYA-bound genes (Fig. EV4C,D). We thus analyzed the changes in Pol II Ser5P occupancy within $\pm 0.5$ kb of the TSSs of NFYA-bound and-unbound poised genes, and randomly chosen 5000 control genes in SpG of *Nfya-CKO* compared to that of *Cre/+* mice (Fig. EV4E). Intriguingly, we found a larger reduction in Pol II Ser5P occupancy around the TSSs of NFYA-bound poised genes in NFYA-deficient SpG compared to that of NFYA-unbound poised genes (NFYA-bound poised genes, *Nfya-CKO* ÷ *Cre/+* = 0.63; NFYA-unbound genes, *Nfya-CKO* ÷ *Cre/+* = 0.77) (Fig. EV4E; Two-sided Wilcoxon matched-pairs signed-rank sum test; ****$P < 0.0001$). However, we note that the reduction in Pol II Ser5P occupancy for NFYA-unbound poised genes was comparable to that for randomly chosen control genes (*Nfya-CKO* ÷ *Cre/+* = 0.76). When taken together, we conclude that prior to meiotic entry, NFYA regulates chromatin accessibility—thereby contributing to the accumulation of paused Pol II—at the promoters of genes that remain poised for activation later during meiosis.

### NFYA regulates accessible chromatin at the promoters of genes activated by the STRA8/MEIOSIN axis

TFs STRA8 and MEIOSIN induce the transition from mitosis to meiosis (Anderson et al, 2008; Kojima et al, 2019; Ishiguro et al, 2020). Similar to the defective phenotypes reported in *Stra8* (Anderson et al, 2008) and *Meiosin* (Ishiguro et al, 2020) mutant mice, spermatogenesis arrests at meiotic entry in *Nfya-CKO* mice. Interestingly, we observed reduced chromatin accessibility at *Stra8* and *Meiosin* promoters that are in fact bound by NFYA (Fig. EV5A;

Dataset EV3; *Stra8*, *Nfya-CKO* ÷ *Cre/+* = 0.90; *Meiosin*, *Nfya-CKO* ÷ *Cre/+* = 0.70). We next sought to examine the change in chromatin accessibility at the promoters of 1,057 genes whose transcription is directly regulated by both STRA8 and MEIOSIN or STRA8 alone, or MEIOSIN alone (Kojima et al, 2019; Ishiguro et al, 2020). The RNAs of STRA8/MEIOSIN target genes first appeared in diff. SpG and peaked in early spermatocytes of prophase I, L/Z/eP, corroborating their role in meiotic initiation and progression (Kojima et al, 2019; Ishiguro et al, 2020) (Fig. EV5B). Intriguingly, of the 1,057 genes, the promoters of 471 were bound by NFYA in SpG (Fig. EV5C, left panel; median distance from TSS to the nearest NFYA peak = 257 bp). We found that chromatin accessibility at the promoters of NFYA-bound STRA8/MEIOSIN target genes was reduced by ~37% in undiff. SpG from *Nfya-CKO* mice compared to that from *Cre/+* mice (Fig. EV5C; right panel; two-sided Wilcoxon matched-pairs signed-ranked sum test; ****$P < 0.0001$). Among the 471 NFYA-bound STRA8/MEIOSIN target genes whose promoter accessibility was reduced in NFYA-deficient SpG, many function in meiotic recombination (e.g., *Sycp2*, *Syce3*, *Prdm9*, *Rad51*, *Msh5*, *Stag3* (Handel and Schimenti, 2010; Baudat et al, 2013)), cell cycle (e.g., *Meioc* (Soh et al, 2017)), and meiotic transcriptional program (e.g., *Taf4b* (Falender et al, 2005)) (Fig. EV5D). Together, our data suggest that NFYA regulates accessible chromatin at the promoters of *Stra8* and *Meiosin* genes as well as genes that are activated by STRA8 and MEIOSIN.

### NFYA participates in the regulation of the mitotic transcriptional program in spermatogonia

NFYA regulates chromatin accessibility at the promoters of poised meiosis-I and spermiogenesis genes in SpG, prior to the onset of their transcription during prophase I of meiosis (Figs. 1, 2, and 5). However, unlike poised genes, transcription of mitotic genes is switched off after entry into meiosis-I (Fig. EV1B,C; Appendix Fig. S1A). We found that NFYA binds ~one-third of the mitotic gene promoters in SpG (Fig. 2F). Deletion of *Nfya* results in compromised differentiation of SpG (Fig. 3G). Moreover, scRNA-seq analysis revealed that NFYA-deficient SpG cluster separately from control SpG in UMAP space (Fig. 4A). These suggest a regulatory function for NFYA in mitotic transcriptional program beyond its role in regulating accessible chromatin at poised meiotic promoters. We found that accessible chromatin at mitotic promoters bound by NFYA was decreased in NFYA-deficient SpG (*Nfya-CKO* ÷ *Cre/+* = 0.75), whereas chromatin accessibility at

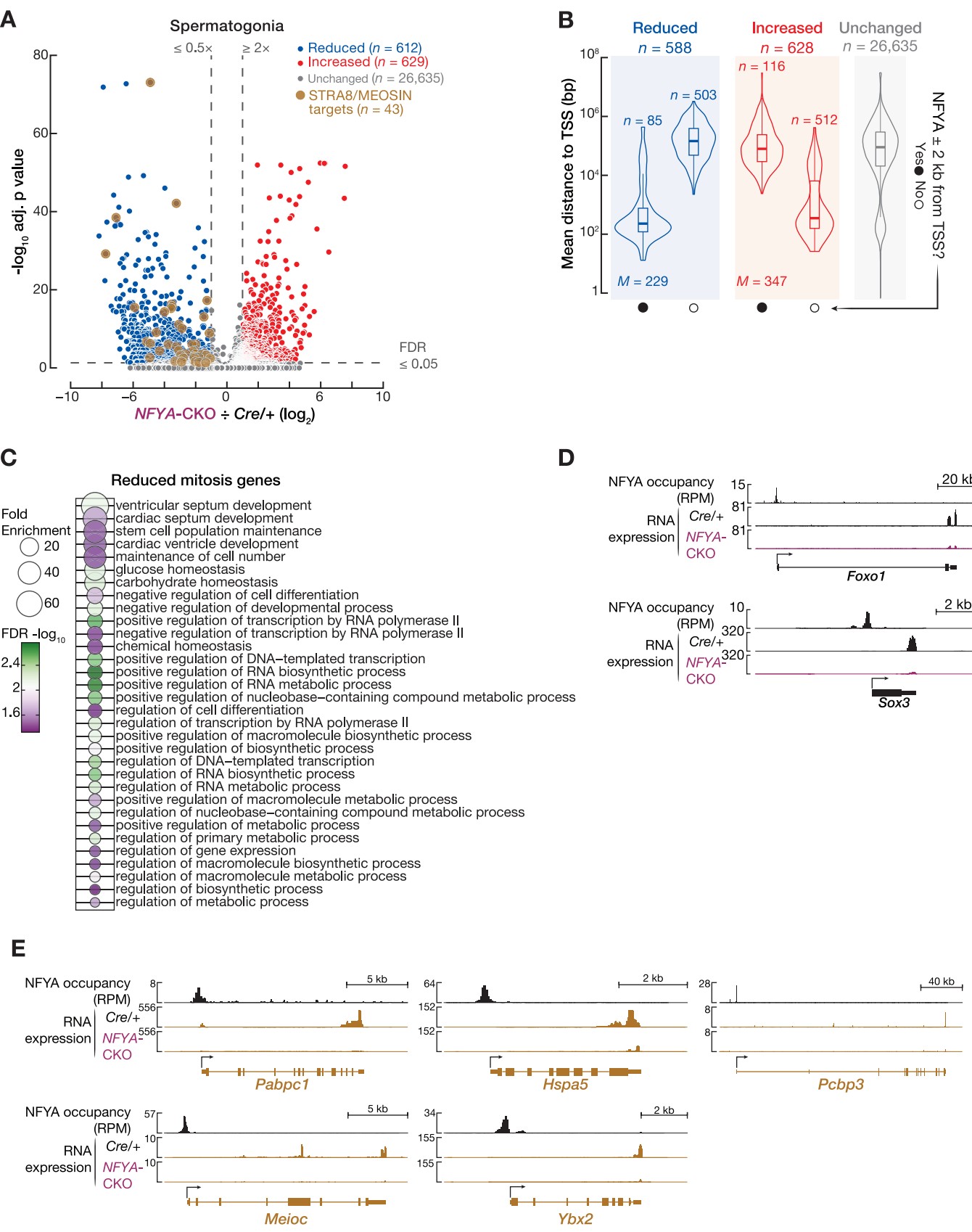

**A** Spermatogonia

**C** Reduced mitosis genes

**D**

**E**

**Figure 6.   NFYA regulates the mitotic transcriptional program in spermatogonia.**

(A) The volcano plot shows the genes whose RNA abundance increased or decreased significantly between the spermatogonia from 8-week-old *Cre/+* and *Nfya-CKO* mice using scRNA-seq. Blue color represents genes whose RNA abundance is reduced (FC ≤ 0.5, FDR ≤ 0.05), while red color denotes genes whose expression is increased in NFYA-deficient spermatogonia (FC ≥ 2, FDR ≤ 0.05). Light brown color represents direct STRA8/MEIOSIN target genes whose RNA abundance is reduced in spermatogonia from *Nfya-CKO* mice compared to that from *Cre/+* mice. Benjamini–Hochberg correction was used to compute the false discovery rate. (B) Distance (average of three biological replicates) from the TSS to the nearest NFYA peak for deregulated and unchanged genes in spermatogonia. Vertical lines represent the median. Whiskers show maximum and minimum values. IQR is represented by boxplots. (C) Gene Ontology (GO) analysis for reduced mitosis genes. The gene categories are sorted by fold enrichment. (D) IGV views for NFYA CUT&RUN and scRNA-seq signals at gene boundaries of deregulated exemplified genes targeted by NFYA. (E) IGV views for NFYA CUT&RUN and scRNA-seq signals at gene boundaries of exemplified STRA8/MEIOSIN target genes that are bound by NFYA.

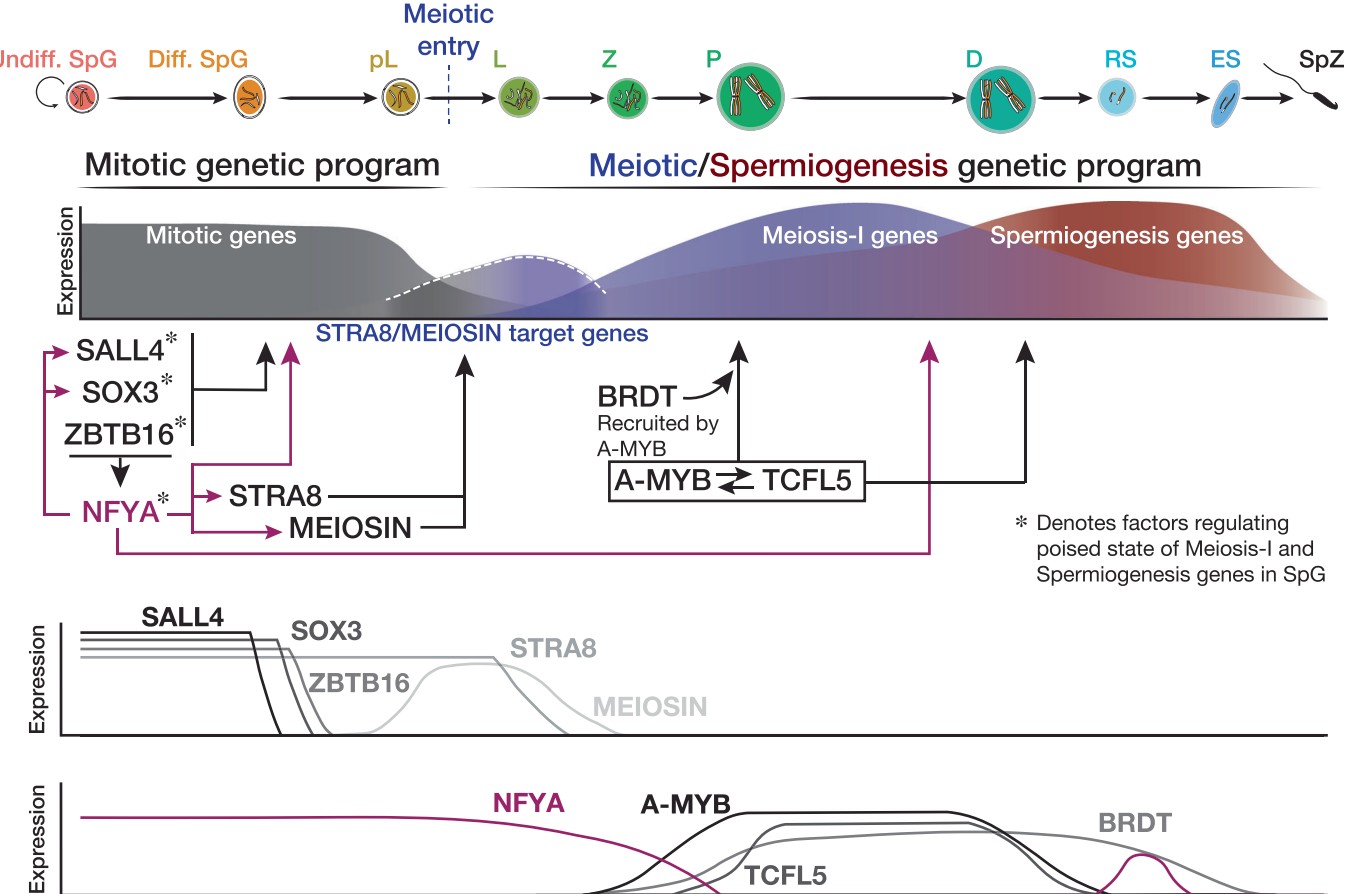

**Figure 7.   Model depicting transcription factors network regulating the timely expression of genes during spermatogenesis.**

The model incorporates findings from refs. (Anderson et al, 2008; Bolcun-Filas et al, 2011; Li et al, 2013; Kojima et al, 2019; Ishiguro et al, 2020; Alexander et al, 2023; Cecchini et al, 2023; Yu et al, 2023; Kaye et al, 2024; Yi et al, 2025) and this study. Germ cell development is aligned with the temporal expression of key transcription factors. The model highlights the sequential roles of key regulatory proteins, i.e, SOX3, SALL4, ZBTB16, NFYA, STRA8, MEIOSIN, A-MYB, TCFL5, and BRDT during spermatogenesis.

the promoters of NFYA-unbound mitotic genes remained nearly unchanged (*Nfya-CKO* ÷ *Cre/+* = 0.96) (Fig. 5D; two-sided Wilcoxon matched-pairs signed-ranked sum test; ****$P < 0.0001$). Using scRNA-seq, we next measured the change in the steady-state RNA abundance between SpG from *Nfya-CKO* and *Cre/+* (Fig. 6A). The RNA abundance of 1,241 genes—that account for <5% of all genes—were significantly deregulated in SpG from *Nfya-CKO* compared with that from *Cre/+* (Fig. 6A; *Nfya-CKO* ÷ *Cre/+* ≤ 0.5 or ≥2; FDR < 0.05). Deregulation of a small fraction of all genes likely explains the mild perturbation we

observed in the differentiation of SpG in *Nfya-CKO* males (Fig. 3G). Of the 1,241 deregulated genes, steady-state RNA abundance of 629 was higher in NFYA-deficient SpG, while the RNA products of 612 genes were reduced (Fig. 6A). Our CUT&RUN analysis of NFYA binding from wild-type SpG revealed that NFYA bound near the TSSs for 201 of 1241 genes whose transcripts were deregulated in NFYA-deficient SpG (Fig. 6B). Of the 201 NFYA-bound deregulated genes, RNA abundance of 85 were reduced in NFYA-deficient SpG (Fig. 6B). Interestingly, among these 85 genes, 11 belonged to mitotic gene category that were enriched for GO related to the

regulation of transcription suggesting that many other NFYA-unbound deregulated genes are likely regulated by transcription factors that are themselves under NFYA control (Fig. 6C,D; Dataset EV5). Indeed, these 11 genes included *Foxo1* and *Sox3*, which encode transcription factors that participate in the regulation of the mitotic genetic program (McAninch et al, 2020; Shen et al, 2022; Yi et al, 2025).

Finally, among the 612 genes whose RNA abundance was reduced in NFYA-deficient SpG, 43 were direct STRA8/MEIOSIN targets that are required for meiotic entry and progression (Kojima et al, 2019; Ishiguro et al, 2020) (Fig. 6A). Our NFYA CUT&RUN analysis from wild-type SpG revealed that NFYA bound near the TSSs for 22 of 43 STRA8/MEIOSIN target genes, whose transcripts were reduced in NFYA-deficient SpG, including *Meioc*, *Hspa5*, *Ybx*, *Pabpc1*, and *Pcbp3* (Fig. 6E; Dataset EV5). MEIOC regulates the transition from mitosis to meiosis-I (Abby et al, 2016; Soh et al, 2017). HSPA5 is required for the maintenance of differentiating SpG (Wen et al, 2023). RNA-binding proteins, YBX2, PABPC1, and PCBP3, regulate the translation of mRNAs required later during spermatogenesis (Kimura et al, 2009; Chapman et al, 2013; He et al, 2019). Together, these findings are consistent with our observation that spermatogenesis arrests at meiotic entry in *Nfya-CKO* males.

## Discussion

Entry into the first and longest stage of meiosis, prophase I, accompanies extensive cellular and chromosomal events (Handel and Schimenti, 2010; Säflund and Özata, 2023; Ishiguro, 2024). Such dramatic changes upon meiotic entry entail synchronous transcriptional activation of genes during the pachytene stage of prophase I (Li et al, 2013; Alexander et al, 2023; Cecchini et al, 2023; Yu et al, 2023). Importantly, in pre-meiotic SpG, the promoters of many of these genes accumulate paused Pol II that is later released into elongation in the pachytene stage of prophase I by the activity of TFs A-MYB and BRDT (Alexander et al, 2023; Kaye et al, 2024). Prior to our findings reported here, how chromatin accessibility is regulated to allow Pol II to bind the promoters of meiotic genes in pre-meiotic cells was unknown. Here, we provide functional and molecular evidence that NFYA regulates chromatin accessibility around the promoters of genes expressed during prophase I, thereby allowing the binding of Pol II, in pre-meiotic SpG. Our data also reveal that during meiotic entry, NFYA regulates permissive chromatin at the promoters of genes that are activated by the two key meiotic initiators STRA8 and MEIOSIN (Anderson et al, 2008; Kojima et al, 2019; Ishiguro et al, 2020).

During step-wise developmental processes, pioneer factors can elicit chromatin opening at gene regulatory sequences to prime following developmental events as opposed to immediate induction of gene expression (Bevington et al, 2016; Zaret, 2020; Yi et al, 2025). Consistently, we reveal that in the mitotic stage, NFYA regulates the chromatin accessibility at the promoters of genes that are indeed transcribed during meiosis-I. Our findings thus raise the possibility that NFYA may function as a pioneer factor in pre-meiotic SpG. Supporting this view, an accumulated line of biochemical, genetic, and molecular evidence has proposed NFYA as a "pioneer factor" in other cellular contexts (Coustry et al, 2001; Fleming et al, 2013; Nardini et al, 2013; Oldfield et al, 2014;

Sherwood et al, 2014; Oldfield et al, 2019). NFYA can bind to its DNA motif in polycomb-repressed chromatin (Fleming et al, 2013). In mouse ESCs, NFYA promotes chromatin accessibility for the binding of master TFs (Oldfield et al, 2014). In fact, the model proposes that NFYA binding induces ~80° bend in DNA, ultimately resulting in nucleosome repositioning (Oldfield et al, 2014). NFYA forms a complex with NFYB and NFYC (Dolfini et al, 2009). The crystal structure of NFY complex bound to its DNA motif demonstrated that NFY complex induces an α helix into the DNA minor groove, thereby establishing permissive chromatin modifications at NFY-bound promoters (Nardini et al, 2013). Moreover, computational modeling that analyzes genome-wide DNase I hypersensitivity profiles classified NFYA among the strongest pioneer activity indices (Sherwood et al, 2014).

An in vitro study demonstrated that NFYA displaces nucleosomes (Coustry et al, 2001). One of the hallmarks of pioneer factors is that they recruit nucleosome-remodeling complexes to promote nucleosome reorganization (Barral and Zaret, 2024). We therefore cannot rule out the possibility that NFYA recruits a chromatin remodeler to regulate permissive chromatin during spermatogenesis. This will be an important subject of investigation in our future studies.

A recent study demonstrated that TFs, ZBTB16, SALL4, and SOX3—which are expressed in undifferentiated SpG (Hobbs et al, 2012; McAninch et al, 2020)—, cooperatively establish accessible chromatin at the promoters of meiotic genes in juvenile SpG (Yi et al, 2025). The promoters bound by ZBTB16, SALL4, and SOX3 retain open chromatin features, such as H3K4me3, H3K27Ac, and hypomethylation (Yi et al, 2025). Given that NFYA protein abundance is lower in undifferentiated type A SpG than in differentiating SpG (Fig. 2C), cooperative activity of ZBTB16, SALL4, and SOX3 in undifferentiated SpG may prime the binding of NFYA at meiotic gene promoters in differentiating SpG. Congruently, we observed that ~one-third of the promoters bound by ZBTB16, SALL4, and SOX3 were occupied by NFYA in differentiating SpG (Dataset EV6; 5519 of the 16,151 promoters) (Yi et al, 2025). Together with our recent studies, it may be possible that sequential and collective activity of factors conducts the establishment and maintenance of accessible chromatin at meiotic promoters, thereby promoting orderly entry and progression of meiosis-I. Therefore, investigating the interplay between NFYA and other key factors regulating open chromatin before entry into meiosis-I will be of great interest to study gene regulatory events of spermatogenesis.

STRA8 and its partner MEIOSIN initiate meiotic entry by regulating both genes required for chromosomal events of prophase I as well as genes required for meiotic G1-S cell cycle regulation (Kojima et al, 2019; Ishiguro et al, 2020). Spermatogenic arrest occurred at meiotic initiation rather than meiotic progression in *Stra8* and *Meiosin* mutant mice (Anderson et al, 2008; Ishiguro et al, 2020). However, ectopic expression of STRA8 or MEIOSIN did not suffice for the induction of meiosis in vitro (Endo et al, 2015; Ishiguro et al, 2020), suggesting that meiotic entry requires additional molecular events acting upstream of STRA8/MEIOSIN. We would thus argue that the permissive chromatin state at the promoters of genes required for meiotic entry is established in pre-meiotic SpG that precedes the activity of STRA8/MEIOSIN. Supporting this idea, our data revealed that NFYA binds the promoters of STRA8/MEIOSIN target genes and that the

chromatin accessibility at such promoters was in NFYA-deficient SpG (Fig. EV5). Indeed, we observed a lesser reduction in chromatin accessibility at the promoter of *Stra8* than that of *Meiosin*, suggesting the existence of another factor compensating for NFYA activity. Because STRA8 and NFYA were expressed earlier than MEIOSIN (Dataset EV4B), it is possible that the expression of this compensatory factor precedes NFYA expression. Although the initiation and the progression of meiosis is sexually dimorphic, the molecular mechanisms, such as the role of STRA8 and MEISOIN, are conserved between males and females. As NFYA is broadly expressed in many tissues including the ovary (Appendix Fig. S5A), it would be interesting to examine whether the regulatory role of NFYA is conserved in oocytes.

While our study provides convincing evidence that NFYA promotes chromatin accessibility at the promoters of poised genes prior to meiosis, it is important to note that there are several additional TF motifs identified under ATAC-seq peaks at poised promoters (Appendix Fig. S4). This suggests that multiple regulators may contribute to the regulation of open chromatin state during spermatogenesis. This will therefore be an important subject of investigation in our future studies aiming to characterize their role in chromatin regulation, as this will provide a more comprehensive understanding of the regulatory networks governing the meiotic genetic program.

In summary, we have elucidated the role of NFYA in spermatogenesis. Specifically, we have shown that prior to meiotic entry, NFYA regulates permissive chromatin and thus facilitates paused Pol II occupancy around the promoters of genes required for meiotic entry, meiotic progression, and spermiogenesis, thereby priming them for timely expression during spermatogenesis. Together with other findings (Anderson et al, 2008; Bolcun-Filas et al, 2011; Li et al, 2013; Kojima et al, 2019; Ishiguro et al, 2020; Alexander et al, 2023; Cecchini et al, 2023; Yu et al, 2023; Kaye et al, 2024; Yi et al, 2025), our data suggests a model for a current TFs network of spermatogenesis, in which NFYA plays a crucial role by regulating the chromatin accessibility required during both meiotic entry and progression of prophase I (Fig. 7). Our discovery will benefit both our understanding of the step-wise development of male germ cells and the efforts for generating in vitro-derived germ cells for therapeutic purposes.

## Methods

### Reagents and tools table

| Reagent/resource | Reference or source | Identifier or catalog number |
|---|---|---|
| **Antibodies** | | |
| Guinea pig polyclonal anti-rabbit | Antibodies online | Cat# ABIN101961 |
| Mouse monoclonal anti-ACTIN | Sigma | Cat# A1978 |
| Mouse monoclonal anti-DMRT1 | Santa Cruz Biotechnology | Cat# sc-377167 |
| Mouse monoclonal anti-H3S10P | Abcam | Cat# ab14955 |
| Mouse monoclonal anti-NFYA | Santa Cruz Biotechnology | Cat# sc-17753 |
| Mouse monoclonal anti-SALL4 | Abcam | Cat# ab57577 |

| Reagent/resource | Reference or source | Identifier or catalog number |
|---|---|---|
| Mouse monoclonal anti-SCP-3 | Santa Cruz Biotechnology | Cat# sc-74569 |
| Mouse monoclonal anti-STRA8 | ProteinTech | Cat# CL488-68071 |
| Rabbit polyclonal anti-cKIT | Abcam | Cat# ab317843 |
| Rabbit polyclonal anti-H2A.X ser139 | Novus Biologicals | Cat# NB100-384 |
| Rabbit polyclonal anti-DDX4 | Abcam | Cat# ab13840 |
| Rabbit polyclonal anti-Phospho-Rpb1 CTD (Ser5) | Cell signaling | Cat# 13523 |
| Rabbit polyclonal anti-HSPA2 | Sigma | Cat# HPA000798 |
| Rabbit polyclonal anti-NFYA | Bethyl laboratories | Cat# A302-105A |
| Rabbit polyclonal anti-NFYA | Sigma | Cat# HPA050779 |
| Rabbit polyclonal IgG | EpiCypher | Cat# 13-004 |
| Rabbit polyclonal anti-SOX9 | Sigma | Cat# ABE571 |
| Rat monoclonal anti-GATA-4 | eBioscience™ | Cat# 14998080 |
| Rat monoclonal anti-GCNA | Abcam | Cat# ab82527 |
| Alexa Fluor 488 anti-rabbit | Thermo Fisher Scientific | Cat# A21206 |
| Alexa Fluor 594 anti-rabbit | Thermo Fisher Scientific | Cat# A11012 |
| Alexa Fluor 488 anti-mouse | Thermo Fisher Scientific | Cat# A21202 |
| Alexa Fluor 594 anti-mouse | Thermo Fisher Scientific | Cat# A21203 |
| Alexa Fluor 647 anti-mouse | Thermo Fisher Scientific | Cat# A21235 |
| Alexa Fluor 594 anti-rat | Thermo Fisher Scientific | Cat# A21209 |
| IRDye 680RD anti-mouse | LI-COR Biosciences | Cat# 926-68072 |
| IRDye 800CW anti-rabbit | LI-COR Biosciences | Cat# 926-32213 |
| **Chemicals, enzymes, and other reagents** | | |
| 10 × PBS | Gibco | Cat# 70011044 |
| 4 × Loading dye | Invitrogen | Cat# NP0007 |
| Agencourt® AMPure® XP magnetic beads | Beckman Coulter | Cat# A63880 |
| Blocking buffer | Thermo Fisher Scientific | Cat# 37565 |
| Bouin´s solution | VWR | Cat# 7000-1 |
| BSA | Sigma | Cat# A9418 |
| Collagenase type IV | Worthington | Cat# LS004188 |
| Concanavalin A-coated magnetic beads | Cell signaling | Cat# 82307S |
| DABCO | Sigma | Cat# D27802 |
| DAPI | Sigma | Cat# D1306 |
| Dnase I | Roche | Cat# 10104159001 |
| FSC 22 | Leica | Cat# 3801480 |
| GBSS | Sigma | Cat# G9779 |

| Reagent/resource | Reference or source | Identifier or catalog number |
|---|---|---|
| Hoechst 33342 | Thermo Fisher Scientific | Cat# 622495 |
| N,N-Dimethylformamide | Sigma | Cat# D4551 |
| Nitrocellulose membrane | Thermo Fisher Scientific | Cat# 88018 |
| pAG-Tn5 preloaded | Epicypher | Cat# 15-1117 |
| Peanut lectin | Sigma | Cat# L7381 |
| Photo-flo 200 | Kodak | Cat# 1026269 |
| ProLong gold antifade mountant with DAPI | Thermo Fisher Scientific | Cat# P36931 |
| Propidium Iodide | Thermo Fisher Scientific | Cat# P3566 |
| Protease inhibitor | Thermo Fisher Scientific | Cat# 1861279 |
| PCR 2X Master mix | New England Biolabs | Cat# ME541L |
| RIPA | Thermo Fisher Scientific | Cat# 89900 |
| RNase inhibitor | BD Biosiences | Cat# 51-9024039 |
| Thermolabile Proteinase K | New England Biolabs | Cat# P8111S |
| Tris glycine gel | Thermo Fisher Scientific | Cat# XP042000 |
| Triton X-100 | Sigma | Cat# T8787 |
| Trypsin | Gibco | Cat# 27250-018 |
| Tween-20 | Sigma | Cat# P9516 |
| Agilent High Sensitivity DNA Kit | Agilent | Cat# 5067-4626 |
| ATAC-Seq Kit | Active Motif | Cat# 53150 |
| BD Rhapsody™ 8-Lane Cartridge | BD Biosciences | Cat# 666262 |
| BD Rhapsody™ cDNA Kit | BD Biosciences | Cat# 633773 |
| BD Rhapsody™ Enhanced Cartridge Reagent Kit V3a | BD Biosciences | Cat# 667052 |
| BD Rhapsody™ Multiomic ATAC-Seq Amplification Kit | BD Biosciences | Cat# 41928 |
| BD Rhapsody™ Tagmentation and Supplemental Reagents Kit | BD Biosciences | Cat# 41926 |
| BD Rhapsody™ WTA Amplification Kit | BD Biosciences | Cat# 633801 |
| ChIC/CUT&RUN Assay Kit | Active Motif | Cat# 53180 |
| DeadEnd™ Fluorometric TUNEL System | Promega | Cat# G3250 |
| NEBNext Ultra II DNA Library Prep Kit for Illumina | New England Biolabs | Cat# E7645S |
| Pierce™ BCA Protein Assay Kit | Thermo Fisher Scientific | Cat# 23227 |
| Hematoxylin and eosin staining Kit | Abcam | Cat# ab245880 |
| **Experimental models** | | |
| C57BL/6 J | The Jackson Laboratory | RRID: IMSR_JAX:000664 |

| Reagent/resource | Reference or source | Identifier or catalog number |
|---|---|---|
| NFYA conditional mutant mice; *Nfya^fl/fl*; *Stra8-Cre* | This study via Cyagen Biosciences | |
| Stra8-Cre mice; *Stra8-Cre^Cre/+* | Cyagen Biosciences | C001536 |
| **Oligonucleotides and other sequence-based reagents** | | |
| sgRNA targeting *Nfya* intron 4: 5'-CAT CAG CAT AAC GTT TAA CTT GG -3' | Integrated DNA Technologies | N/A |
| sgRNA targeting *Nfya* intron 6: 5'- AGG AAG AGG AGG TAT GTA ACA GG -3' | Integrated DNA Technologies | N/A |
| Forward genotyping primer for *loxp* residing in intron 4 (PCR-1): 5'-TTT CAG TTT CTG TGG ATC GGA AGG-3' | Integrated DNA Technologies | N/A |
| Reverse genotyping primer for *loxp* residing in intron 4 (PCR-1): 5'-ATA CTC TGA AGT GCT GTT ATC AAC G-3' | Integrated DNA Technologies | N/A |
| Forward genotyping primer for *loxp* residing in intron 6 (PCR-2): 5'-CTT TGG GCC TCT ACA TCT ATT CAC-3' | Integrated DNA Technologies | N/A |
| Reverse genotyping primer for *loxp* residing in intron 6 (PCR-2): 5'-CTG AAG CCA AGT CTA GTC CTG TT-3' | Integrated DNA Technologies | N/A |
| Forward genotyping primer for *Cre* knock-in (PCR-3): 5'-GCT GGA GTT TCA ATA CCG GAG ATC-3' | Integrated DNA Technologies | N/A |
| Forward genotyping primer for *Cre* knock-in (PCR-3): 5'-CCT TGT ATT CGA CAA GCC CCA AA-3' | Integrated DNA Technologies | N/A |
| Forward internal control primer (PCR-3): 5'-GCA GAA GAG GAC AGA TAC ATT CAT-3' | Integrated DNA Technologies | N/A |
| Reverse internal control primer (PCR-3): 5'-CCT ACT GAA GAA TCT ATC CCA CAG-3' | Integrated DNA Technologies | N/A |
| **Software** | | |
| Bowtie2 version v2.5.4 | Langmead and Salzberg, 2012 | http://bowtie-bio.sourceforge.net/index.shtml |
| MACS3 version v3.0.2 | Zhang et al, 2008 | https://macs3-project.github.io/MACS/docs/callpeak.html |
| STAR version v2.7.10b | Dobin et al, 2013 | https://github.com/alexdobin/STAR/releases |
| SAMtools version v1.20 | Danecek et al, 2021 | http://samtools.sourceforge.net |
| deepTools version v3.3.2 | Ramírez et al, 2016 | https://deeptools.readthedocs.io/en/latest/ |
| BEDTools version v2.31.0 | Quinlan and Hall, 2010 | https://github.com/arq5x/bedtools2 |

| Reagent/resource | Reference or source | Identifier or catalog number |
|---|---|---|
| Cutadapt version v4.8 | Martin, 2011 | https://cutadapt.readthedocs.io/en/v4.7/index.html |
| Fastp version v0.24.0 | Chen et al, 2018b | https://github.com/OpenGene/fastp |
| HOMER version v4.0 | Heinz et al, 2010 | http://homer.ucsd.edu/homer/ |
| Seurat version v5.2.1 | Hao et al, 2021 | https://github.com/satijalab/seurat/releases |
| DoubletFinder version v2.0.4 | McGinnis et al, 2019 | https://github.com/chris-mcginnis-ucsf/DoubletFinder |
| ArchR version v1.0.3 | Granja et al, 2021 | https://github.com/GreenleafLab/ArchR |
| Empiria Studio® Software | LI-COR Biosciences | https://bio.licor.com/bio/help/empiria_studio/ |
| ImageJ2 | Imagej.net | http://Imagej.net |
| R Studio version v4.3.2 | CRAN | https://www.r-project.org |
| GraphPad Prism version v10 | GraphPad Software | https://www.graphpad.com |

## Methods and protocols

### Mice

Mice were maintained and used according to the guidelines of the Regional Animal Experimentation Ethics Committee of the Swedish Board of Agriculture (8246-2021). C57BL/6 J mice (RRID: IMSR_JAX:000664) were used as wild-type mice.

Male germ cell-specific *Nfya* mutant mice were generated by conditional deletion of exons 5–6 of *Nfya* using Stra8-Cre-loxP system (Cyagen Biosciences, Jiangsu, China). Stra8-P2A-ZsGreen1-T2A-Cre mice were purchased from Cyagen (Cyagen Biosciences; C001536). 5′ *loxP* was inserted in intron 4 and 3′ *loxP* was inserted in intron 6. Briefly, two guide RNAs targeting sequences in *Nfya* intron 4 (5′-CAT CAG CAT AAC GTT TAA CTT GG-3′) and intron 6 (5′-AGG AAG AGG AGG TAT GTA ACA GG-3′), the donor vector containing *loxP* sites, and Cas9 protein were co-injected into the cytoplasm of pronuclear stage embryos. The injected embryos were cultured in KSOM medium overnight and those which developed into the two-cell stage were transferred into the uterus of pseudo-pregnant ICR females. F0 founder mice were back-crossed with C57BL/6 J mice in order to identify animals with germline transmission in F1.

### Isolation of mouse germ cells by Fluorescence-Activated Cell Sorting

Male germ cell sorting was performed as previously described (Cecchini et al, 2023; Yu et al, 2023) using the BD FACSMelody Cell Sorter (BD Biosciences). After dissecting testis from one mouse, the tubules were squeezed into 6 ml 1× Gey's Balanced Salt Solution (GBSS, Sigma; G9779) containing 0.4 mg/ml collagenase

type IV (Worthington; LS004188), which was then incubated at 33 °C for 15 min at 600 rpm. Afterwards, the tubules were washed twice with 6 ml GBSS and incubated with 1× GBSS containing 0.5 mg/ml trypsin (Gibco; 27250-018) and 1 μg/ml DNase I (Roche; 10104159001) at 33 °C for 15 min at 600 rpm. The tubules were then gently homogenized at 4 °C on ice using a Pasteur pipette. 400 μl FBS added to inactivate trypsin, and the cells were passed through a pre-wetted 70 μm cell strainer (Corning; CLS431751). Cells were then pelleted in a swing bucket centrifuge at 500×g for 10 min at 4 °C. Thereafter, cells were resuspended in 1× GBSS containing 5% (v/v) FBS, 1 μg/ml DNase I, and 5 μg/ml Hoechst 33342 (Thermo Fisher Scientific; 62249) and incubated at 33 °C for 45 min at 150 rpm. The cell suspension was then placed on ice and propidium iodide (PI; 0.2 μg/ml, f.c.; Thermo Fisher Scientific; P3566) was added. Finally, cells were filtered through a pre-wetted 40 μm cell strainer (Corning; CLS431750). Four-way cell sorting was performed to obtain the populations of spermatogonia (SpG), leptotene and zygotene (L/Z), pachytene and diplotene (P/D), and round spermatids (RS) on BD FACSMelody Cell Sorter (BD Biosciences) using the gates as described in Appendix Fig. S2A. In all, 488-nm laser was used to excite PI, while 405-nm laser was used to excite Hoechst 33342. The emission for PI was recorded using 700/54 nm bandpass filter, while the emission of Hoechst 33342 was recorded using 660/10-nm (red) and 448/45-nm (blue) bandpass filters.

The purity of each sorted cell population was further assessed by chromosome spread followed by immunofluorescence staining on aliquots of cells. The purity of SpG population was assessed by DNA, GCNA, and DMRT1 staining, while the purity of L/Z, P/D, and RS was assessed by the pattern of DNA, γH2AX, and SYCP3 staining (Appendix Fig. S2B). The purity of each population was calculated by assessing the staining from 42–188 cells in six independent replicates (Appendix Fig. S2B):

SpG population, ~95% spermatogonia;

L/Z population, ~5% spermatogonia, ~50% leptotene spermatocytes, ~29% zygotene spermatocytes, ~10% early pachytene spermatocytes, <1% pachytene & diplotene spermatocytes and round spermatids, ~5% spermatozoa;

P/D population, <1% spermatogonia, ~4% leptotene, zygotene, and early pachytene spermatocytes, ~48% pachytene spermatocytes, ~41% diplotene spermatocytes, ~6% round spermatids, <1% spermatozoa;

R/S population, ~88% round spermatids, ~12% spermatozoa.

### Chromosome spread

Cells were first incubated with 25 mM sucrose solution at room temperature for 20 min and then were fixed in 0.33% (w/v) paraformaldehyde solution containing 0.05% (v/v) Triton X-100 at room temperature. We then added the cells to SuperFrost Plus microscope slides (VWR; 631-108) in a dropwise manner, in which the slides were pre-dipped in 1% (w/v) paraformaldehyde solution containing 0.15% (v/v) Triton X-100. Slides were then incubated for at least 2 h in a humidifying chamber. Afterwards, the lid of the chamber was removed to dry the slides completely. Dried slides were washed at room temperature for 10 min in a step-wise manner with (i) 1× PBS containing 0.4% (v/v) Photo-Flo 200 (Kodak; 1026269), (ii) 1× PBS containing 0.1% (v/v) Triton X-100, (iii) ADB/PBS solution [10% ADB (0.03% (w/v) BSA, 0.01% (v/v)

Triton X-100, 10% (v/v) Donkey serum (Sigma, D9663), 2× PBS) and 90% 1× PBS). Slides were then incubated at 4 °C overnight with primary antibodies diluted in ADB solution (anti-γH2AX, Novus Biologicals, NB100-384, 1:1000 dilution; anti-SCP-3 (D-1), Santa Cruz, sc-74569, 1:400 dilution; anti-GCNA, Abcam, ab82527 1:1000 dilution; anti-DMRT1, Santa Cruz, sc-377167, 1:200 dilution). The next day, slides were washed sequentially as (i–iii, above), and incubated at 37 °C for 1 h with secondary antibodies diluted 1:2,000 in ADB (Alexa Fluor anti-mouse 594, Thermo Fischer Scientific, A21203; Alexa Fluor anti-rabbit 488, Thermo Fisher Scientific, A21206; Alexa Fluor anti-rat 594, Thermo Fisher Scientific, A21209). Finally, slides were washed three times in 1× PBS containing 0.4% (v/v) Photo-Flo 200 at room temperature each for 10 min and once in 0.4% (v/v) Photo-Flo 200 at room temperature for 10 min. Slides were dried in the dark and mounted with ProLong Gold Antifade Mountant with DAPI (Thermo Fisher Scientific, P36931). Images of chromosome spread were captured by a Zeiss Axio Observer 7 inverted widefield microscope.

## Testis histology by hematoxylin and eosin staining

Testis tissues from wild-type, *Cre/+*, and *Nfya-CKO* mice were fixed with Bouin's solution (VWR; 7000.1000) at room temperature overnight. The next day, fixed tissues were washed with 70% (v/v) ethanol and embedded in paraffin to section them at 5 μm thickness. We performed hematoxylin and eosin (H&E) staining using H&E Staining Kit according to the manufacturer's protocol (Abcam; ab245880). Images for H&E staining were captured using a Zeiss Axio Observer 7 inverted widefield microscope.

## Immunohistochemistry staining

Immunohistochemistry (IHC) was performed according to the standard protocols. In brief, we de-waxed 5-μm sections in xylene and dehydrated them in descending percentages of ethanol [100%, 95%, and 70% (v/v) ethanol; sections were incubated 5 min in each ethanol solution]. To retrieve antigens, sections were boiled in 1 mM citrate buffer (pH 6.0). The ImmPRESS Excel Amplified HRP Polymer Staining kit (Vector Laboratories; MP-7401) was used for IHC staining. To inactivate the endogenous peroxidase, tissue sections were incubated with 3% (v/v) hydrogen peroxide at room temperature for 10 min, then were blocked with 2.5% (v/v) horse serum. Sections were incubated with rabbit anti-NFYA (Sigma; HPA050779; 1:500 dilution) antibody at 4 °C overnight. The next day, sections were covered with secondary HRP anti-rabbit antibody (Vector Laboratories; MP-7401) and incubated at room temperature for 1 h. Following the secondary antibody incubation, chromogenic substrate (Thermo Fisher Scientific; TA-125-QHDX) was added and incubated at room temperature for 10 min. Thereafter, slides were counterstained with hematoxylin and dehydrated with ascending percentages of ethanol [70%, 95%, and 100% (v/v) ethanol]. Slides were finally covered with coverslips. IHC images were captured by a Zeiss Axio Observer 7 inverted widefield microscope.

## Immunofluorescence staining

### Staining on Bouin's fixed sections
In all, 5-μm tissue sections were deparaffinized in three sequential incubations in xylene for 7 min each and then washed in descending percentages of ethanol for 7 min each [100%, 95%, and 70% (v/v) ethanol]. Antigen retrieval was performed by boiling the slides in 1 mM citrate buffer (pH 6.0) at 95–100 °C for 20 min. We next cooled down the slides at room temperature for 10 min and washed them with 1× PBS. Tissue sections on slides were outlined using PAP pen (Thermo Fischer Scientific; 008899). Following three washes in 1× PBS (each wash was 5 min), sections were blocked with 5% (w/v) bovine serum albumin (Sigma; A9418) at room temperature for 1 h. Slides were then incubated at 4 °C overnight with primary antibodies diluted in 1× PBS (anti-DDX4/MVH, Abcam, ab13840, 1:500 dilution; anti-γH2AX, Novus Biologicals, NB100-384, 1:2000 dilution; FITC-conjugated peanut agglutinin, Sigma, L7381, 1:1000 dilution; anti-HSPA2, Sigma, A000798, 1:500 dilution; anti-SALL4, Abcam, ab57577, 1:1000 dilution; anti-c-KIT, Abcam, ab317843, 1:250 dilution; anti-NFYA (G-2), Santa Cruz, sc-17753 1:500 dilution; anti-STRA8, Proteintech, cL488-68071, 1:500 dilution; anti-H3S10P, Abcam, ab14955, 1:500 dilution; anti-GATA4, eBiosciences, 14998080, 1:100 dilution; anti-SOX9, Sigma, ABE571, 1:700 dilution). Following primary antibody incubation, slides were washed three times in 1× PBS containing 0.02% (v/v) Tween-20 (Sigma; P9516) for 10 min each. Secondary antibodies were then applied at room temperature for 1 h (anti-rabbit Alexa Fluor 488, Thermo Fisher Scientific, A21206, 1:700 dilution; anti-rabbit Alexa Fluor 594, Thermo Fisher Scientific, A211012, 1:700 dilution; anti-mouse Alexa Fluor 594, Thermo Fisher Scientific, A21203, 1:700 dilution; anti-mouse Alexa Fluor 488, Thermo Fisher Scientific, A21202, 1:700 dilution;anti-mouse Alexa Fluor 647, Thermo Fisher Scientific, A21235, 1:200 dilution; anti-rat Alexa Fluor 594, Thermo Fisher Scientific, A21209, 1:700 dilution). Nuclei were counterstained with DAPI (Sigma, D1306, 1:1000 dilution) at room temperature for 10 min. Finally, slides were washed twice with 1× PBS for 10 min each and mounted with coverslips using DABCO mounting media (Sigma, D27802). Immunofluorescence images were captured by a Zeiss Axio Observer 7 inverted widefield microscope.

### Staining on cryosections
Testis tissues from *Cre/+*, and *Nfya-CKO* mice were fixed in 1× PBS containing 4% (w/v) paraformaldehyde at 4 °C overnight. The next day, fixed tissues were washed five times in 1× PBS each for 30 min using a rocker platform. We then incubated the tissues in 30% (v/v) sucrose at 4 °C overnight. Thereafter, tissues were embedded in FSC 22 Frozen Section Media (Leica, 3801480) and were frozen at −80 °C. 5 μm cryosections were washed three times in 1× PBS for 5 min each and proceeded with antigen retrieval as described for Bouin's-fixed sections. Slides were incubated with anti-SCP-3 (D-1) antibody (Santa Cruz; sc-74569; 1:2000 dilution) and the secondary antibody anti-mouse Alexa Fluor 594 (Sigma; A21203; 1:700 dilution). Both primary and secondary antibodies were diluted in 1× PBS. Nuclei were counterstained with DAPI (Sigma, D1306, 1:1000 dilution) at room temperature for 10 min and slides were mounted with coverslips using DABCO mounting media (Sigma, D27802). Images for immunofluorescence staining on cryosections were captured by a Zeiss Axio Observer 7 inverted widefield microscope.

## TUNEL staining

The TdT-mediated dUTP Nick-End Labeling (TUNEL) assay was performed using the DeadEnd Fluorometric TUNEL System

(Promega; G3250) according to the manufacturer's protocol. In all, 5-µm tissue sections were first deparaffinized in two sequential incubations in xylene for 5 min each and then washed in 100% (v/v) ethanol at room temperature for 5 min. We next rehydrated the sections by sequential incubation in descending percentages of ethanol for 3 min each [100%, 95%, 85%, 70%, and 50% (v/v) ethanol]. After rehydrating the samples, slides were washed once in 0.85% (w/v) NaCl and once in 1× PBS at room temperature for 5 min each. The tissue sections were then fixed with 1× PBS containing 4% (w/v) paraformaldehyde at room temperature for 15 min. After washing the slides two times in 1× PBS for 5 min each, tissue sections were treated with 20 µg/ml Proteinase K at room temperature for 9 min. Following Proteinase K treatment, the tissue sections were washed once in 1× PBS at room temperature for 5 min and fixed with 1× PBS containing 4% (w/v) paraformaldehyde at room temperature for 5 min. Excess liquid was removed from the slides, and tissue sections were covered with 100 µl Equilibration Buffer at room temperature for 5 min. Tissue sections were then incubated in 50 µl rTdT incubation buffer (45 µl Equilibration Buffer, 5 µl Nucleotide Mix containing fluorescein-12-dUTP, 1 µl rTdT Enzyme) in a humidifying chamber at 37 °C for 1 h. To stop the reaction, the slides were immersed in 2× saline-sodium citrate for 15 min followed by three sequential washes in 1× PBS for 5 min each. Thereafter, tissue sections were stained with in 1× PBS containing 1 µg/ml propidium iodide at room temperature for 15 min in the dark. Finally, slides were washed three times in deionized water for 5 min each and mounted with one drop of DABCO (Sigma; D27802), and the slides were covered with a glass coverslip. Images were captured using a Zeiss Axio Observer 7 inverted widefield microscope.

## Western blotting

Frozen tissues were finely minced and homogenized in a Dounce homogenizer using 30 strokes of pestle B in RIPA lysis buffer (Thermo Fisher Scientific; 89900; 25 mM Tris-HCl, pH 7.6, 150 mM NaCl, 1% (v/v) NP-40, 1% (w/v) sodium deoxycholate, and 0.1% (w/v) SDS) supplemented with 1% (v/v) protease inhibitor (Thermo Fisher Scientific; 1861279). The homogenized tissues were incubated on ice for 30 min and then centrifuged at 20,000×*g* at 4 °C for 30 min. The supernatant was collected into a new tube, and protein concentration was determined using the BCA Protein Assay Kit (Thermo Fisher Scientific; 23227). 75 µg total protein from each sample was mixed with ¼ volume of loading dye (Invitrogen; NP0007) containing 0.2 M dithiothreitol and denatured at 95 °C for 5 min. Proteins were resolved by electrophoresis through a 4–20% Tris-Glycine gradient SDS gel (Thermo Fisher Scientific; XP04200) and transferred onto a 0.45 µm nitrocellulose membrane (Thermo Fisher Scientific; 88018). Membranes were blocked with Blocking Buffer (Thermo Fisher Scientific; 37565) at room temperature for 1.5 h. After blocking, the membranes were incubated at 4 °C overnight with primary antibodies (rabbit anti-NFYA, Bethyl, A302-105A, 1:500; mouse anti-ACTIN, Sigma, A1978, 1:5000). Next day, membranes were washed three times with 1× PBS-T (0.1% (v/v) Tween-20 in 1× PBS) for 30 min each, followed by incubation with secondary antibodies at room temperature for 1 h (anti-rabbit IRDye 800CW, LI-COR Biosciences, 926-32213, 1:15,000; and anti-mouse IRDye 680RD, LI-COR Biosciences, 926-68072, 1:15000). After three

additional washes with 1× PBS-T for 15 min each, protein signal was detected using the Odyssey Infrared Imaging System (LI-COR). Detected bands were quantified and normalized to ACTIN band intensity using ImageJ2.

## ATAC-seq library construction

We performed assay for transposase-accessible chromatin and sequencing (ATAC-seq) according to the manufacturer's protocol (Active Motif; 53150). Briefly, 100,000 FACS-purified spermatogonia (SpG), leptotene & zygotene (L/Z), and pachytene & diplotene (P/D) cells were pelleted by centrifuging at 500×*g* at 4 °C for 5 min. After carefully discarding supernatant, cells were gently washed with 100 µl ice-cold 1× PBS which was followed by an additional centrifugation at 500×*g* at 4 °C for 5 min. After carefully removing 1× PBS, cell pellet was resuspended in 100 µl ATAC lysis buffer which was immediately followed by centrifugation at 500×*g* at 4 °C for 10 min. After removing supernatant, nuclei were incubated in 50 µl Tagmentation Master Mix (25 µl 2× tagmentation buffer, 12 µl water, 10 µl assembled transposase, 2 µl 10× PBS, 0.5 µl 1% (v/v) Digitonin, and 0.5 µl 10% Tween-20) at 37 °C for 30 min using PCR machine equipped with a heated lid. Immediately following transposition, tagmented DNA was mixed with 250 µl DNA binding buffer and 5 µl sodium acetate. The mixture was then added onto a column and centrifuged at 17,000×*g* for 1 min. The column was then washed once with 750 µl wash buffer, and the tagmented DNA was eluted from the column with DNA Purification Elution Buffer. After purifying tagmented DNA, the library of fragments was amplified in 1× NEB Q5 PCR master mix (1× Q5 Reaction buffer [NEB; B9027S], 1 µM Illumina i7 Indexed primer, 1 µM Illumina i5 Indexed primer, 0.2 mM dNTPs, and 0.5 U Q5 High-Fidelity DNA polymerase [NEB; M0491S]). The following PCR condition was applied to generate ATAC-seq libraries: 72 °C for 5 min; 98 °C for 30 s; 10 cycles of thermocycling at 98 °C for 10 s, 63 °C for 30 s, and 72 °C for 1 min. ATAC-seq libraries were sequenced as 79 + 79 nt paired-end reads using NextSeq550 system (Illumina).

## Simultaneous single-cell ATAC-seq and mRNA-seq profiling

We performed BD Rhapsody Single-Cell Analysis System to simultaneously analyze the profile of mRNA expression and accessible chromatin state in cells from the testis of 11, 17, and 25 dpp wild-type mice and 8-week-old *Cre/+* and *Nfya-CKO* mice.

### Nuclei isolation for single-cell multiomics

Testes were dissected and squeezed into 6 ml 1× Gey's Balanced Salt Solution (GBSS, Sigma; G9779) containing 0.4 mg/ml collagenase type IV (Worthington; LS004188) and incubated at 33 °C for 15 min at 600 rpm. We then washed the tubules twice with 6 ml GBSS and incubated them in 1× GBSS containing 0.5 mg/ml trypsin (Gibco; 27250-018) and 1 µg/ml DNase I (Roche; 10104159001) at 33 °C for 15 min at 600 rpm. We thereafter homogenized the tubules using a Pasteur pipette at 4 °C on ice for 3 min. In total, 400 µl FBS was added to inactivate trypsin, and the cells were passed through a pre-wetted 70 µm cell strainer (Corning; CLS431751). Cells were then pelleted in a swing bucket centrifuge at 500×*g* for 10 min at 4 °C and resuspended in 3.5 ml 1×

GBSS, 175 µl FBS, 6 µl DNase I. After resuspension, the cells were filtered through a pre-wetted 40 µm cell strainer (Corning; 43750). In all, 300,000 cells were used for nuclei isolation. Cells were pelleted at 400×*g* for 10 min at 4 °C and gently resuspended in 195 µl swelling buffer (10 mM (w/v) Tris-HCl pH 7.5, 2 mM (w/v) MgCl₂, 3 mM (w/v) CaCl₂). Additional 1.8 ml swelling buffer was added to resuspended cells dropwise. Cells were then incubated on ice for 5 min and pelleted at 400×*g* for 10 min at 4 °C. Supernatant was carefully removed, and the cell pellet was resuspended in 100.5 µl prechilled lysis buffer *1* (9 mM (w/v) Tris-HCl pH 7.5, 1.8 mM (w/v) MgCl₂, 2.7 mM (w/v) CaCl2, 10% (v/v) glycerol, 1× protease inhibitor cocktail (PIC), 0.4 U/µl RNasin [Promega; N2615]). Afterwards, 100.5 µl lysis buffer *2* (9 mM (w/v) Tris-HCl pH 7.5, 1.8 mM (w/v) MgCl₂, 2.7 mM (w/v) CaCl2, 10% (v/v) glycerol, 10% (v/v) Igepal CA-360, 1× PIC, 0.4 U/µl RNasin [Promega; N2615]) was added dropwise to the mix and incubated on ice for 5 min. The nuclei were pelleted at 600×*g* at 4 °C for 5 min. The supernatant was discarded, and the nuclei were resuspended in 200.25 µl lysis buffer *3* (9 mM (w/v) Tris-HCl pH 7.5, 1.8 mM (w/v) MgCl₂, 2.7 mM (w/v) CaCl₂, 10% (v/v) glycerol, 0.01% (v/v) Digitonin, 1× PIC, 0.4 U/µl RNasin [Promega; N2615]). Afterward, an additional 1.8 ml lysis buffer *3* was added dropwise and carefully mixed. The nuclei were centrifuged at 500×*g* at 4 °C for 5 min. The supernatant was removed, and the nuclei were resuspended in 200.25 µl freezing buffer (50 mM (w/v) Tris-HCl pH 8, 5 mM (w/v) MgCl2, 40% (v/v) glycerol, 1× PIC, 0.4 U/µl RNasin [Promega; N2615]). The nuclei were centrifuged at 4 °C at 900×*g* for 6 min. The supernatant was discarded, and the nuclei were resuspended in 50 µl modified nuclei buffer (BD Biosciences; 51-9023091) supplemented with 2.5% (v/v) RNase inhibitor (BD Biosciences; 51-9024039) and 1 mM (v/v) DTT. ATAC-seq and mRNA-seq libraries were prepared from 50,000 nuclei.

## Simultaneous ATAC-seq and mRNA whole transcriptome (WTA) library preparation

To tagment DNA, 50,000 nuclei in 5 µl modified nuclei buffer (BD Biosciences; 667052) were mixed with 11.75 µl water, 2 µl 10× PBS, 1.25 µl RNase inhibitor (BD Biosciences; 51-9024039), 0.5 µl 1% (v/v) Digitonin, 0.5 µl 10% (v/v) Tween-20, 4 µl Tagmentase, and incubated at 37 °C for 30 min. After incubation, 300 µl modified sample buffer was added to the mix, and the nuclei were loaded into the BD Rhapsody 8-Lane Cartridge (BD Biosciences; 666262). The cartridge was incubated at room temperature for 8 min to allow nuclei to sediment in the micro-wells of the cartridge. Nuclei were then washed twice with cold sample buffer and incubated with splint beads (BD Biosciences; 667052) at room temperature for 3 min. Next, the cartridge was quickly agitated at room temperature for 10 s at 1000 rpm using ThermoMixer (Eppendorf; EP5382000015) and washed twice with cold sample buffer. Afterwards, 280 µl lysis buffer, supplemented with 10 mM DTT and 25 µl proteinase K, was added and incubated at 37 °C 10 min. After the lysis, the tagmented DNA and mRNA were immobilized onto beads via splint-oligo-bonded TSO and poly(T) oligo, respectively. We thereafter collected the beads retaining tagmented DNA and mRNA into a 1.7-ml tube, where we first ligated tagmented DNA to BD Rhapsody bead oligo in ligation mix (20 µl 10× ligation buffer, 10 µl DNA ligase [BD Biosciences; 41928], 5 µl RNase inhibitor [BD Biosciences; 51-9024039], 170 µl water) at

25 °C for 30 min at 1200 rpm. Afterwards, the ligation mix was discarded and reverse transcription (RT) was performed in 200 µl RT mix (40 µl RT buffer [BD Biosciences; 633773], 20 µl 10 mM dNTPs, 10 µl 0.1 M DTT, 12 µl RT/PCR enhancer, 10 µl RNase inhibitor [BD Biosciences; 633773], 10 µl reverse transcriptase [BD Biosciences; 633773], 98 µl water) was incubated at 42 °C for 30 min at 1200 rpm. Following RT reaction, RT mix was replaced by splint-oligo removal buffer and the tube was incubated at 60 °C for 5 min at 1200 rpm. To remove unused oligos on the beads, samples were then incubated in 200 µl Exonuclease I mix (20 µl 10 × exonuclease I buffer, 10 µl Exonuclease I [BD Biosciences; 633773], 170 µl water) at 37 °C for 30 min at 1200 rpm. We then quenched Exonuclease I by adding 4 µl 0.5 M EDTA to mix. To prepare ATAC-seq libraries, two rounds of DNA elution (80 µl elution) from the beads were performed. Note that after eluting DNA, the beads were stored in bead resuspension buffer at 4 °C to later prepare WTA libraries. 80 µl ATAC-seq libraries were amplified and indexed in 42 µl PCR index mix (30 µl PCR master mix [BD Biosciences; 41928], 6 µl ATAC-seq forward primer, 6 µl ATAC-seq reverse primer) with the following PCR condition: 98 °C 45 s, 12 cycles of thermocycling at 98 °C 10 s, 66 °C 30 s, 72 °C 30 s, and a final extension at 72 °C for 1 min.

To prepare WTA libraries, the beads, which were stored at 4 °C, were first heated at 95 °C for 5 min. Using a magnet, the supernatant was discarded, and the beads were incubated with 87 µl Random Primer mix (10 µl WTA extension buffer [BD Biosciences; 633801], 10 µl WTA extension primer, 67 µl water) at 95 °C for 5 min, followed by sequential incubation at 37 °C and 25 °C for 5 min each at 1200 rpm. Next, 13 µl Extension Enzyme mix (8 µl 10 mM dNTP, 12 µl ET/PCR enhancer, 6 µl WTA Extension Enzyme [BD Biosciences; 633801]) was added, and the mixture was incubated in ThermoMixer (Eppendorf; EP5382000015) at 1200 rpm with the following program: 25 °C 10 min, 37 °C 15 min, 45 °C 10 min, 55 °C 10 min. We then incubated the beads in 205 µl elution buffer at 95 °C for 5 min to purify Random Primer Extension (RPE) product from the beads. Note that WTA Random Priming and Extension, followed by purification, was repeated once more. A total of 400 µl RPE product was purified with 1.8× AMPure XP Beads (Beckman Coulter; A63881) and eluted in 40 µl Elution buffer. After elution, we added 80 µl PCR Master Mix containing 10 µl Universal oligo and 10 µl WTA Amplification primer directly to 40 µl eluted RPE product to perform initial amplification of WTA libraries with the PCR condition: 95 °C 3 min, 11 cycles of thermocycling at 95 °C 30 s, 60 °C 1 min, 72 °C 1 min, and a final extension at 72 °C for 2 min. The reaction was cleaned using 0.8 × AMPure XP Beads (Beckman Coulter; A63881). Before performing the final amplification, WTA libraries were analyzed on Bioanalyzer (Agilent; 2100 Bioanalyzer) to calculate the molar concentration of the DNA ranging from 150 to 600 bp. 10 µl 2 nM WTA libraries were indexed in 40 µl PCR index mix (25 µl PCR master mix [BD Biosciences; 633801], 5 µl BD Rhapsody library forward primer, 5 µl BD Rhapsody library reversed indexed primer, 5 µl water) with following PCR condition: 95 °C 3 min, 8 cycles of thermocycling at 95 °C 30 s, 60 °C 30 s, 72 °C 30 s, and a final extension at 72 °C for 1 min.

ATAC-seq and WTA libraries were sequenced using NovaSeq X Plus Platform (Illumina) with the following sequencing setup: for ATAC-seq libraries, Read 1, 50 cycles, Read 2, 50 cycles, Index 1, 8 cycles, Index 2, 60 cycles; for WTA libraries, Read 1, 51 cycles, Read

2, 71 cycles, Index 1, 8 cycles. We sequenced the libraries with the following manufacturer's recommended sequencing depth: for ATAC-seq libraries, 50,000 read pairs per cell; for WTA libraries, 100,000 read pairs per cell. Sequencing statistics are provided in Dataset EV7.

## CUT&RUN sequencing

Cleavage Under Targets and Release Using Nuclease (CUT&RUN) sequencing from FACS-purified germ cells was performed according to the manufacturer's protocol (Active Motif; 53180). In total, 500,000 FACS-purified spermatogonia (SpG), leptotene & zygotene (L/Z), and pachytene & diplotene (P/D) cells were pelleted by centrifugation at room temperature for 3 min at 600×g. The cell pellet was then gently resuspended in 100 µl nuclear isolation buffer (20 mM (w/v) HEPES-KOH, pH 7.9, 10 mM (w/v) KCl, 0.5 mM (v/v) Spermidine, 0.1% (v/v) Triton X-100, 20% (v/v) glycerol) supplemented with 1× protease inhibitors, 0.5 mM Spermidine) and incubated on ice for 10 min. After incubation, cells were centrifuged at 600×g at 4 °C for 3 min. After removing the supernatant carefully, nuclei were washed once and resuspended in 100 µl wash buffer. 100 µl nuclei suspension was incubated at room temperature for 10 min with 10 µl Concanavalin A-coated paramagnetic beads to immobilize nuclei on beads. After immobilizing beads, supernatant was removed using magnet and 50 µl antibody buffer containing 1 µg anti-NFYA (Bethyl; A302-105A) or 0.5 µg IgG (EpiCypher; 13-0042) antibodies was added directly to the beads. Samples were incubated at 4 °C overnight on a nutator. The next day, beads were washed twice with 200 µl cell permeabilization buffer (CPB) and resuspended in 50 µl CPB. Afterward, 2.5 µl ChIC/CUT&RUN pAG-MNase was added to each sample, and samples were incubated at room temperature for 10 min. Beads were then washed twice with 200 µl CPB and resuspended in 50 µl CPB. We then added 1 µl 0.1 M CaCl₂ to each sample and incubated them at 4 °C for 2 h on a nutator. The reaction was stopped by adding 40 µl STOP solution and incubated at 37 °C for 10 min in a thermalcycler whose lid was at 65 °C. Samples were then placed on a magnet, and the clear supernatant containing the DNA fragments was moved to a new tube. DNA was then purified by column purification provided with the kit. After purifying the DNA, CUT&RUN libraries were prepared using NEBNext Ultra II DNA Library Prep Kit (NEB; E7645) according to the manufacturer's protocol. CUT&RUN libraries were sequenced as 79 + 79 nt paired-end reads using NextSeq550 system (Illumina). Sequencing statistics are provided in Dataset EV7.

## CUT&Tag sequencing

Cleavage Under Targets and Tagmentation (CUT&Tag) protocol was performed as previously described (Henikoff et al, 2020). In all, 80,000 cells were bound to 5 µl Concanavalin A-coated magnetic beads (Cell Signaling; 82307S) at room temperature for 10 min. After binding, the supernatant was removed from the beads and 40 µl antibody buffer (20 mM (w/v) HEPES pH 7.5, 150 mM (w/v) NaCl, 0.5 mM (w/v) Spermidine, 1% (v/v) Triton-X100, 1× protease inhibitor cocktail (PIC), 0.1% (w/v) BSA, 2 mM (w/v) EDTA) containing 1:50 (v/v) Pol II Ser5P antibody (Cell signaling; 13523) was added and the beads were carefully resuspended by gentle

vortexing. Primary antibody was incubated at room temperature for 1 h, after which the supernatant was removed and the cell/bead mixture was resuspended in 40 µl Triton-Wash buffer (20 mM (w/v) HEPES pH 7.5, 150 mM (w/v) NaCl, 0.5 mM (w/v) Spermidine, 1% (v/v) Triton-X100, 1× PIC) containing 1:100 of secondary antibody (guinea pig anti-rabbit (Antibodies online; ABIN101961)) and incubated at room temperature for 30 min. Next, the cells were washed once with 200 µl Triton-Wash buffer before being resuspended in 40 µl Triton-300-wash buffer (20 mM (w/v) HEPES pH 7.5, 300 mM (w/v) NaCl, 0.5 mM (w/v) Spermidine, 1% (v/v) Triton-X100, 1× PIC) containing 1:25 preloaded pAG-Tn5 fusion protein (Epicypher; 15-1117) and incubated at room temprature for 1 h. After binding of the pAG-Tn5, cells were washed once with 200 µl Triton-300-wash buffer, before being resuspended in 50 µl CUTAC-DMF tagmentation solution (10 mM (w/v) TAPS pH 8.5, 5 mM (w/v) MgCl₂, 20% (v/v) DMF) and incubated in a thermocycler at 55 °C for 1 h. After tagmentation, cells were washed once with 50 µl TAPS wash (10 mM (w/v) TAPS pH 8.5, 0.2 mM (w/v) EDTA) before resuspended in 5 µl 1% SDS-ProtK release solution (1% (w/v) SDS, 10 mM (w/v) TAPS pH 8.5) containing 1:11 Thermolabile Proteinase K (NEB; P8111S), thoroughly mixed and incubated in a thermalcycler at 37 °C for 1 h followed by an additional incubation at 58 °C for 1 h. After incubation, 15 µl 6% Triton-X100 was added with 2 µl 10 µM i7 and 2 µl 10 µM i5 indexed primers. The supernatant was then transferred to a new PCR tube, and 25 µl NEBNext 2X PCR master mix (NEB; ME541L) was added, and the samples were incubated in a thermalcycler with the following program: 58 °C for 5 min, 72 °C for 5 min, 98 °C for 5 min, 11 cycles of 98 °C for 10 s, 60 °C for 10 s. Final extension at 72 °C for 1 min. After PCR, the libraries were purified using 65 µl Ampure XP beads (Beckman Coulter; A63880) and resuspended in 22 µl (w/v) Tris-HCl pH 8.5. CUT&Tag libraries were sequenced as 79 + 79 nt paired-end reads using NextSeq550 system (Illumina). Sequencing statistics are provided in Dataset EV7.

## Long RNA library analysis

RNA-seq analysis was performed as previously described (Gainetdinov et al, 2021; Cecchini et al, 2023; Yu et al, 2023). Molecules of transcripts per cell were calculated as previously described (Gainetdinov et al, 2018). Given that each sample contained ~623,291,645 molecules of ERCC spike-in mix, the abundance of each gene = (number of mapped reads × 623291645)/(number of cells used to prepare the library × the number of reads mapping to the ERCC spike-in sequences).

## PRO-seq analysis

We re-analyzed publicly available, GSE228454, Precision Run-On sequencing (PRO-seq) data from purified spermatogonia (SpG), primary spermatocytes (SpI), and round spermatids (RS). Adapter sequences were trimmed from raw read pairs using Cutadapt (v4.8 (Martin, 2011)) with a 10% error rate threshold. Afterwards, additional one nucleotide was removed from the end of read 1 (R1). Subsequently, reads were aligned to mouse reference genome (mm10) and *Drosophila melanogaster* reference genome (dm6), which served as spike-in control, using Bowtie2 (v2.5.4 (Langmead and Salzberg, 2012)) with the parameters —local —very-

sensitive —no-unal —no-mixed —no-discordant -|10 -X700. We specified insert size ranging from 10 to 700 bp to ensure accurate and high-confidence alignments. SAM files were converted to BAM format using SAMtools (v1.20 (Danecek et al, 2021)), and BAM files were subsequently sorted, indexed, and filtered to retain only uniquely mapped reads. Final indexed BAM files were then converted into bigwig format using deepTools (v3.3.2 (Ramírez et al, 2016)). Each bigwig file was normalized to scale factor. Scale factor was calculated by dividing the number of spike-in reads by the total number of mapped reads. Because round spermatids have haploid genome, scale factor for RS was halved. Metagene plots were generated using PRO-seq reads normalized to scale factor with plotProfile function from deepTools (v3.3.2 (Ramírez et al, 2016)). PRO-seq statistics are provided in Dataset EV7.

To examine the PRO-seq peaks near transcription start sites (TSSs) of genes, peaks were called using MACS3 (v3.0.2 (Zhang et al, 2008); FDR < 0.1) with parameters —q 0.1 —min-length 150 in each biological replicate from SpG, SpI, and RS. Bed file containing genomic coordinates for all annotated TSSs was retrieved from UCSC Genome Browser (mm10; GENCODE VM23). Distance from the TSS to the nearest PRO-seq peak summit for each gene was calculated. If a meiosis-I or spermiogenesis gene had a PRO-seq peak ±2 kb of its TSS in at least two replicates of PRO-seq from SpG, it was considered as *poised* gene. Note that for those genes with multiple TSSs, poised classification was assigned if at least one of the TSSs had a PRO-seq peak within ±2 kb in at least two biological replicates.

To study promoter-proximal Pol II pausing, we quantified PRO-seq reads at promoter-proximal regions and gene bodies of genes using BEDTools (v.2.31.0 (Quinlan and Hall, 2010)). Promoter-proximal region is defined as the first 5% of the gene from its transcription start site, whereas gene body is defined as a region starting from the first 30% a gene to the transcription end site (TES). The pausing index is calculated by dividing the reads at promoter-proximal region by gene body reads. A two-sided Wilcoxon matched-pairs signed-ranked sum test was applied to measure the statistical difference for the same gene categories between different germ cells.

## ATAC-seq analysis

Adapter sequences and low-quality reads were removed from the raw ATAC-seq reads using Fastp (v.0.24.0 (Chen et al, 2018b)). Subsequently, reads were aligned to the mouse reference genome (mm10) using Bowtie2 (v2.5.4 (Langmead and Salzberg, 2012)) with the parameters —very-sensitive —no —unal —no-mixed —no-discordant. Multiple aligning read pairs were removed using SAMtools (v1.20 (Danecek et al, 2021)). We then identified and removed duplicate reads using the MarkDuplicates function from picard-tools (v.3.3.0). BAM alignments were then converted to bigwig format using deepTools (v.3.3.2 (Ramírez et al, 2016)) and normalized to reads per million mapped reads (RPM) accounting for differences in sequencing depth. ATAC-seq peaks were called in each biological replicate from spermatogonia (SpG), leptotene/zygotene (L/Z), and pachytene/diplotene (P/D) using MACS3 (v3.0.2 (Zhang et al, 2008); FDR < 0.01). ATAC-seq statistics are provided in Dataset EV7.

Distance from the TSS to the nearest ATAC-seq peak summit for each gene was calculated. Genes that retain ATAC-seq peak ±2 kb of their TSSs in at least two replicates of ATAC-seq experiments were considered as genes with accessible chromatin. HOMER (v4.0 (Heinz et al, 2010)) was employed to identify TF binding motifs under ATAC-seq peaks around the promoters of poised genes in SpG cells. Motif analysis scanning ±75 bp from the center of each peak was allowed for up to 25 motifs per peak. Promoters of genes, whose transcript abundance remained constant across germ cells, served as background control (Dataset EV1).

## CUT&RUN analysis

Adapter sequences were trimmed, and low-quality reads were removed from the raw paired-end CUT&RUN reads using Fastp (v0.24.0 (Chen et al, 2018b)). Reads were then aligned to the mouse reference genome (mm10) using Bowtie2 (v2.5.4 (Langmead and Salzberg, 2012)) with parameters —local —very-sensitive —no-unal —no-mixed —no-discordant -|10 -X700. Subsequently, SAM files were converted to BAM format using SAMtools (v1.20 (Danecek et al, 2021)), and alignments whose quality is less than 20 were removed. BAM files were then sorted and indexed, and multiple aligning read pairs were removed. Thereafter, duplicate reads were removed using MarkDuplicates function from picard-tools (v.3.3.0). Bigwig files were generated using bamCoverage function from deepTools (v3.3.2 (Ramírez et al, 2016)) and normalized to reads per million (RPM). CUT&RUN statistics are provided in Dataset EV7.

NFYA peaks were identified using MACS3 (v3.0.2 (Zhang et al, 2008); FDR < 0.1). CUT&RUN for IgG antibody was used as the background control to call significant NFYA peaks. Genes with NFYA peak within ±2 kb of their transcription start sites in at least two replicates of CUT&RUN experiments were considered as NFYA-bound genes.

## CUT&Tag analysis

Adapter sequences and low-quality bases were removed from the raw paired-end CUT&Tag reads using Fastp (v0.24.0 (Chen et al, 2018b)). Reads were then aligned to the mouse reference genome (mm10) and *E. coli* spike-in genome (strain MG1655, GenBank accession, U00096.3) using Bowtie2 (v2.5.4 (Langmead and Salzberg, 2012)) with the parameters —local —very-sensitive —no-unal —no-mixed —no-discordant -|10 -X700. Aligned reads were converted from SAM to BAM format, sorted and indexed using SAMtools (v1.20 (Danecek et al, 2021)). Thereafter, duplicate reads were removed using MarkDuplicates function from picard-tools (v.3.3.0). Only uniquely mapped reads were used for downstream analysis. Normalized Bigwig coverage tracks were generated bamCoverage function from deepTools (v3.3.2 (Ramírez et al, 2016)) after applying a scale factor that is calculated by dividing 10,000 to the number of *E. coli* reads in each sample.

## Simultaneous single-cell ATAC-seq and mRNA-seq analysis

### Cell barcode annotation

Valid cell barcodes were downloaded from BD Biosciences. For scRNA-seq, read 1 (R1) contains the information of barcodes and

unique molecular indices (UMIs), while read 2 (R2) are from complementary DNA (cDNA). R1 retains 35 to 38 nucleotides cell barcodes followed by 8 nucleotide UMIs. For scATAC-seq, read I1 (Rl1) marked barcode and UMIs, while R1 and R2 are paired DNA sequences. The construction of Rl1 is 35 nucleotides cell barcodes followed by 8 nucleotides UMIs in reverse complement. Allowing 0 mismatch, raw reads without valid cell barcodes were filtered. For the remaining reads, information of cell barcodes and UMIs was assigned to the read names of the cDNA end (scRNA-seq) or the paired DNA end (scATAC-seq).

### Read alignment

We used STAR (v2.7.10b (Dobin et al, 2013)) to align scRNA-seq reads to the mouse reference genome (mm10) using parameters `--outFilterIntronMotifs RemoveNoncanonicalUnannotated –outFilterMultimapNmax 1`. PCR duplications with the same UMIs and the same alignment coordinates from the same cells were removed. Afterwards, a cell-by-gene sparse matrix was made for each sample.

We used bwa-mem (v0.7.12) to align scATAC-seq reads to the mouse reference genome (mm10) using default parameters. The aligned files were converted to bed format using BEDTools (v2.27.1 (Quinlan and Hall, 2010)), with $+4/-5$ coordinate shift for plus/minus strand reads to account for the 9 bp apart cuts by Tn5 transposase. PCR duplications with the same UMIs and the same alignment coordinates from the same cells are removed. A fragment file was then generated for each sample.

### Cell type annotation

Cell type annotation was done using the Seurat (v5.2.1 (Hao et al, 2021)) R package for scRNA-seq. We first removed cells with <500 expressed genes or ≥5% UMIs from mitochondrial genes. The gene UMI counts for each of the remaining cell were log normalized with a factor equal to 10,000. The top 5000 variable genes were then extracted for principal component analysis (PCA). The top 20 PCs were further used for clustering (resolution = 0.8) and uniform manifold approximation and projection (UMAP) analysis. For each sample, we then used the DoubletFinder R package (v2.0.4 (McGinnis et al, 2019)) to remove doublets and multiplets using the top 10 PCs and default parameters. A total of 33 cell clusters were identified. We assigned cell-type to each cell cluster using the expression of marker genes (Appendix Fig. S11; Dataset EV4B). Clusters without clear marker gene expression were removed from downstream analysis.

Cell-type annotation for scATAC-seq was performed using ArchR R package (v1.0.3 (Granja et al, 2021)). We filtered cells with <1000 fragments or a TSS score <4. Doublets/multiplets were removed using ArchR for each sample with default parameters. Genome-wide tile matrix of 500 bp and UMAP analysis with default parameters were used for clustering. A total of 25 cell clusters were identified. We then assigned cell-type for each cluster based on the cell-type annotation generated from scRNA-seq analysis for those cells with both open chromatin and transcriptome captured, along with the chromatin accessibility of marker genes.

### Pseudo-bulk analysis

According to the cell-type annotation, we generated pseudo-bulks by merging all aligned reads/fragments for each cell type for scRNA-seq and scATAC-seq, respectively. We then normalized each pseudo-bulk to reads/fragments per million mapped reads/fragments and generated bigwig tracks. Chromatin accessibility of gene promoters is defined by the average normalized coverage at ±200 bp from TSS.

### ChIP-seq analysis

To identify distal enhancer regions in the genomes of spermatogonia (SpG) and pachytene/diplotene spermatocytes (P/D), we re-analyzed publicly available H3K4me1 (GSE131656 (Cheng et al, 2020)) and H3K4me3 (GSE132446 (Cheng et al, 2020)) ChIP-seq data from SpG, as well as H3K27Ac (GSE107398 (Adams et al, 2018)) ChIP-seq data from P/D, along with our H3K4me3 CUT&RUN from P/D. ChIP-seq data processing was performed according to the pipeline used for CUT&RUN analysis. Because H3K4me3 is associated with promoter-proximal regions, while H3K4me1 and H3K27Ac are found both at distal enhancers and promoter-proximal regions (Heintzman et al, 2007; Zentner et al, 2011), we employed an exclusion criterion to identify distal enhancer regions. Specifically, for SpG cells, we excluded H3K4me1 peaks that overlapped with H3K4me3 peaks. Similarly, for P/D cells, H3K27Ac peaks overlapping with H3K4me3 were removed. Remaining H3K4me1 and H3K27Ac were then considered as distal enhancer regions. Subsequently, those NFYA peak summits that reside within the coordinates of H3K4me1 and H3K27Ac peaks non-overlapping with H3K4me3 were considered as NFYA peaks at enhancers.

### Quantification and statistical analysis

Statistical analyses and graph generation were conducted using R v4.3.2 (https://www.rstudio.com) or Prism 10.1.1 (GraphPad Software, LLC). Boxplots were used to present data distribution. Boxes represent interquartile ranges (IQRs), spanning the first and third quartiles. Outliers were defined as data points with values > third quartile + 1.5 × IQR or lower than the first quartile − 1.5 × IQR, where IQR is the difference between the maximum of the third and the minimum of the first quartile. Relationship between variables was measured using Spearman's correlation ($\rho$). Two-sided Wilcoxon matched-pairs signed-ranked sum test was used to calculate $P$ values in Figs. 1B, 5D, EV2B, EV3B, EV4B,E, and EV5C. Two-sided unpaired $t$ test was used to calculate $P$ values in Fig 3B,C, 3E–G, Appendix Figs. S8E,F, S9B, and S10. Benjamini–Hochberg correction was used to compute the false discovery rate in Fig. 6A.

## Data availability

Sequencing data are available from the National Center for Biotechnology Information Sequence Read Archive using Gene Expression Omnibus (GEO) accession number GSE295991 and the BioStudies database with accession number E-MTAB-16210. All Scripts used to analyze sequencing data can be found here: https://github.com/Ozatalab. Microscopy images for Fig. 3 are uploaded to BioImage Archive: S-BIAD2388. All publicly available data were downloaded from GEO from the following databases:

RNA sequencing from FACS-sorted C57BL/6J germ cells (Gainetdinov et al, 2021; Cecchini et al, 2023; Yu et al, 2023): SRA; PRJNA660633.

Precision Run-On sequencing data from purified male germ cells (Kaye et al, 2024): GSE228454.

Assay for transposase-accessible chromatin sequencing from embryonic stem cells (Oldfield et al, 2019): GSE115110.

H3K4me1 ChIP-seq from spermatogonia (Cheng et al, 2020): GSE131656.

H3K4me3 ChIP-seq from spermatogonia (Cheng et al, 2020): GSE132446.

H3K27Ac ChIP-seq from pachytene spermatocytes (Adams et al, 2018): GSE107398.

The source data of this paper are collected in the following database record: biostudies:S-SCDT-10_1038-S44318-026-00756-6.

## Peer review information

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

## Acknowledgements

We thank the personnel of Stockholm University animal facility, with particular gratitude to S Oerther and R Askar for their excellence in mouse colony management; C Molenaar for his expert help with Zeiss Axio Observer 7 inverted widefield microscope; Matthew Hunt for proofreading English grammar and clarity; Shruti Jain for scientific discussions; National Genomics Infrastructure—funded by Science for Life Laboratory, the Knut and Alice Wallenberg Foundation and the Swedish Research Council—, and SNIC/Uppsala Multidisciplinary Center for assistance with massively parallel sequencing and access to the UPPMAX computational infrastructure. This work was supported by the Swedish Research Council grant 2020-03818 (DMÖ), the Swedish Research Council grant 2024-04321 (DMÖ), and Carltryggersstiftelse CTS 21:1158 (DMÖ).

## Author contributions

**Martin Säflund**: Conceptualization; Investigation; Visualization; Writing—original draft; Writing—review and editing. **Masomeh Askari**: Data curation; Visualization; Writing—original draft; Writing—review and editing. **Atiyeh Eghbali**: Investigation; Writing—original draft; Writing—review and editing. **Mukhtar Mohamed Abdi**: Investigation. **Dilay Deren Er**: Investigation. **Ann-Kristin Iréne Östlund Farrants**: Supervision. **Tianxiong Yu**: Conceptualization; Visualization; Writing—original draft; Writing—review and editing. **Deniz M Ozata**: Conceptualization; Supervision; Funding acquisition; Investigation; Writing—original draft; Project administration; Writing—review and editing.

Source data underlying figure panels in this paper may have individual authorship assigned. Where available, figure panel/source data authorship is listed in the following database record: biostudies:S-SCDT-10_1038-S44318-026-00756-6.

## Funding

## Disclosure and competing interests statement

The authors declare no competing interests.

# Expanded View Figures

**Figure EV1.** **Paused polymerase accumulates around the promoters of genes expressed during meiosis in spermatogonia.**

Related to Fig. 1. (**A**) Violin plots represent the distance (bp) from TSS to nearest promoter PRO-seq signal of poised and non-poised meiosis-I (left) and spermiogenesis (right) genes in spermatogonia (SpG), primary spermatocytes (SpI), and round spermatids (RS). Horizontal lines represent the mean. Whiskers represent maximum and minimum values. Interquartile range (IQR) represented by boxplots. (**B**) Heatmaps show the relative promoter (left; from TSS to first 5% of gene length) and gene body (right; from first 30% of gene length to TES) PRO-seq signal for mitosis genes in spermatogonia (SpG), primary spermatocytes (SpI), and round spermatids (RS). (**C**) Line plots represent the average promoter (dashed line with triangle) and gene body (line with squares) relative PRO-seq signal for mitosis, meiosis-I, and spermiogenesis genes across SpG, SpI, and RS cells.

**A**

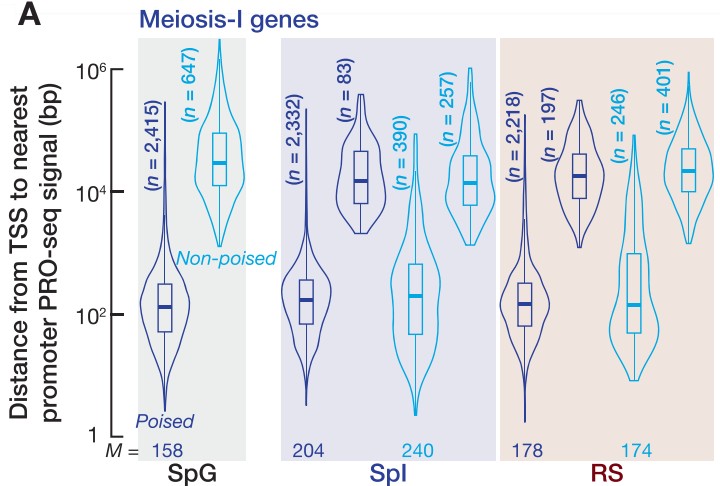

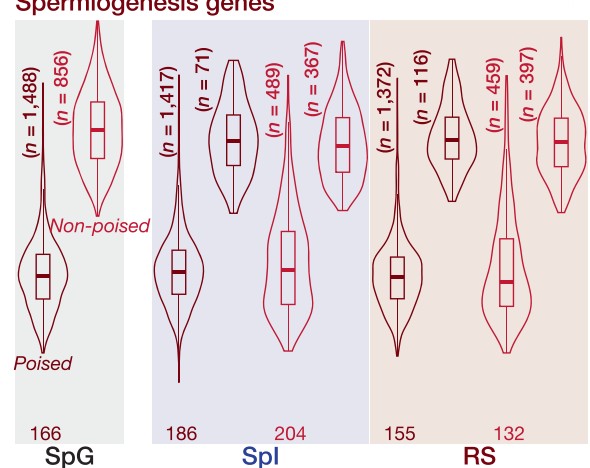

**B**

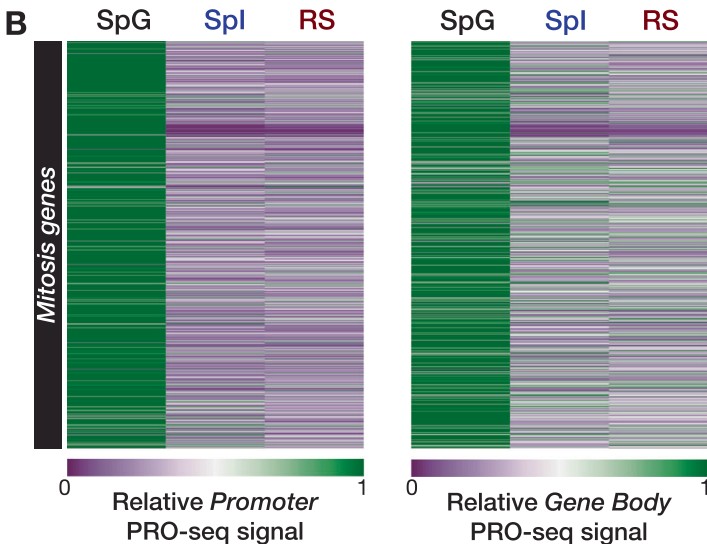

**C**

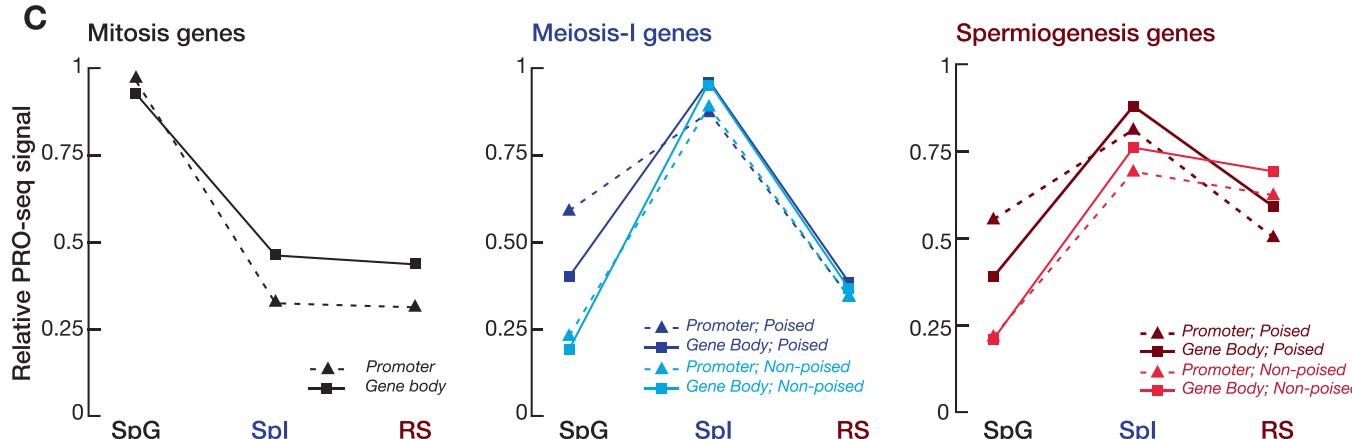

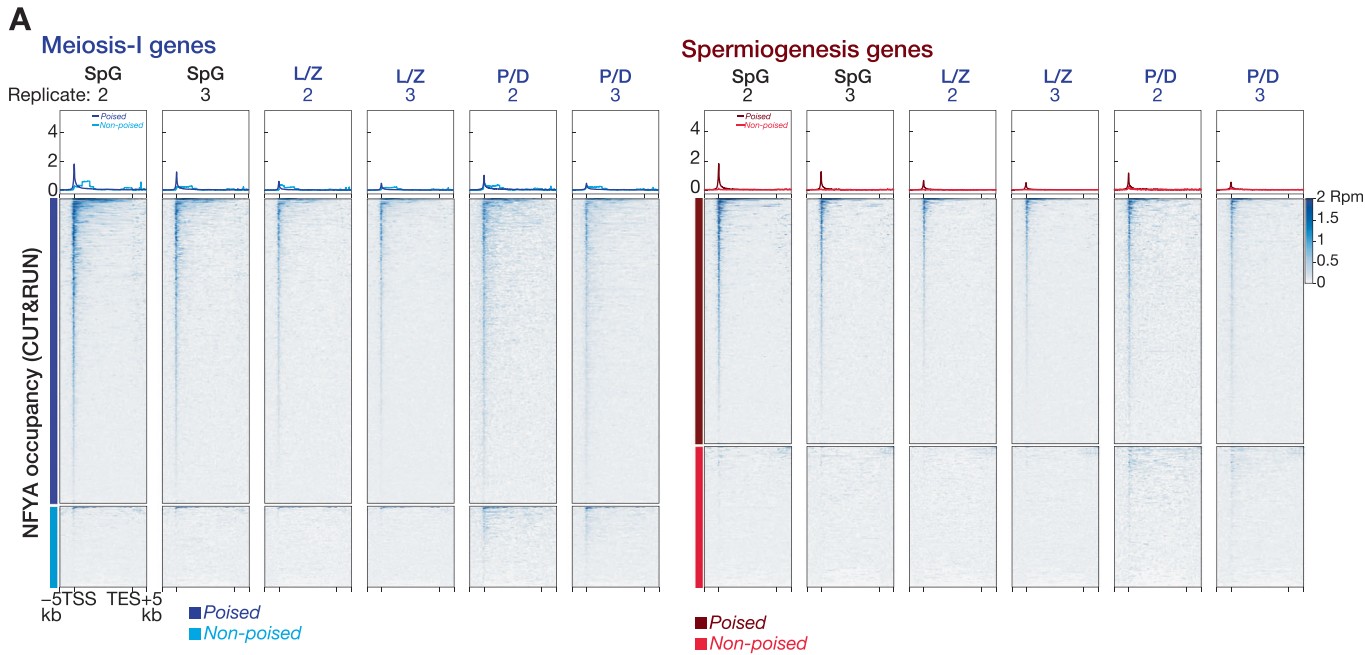

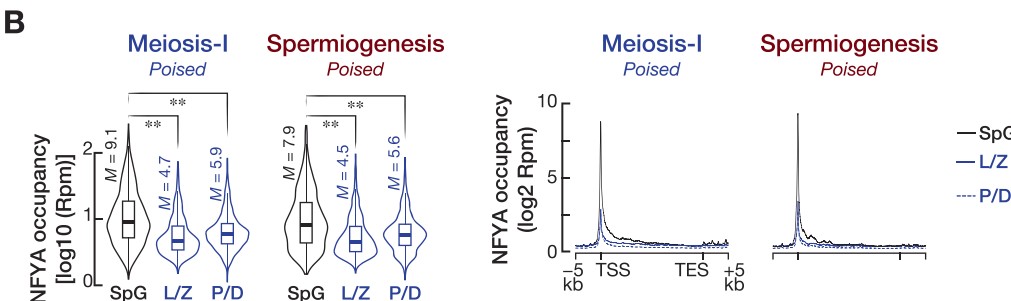

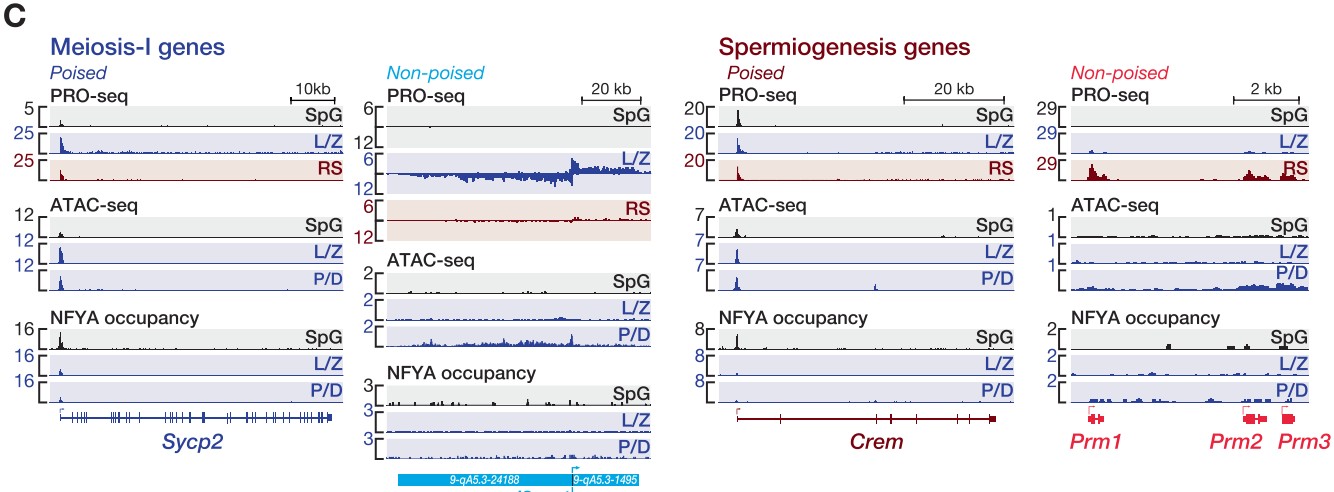

◀ **Figure EV2. NFYA binds the promoters of poised genes in spermatogonia.**

Related to Fig. 2. (A) Metagene plots (top) and heatmaps (bottom) show Rpm-normalized NFYA CUT&RUN signals in the −5 kb to +5 kb window flanking transcription start sites (TSSs) and transcription end sites (TESs) of poised and non-poised genes in the second and third replicates from SpG, L/Z, P/D. (B) Violin plots (left) show NFYA occupancy around the promoters of poised meiosis-I and spermiogenesis genes in SpG, L/Z, and P/D. Horizontal lines represent the median. Whiskers represent maximum and minimum values. IQR represented by boxplots. **$P < 0.01$; two-sided Wilcoxon matched-pairs signed-ranked sum test. Meiosis-I genes in SpG vs L/Z $P < 2.2 \times 10^{-16}$, SpG vs P/D $P = 1.566 \times 10^{-249}$; Spermiogenesis genes in SpG vs L/Z $P = 1.474 \times 10^{-220}$, SpG vs P/D $P = 3.384 \times 10^{-131}$. Metagene plots (right) show Rpm-normalized NFYA CUT&RUN signals in the −5 kb to +5 kb window flanking transcription start sites (TSSs) and transcription end sites (TESs) of poised and non-poised genes in the second and third replicates from SpG, L/Z, P/D. Average of three biological replicates ($n = 3$). (C) IGV tracks of PRO-seq signal, ATAC-seq signal, and NFYA occupancy (CUT&RUN) at gene boundaries of exemplified poised and non-poised meiosis-I (*Sycp2, pi9*) and spermiogenesis (*Crem, Prm1, Prm2, Prm3*) genes from SpG, L/Z, P/D, and RS.

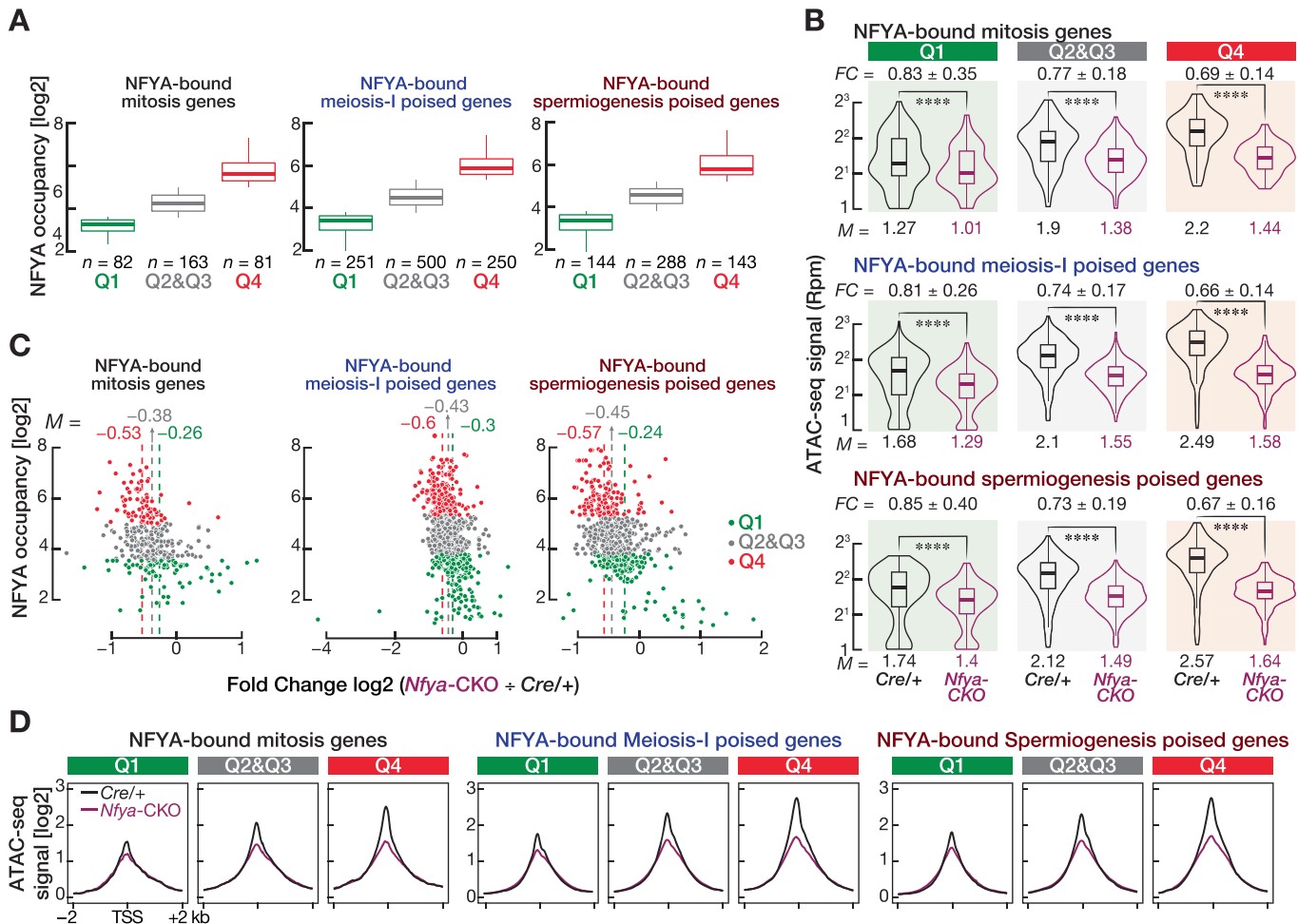

**Figure EV3. Degree of NFYA occupancy correlates with the degree of reduction in accessible chromatin.**

Related to Fig. 5. (A) Boxplots show rank-ordered NFYA-bound mitosis, meiosis-I and spermiogenesis poised genes relative to the NFYA occupancy at their promoters in SpG of *Cre/+* mice. Q1 denotes the first quartile; Q2 and Q3 represent the second and third quartiles; Q4 represents the fourth quartile. Horizontal lines represent the mean. Whiskers represent maximum and minimum values. IQR represented by boxplots. (B) Reads per million (Rpm)-normalized ATAC-seq signals at the promoters of NFYA-bound mitosis, poised meiosis-I, and poised spermiogenesis genes that are classified according to quartiles (NFYA-bound mitosis genes, Q1, $n = 82$, Q2&Q3, $n = 163$, Q4, $n = 81$; NFYA-bound meiosis-I genes, Q1, $n = 251$, Q2&Q3, $n = 500$, Q4, $n = 250$; NFYA-bound spermiogenesis genes, Q1, $n = 144$, Q2&Q3, $n = 288$, Q4, $n = 143$). FC fold change. Horizontal lines represent the mean. Whiskers represent maximum and minimum values. Interquartile range (IQR) represented by boxplots. Two-sided Wilcoxon matched-pairs signed-ranked sum test. ****$P < 0.0001$. NFYA-bound mitosis genes Q1 *Nfya*-CKO vs *Cre/+* $P = 7.032 \times 10^{-8}$, Q2&Q3 $P = 8.177 \times 10^{-25}$, Q4 $P = 9.549 \times 10^{-15}$; NFYA-bound meiosis-I poised genes Q1 *Nfya*-CKO vs *Cre/+* $P = 1.358 \times 10^{-29}$, Q2&Q3 $P = 4.496 \times 10^{-79}$, Q4 $P = 2.843 \times 10^{-42}$; NFYA-bound spermiogenesis poised genes Q1 *Nfya*-CKO vs *Cre/+* $P = 1.729 \times 10^{-17}$, Q2&Q3 $P = 2.152 \times 10^{-45}$, Q4 $P = 1.283 \times 10^{-24}$. (C) Scatter plots show the change in accessible chromatin between the SpG from *Nfya*-CKO and *Cre/+* relative to their NFYA occupancy. NFYA-bound genes are presented in quartiles. (D) Metagene plots of ATAC-seq signal (log2) around the TSS ± 2 kb of NFYA-bound mitosis, meiosis-I, and spermiogenesis poised genes in SpG of *Cre/+* and *Nfya*-CKO mice.

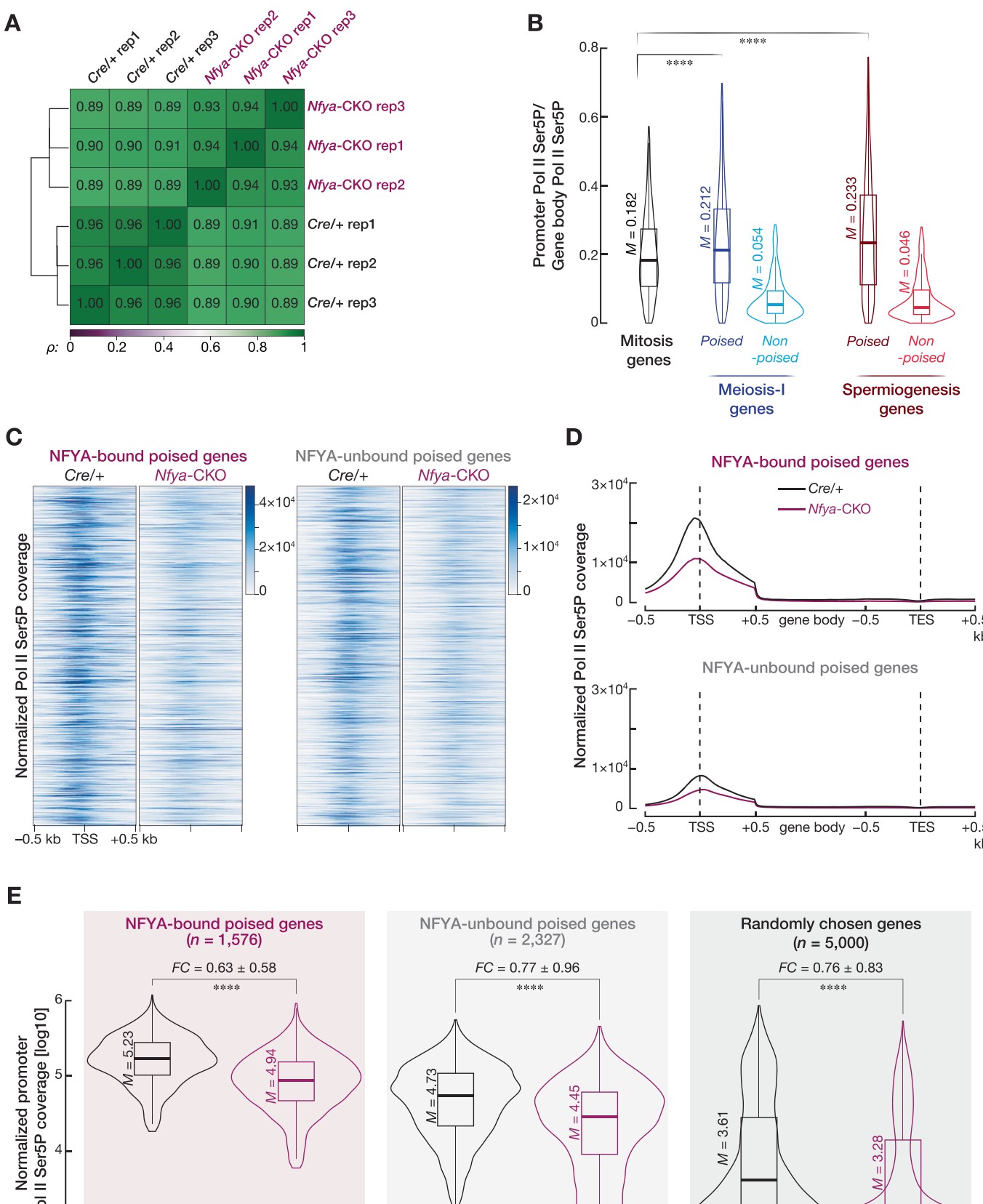

**Figure EV4.** *Nfya* **deletion impairs paused Pol II occupancy around the TSSs of poised genes in spermatogonia.**

Related to Fig. 5. (A) Heatmap shows Spearman's correlation between the biological replicates of Pol II Ser5P CUT&Tag from SpG of *Cre/+* and *Nfya-CKO*. Agreement between replicates; SpG from *Cre/+* Spearman's ρ > 0.95; SpG from *Nfya-CKO* Spearman's ρ > 0.92. (B) Violin plots represent the density (average of three biological replicates) of Pol II Ser5P between +31 to +60 nt from the TSS relative to the Pol II Ser5P density at the gene body for poised and non-poised meiosis-I and spermiogenesis genes, and mitosis genes in SpG from *Cre/+* (*n* = 3). Horizontal lines represent the mean. Whiskers represent maximum and minimum values. Interquartile range (IQR) represented by boxplots. Two-sided Wilcoxon matched-pairs signed-rank sum test. ****$P < 0.0001$. Mitosis vs poised meiosis-I genes $P = 1.253 \times 10^{-13}$, mitosis vs poised spermiogenesis genes $P = 5.454 \times 10^{-9}$. (C) Heatmaps (average of three biological replicates) show the normalized Pol II Ser5P coverage around the TSSs of NFYA-bound and -unbound poised genes in SpG of *Cre/+* and *Nfya-CKO*. (D) Metagene plots (average of three biological replicates) show the normalized Pol II Ser5P coverage around the TSSs, gene body, and TESs of NFYA-bound and -unbound poised genes in SpG of *Cre/+* and *Nfya-CKO*. (E) Violin plots show the change in the *E. coli* spike-in normalized coverage of Pol II Ser5P within ±0.5 kb of the TSSs of NFYA-bound and -unbound poised genes and 5,000 randomly chosen genes in SpG of *Cre/+* and *Nfya-CKO*. Horizontal lines represent the mean. Whiskers represent maximum and minimum values. Interquartile range (IQR) represented by boxplots. Two-sided Wilcoxon matched-pairs signed-rank sum test. ****$P < 0.0001$. NFYA-bound poised genes *Cre/+* vs *Nfya-CKO* $P = 3.33 \times 10^{-98}$; NFYA-unbound poised genes *Cre/+* vs *Nfya-CKO* $P = 3.28 \times 10^{-70}$; Randomly chosen genes *Cre/+* vs *Nfya-CKO* $P = 1.77 \times 10^{-88}$.

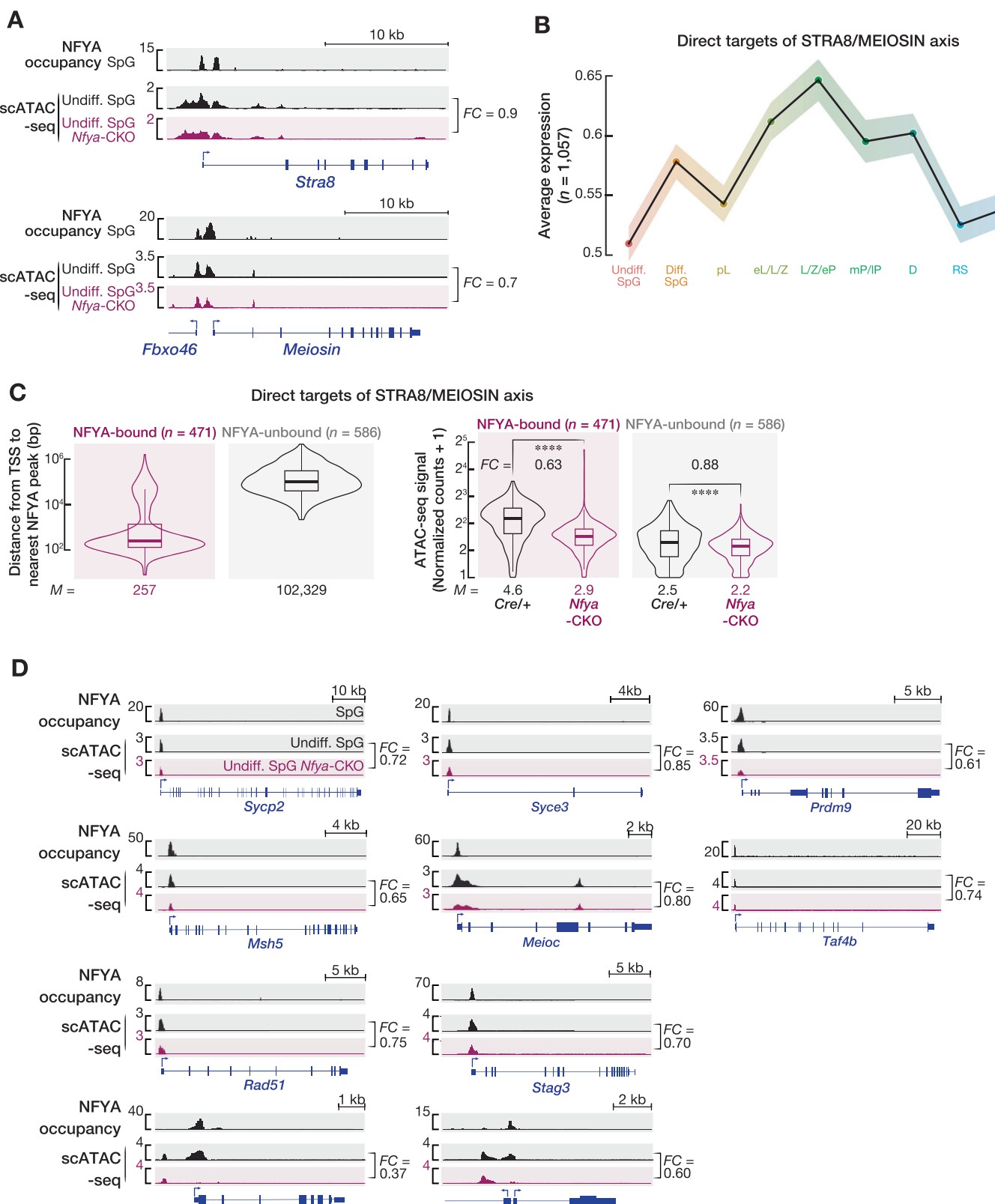

**Figure EV5.  NFYA regulates accessible chromatin at the promoters of genes activated by the STRA8/MEIOSIN axis.**

Related to Figs. 4, 5, and 6. (**A**). IGV tracks of NFYA occupancy (CUT&RUN) and scATAC-seq signal, at gene boundaries of *Stra8* and *Meiosisn* in SpG of *Cre/+* and *Nfya-CKO* mice. (**B**) Line plot of average expression of STRA8/MEIOSIN targets in nine different germ cell populations defined by scRNA-seq. (**C**) Violon plots of distance from TSS of nearest NFYA peak (left) of genes directly targeted of STRA8/MEIOSIN axis in SpG. ATAC-seq signal (right) of genes directly targeted of STRA8/MEIOSIN axis in SpG of *Cre/+* and *Nfya-CKO* mice. Horizontal lines represent the median. Whiskers represent maximum and minimum values. IQR represented by boxplots. ****$P < 0.0001$; two-sided Wilcoxon matched-pairs signed-ranked sum test. NFYA-bound *Cre/+* vs *Nfya-CKO* $P = 2.85 \times 10^{-70}$; NFYA-unbound *Cre/+* vs *Nfya-CKO* $P = 8.85 \times 10^{-32}$. (**D**) IGV tracks of NFYA occupancy (CUT&RUN) and scATAC-seq signal, at gene boundaries of genes that are direct targets of the STRA8/MEIOSIN axis.

