## [Peer Review File · The EMBO Journal]

Transcription factor NFYA directs male meiotic entry by regulating accessible chromatin at meiotic promoters in mice

Martin Säflund, Masomeh Askari, Atiyeh Eghbali, Mukhtar Abdi, Dilay Er, Ann-Kristin Östlund Farrants, Tianxiong Yu, and Deniz Ozata

Corresponding author(s): Deniz Ozata (deniz.ozata@su.se) , Tianxiong Yu (Tianxiong.Yu@umassmed.edu)

Review Timeline:

Submission Date:	29th Jun 25
Editorial Decision:	12th Sep 25
Revision Received:	8th Nov 25
Editorial Decision:	19th Jan 26
Revision Received:	30th Jan 26
Accepted:	25th Feb 26

Editor: Cornelius Schneider

Transaction Report:

Dear Prof. Ozata,

Thank you for submitting your manuscript for consideration by the EMBO Journal. It has now been seen by three referees whose comments are shown below.

Thank you also for sharing the preliminary point-by-point response. Given the referees' positive recommendations and your willingness to commit to major revisions, I would like to invite you to submit a revised version of the manuscript, addressing the comments of all three reviewers as detailed in the preliminary point-by-point response. I should add that it is EMBO Journal policy to allow only a single round of revision, and acceptance of your manuscript will therefore depend on the completeness of your responses in this revised version.

Thank you for the opportunity to consider your work for publication. I look forward to your revision.

Yours sincerely,

Cornelius Schneider, PhD
Editor
The EMBO Journal
c.schneider@embojournal.org

Please remember: Digital image enhancement is acceptable practice, as long as it accurately represents the original data and conforms to community standards. If a figure has been subjected to significant electronic manipulation, this must be noted in the

figure legend or in the 'Materials and Methods' section. The editors reserve the right to request original versions of figures and the original images that were used to assemble the figure.

We realize that it is difficult to revise to a specific deadline. In the interest of protecting the conceptual advance provided by the work, we recommend a revision within 3 months (11th Dec 2025). Please discuss the revision progress ahead of this time with the editor if you require more time to complete the revisions. Use the link below to submit your revision:

Referee #1:

The dynamic shift in the transcriptome plays a critical role in spermatogenesis, particularly at the onset of meiosis. Previous studies have shown that RNA polymerase II pauses at the transcription start sites of meiotic and spermiogenic genes prior to their activation. However, the factors responsible for establishing a permissive chromatin state at these loci remain largely unknown. In this study, the authors identify NFYA as a key regulator of timely gene activation during meiosis and spermiogenesis. NFYA functions as a pioneer factor in pre-meiotic germ cells, facilitating chromatin accessibility at the promoters of meiotic genes-including those regulated by the STRA8/MEIOSIN axis-and thereby establishing a coherent feedforward loop essential for the timely execution of the meiotic gene expression program. This model is supported by the observation that conditional germline deletion of *Nfya* in male mice results in a failure to enter meiosis. The findings convincingly demonstrate the essential role of NFYA in spermatogenesis and offer important insights into the molecular mechanisms governing germ cell development. The manuscript is well written, and the data are rich and thoroughly analyzed. Nevertheless, the current version of the manuscript would benefit from careful revision and clarification in several areas. Incorporating additional data that provide mechanistic insight and/or refining the presentation of key findings would substantially enhance the impact and clarity of the study.

Major Comments:

(1) The causal relationship between reduction in chromatin accessibility and decrease in Pol II pausing at target promoters in *Nfya*-cKO was unclear.

The study does not experimentally demonstrate that the observed reduction in chromatin accessibility in *Nfya*-cKO spermatogonia leads to a corresponding decrease in Pol II pausing at target promoters. The direct link between NFYA-mediated chromatin accessibility and Pol II pausing in spermatogonia remains unproven. The mechanistic model in Figure 6E, which explains the meiotic arrest phenotype, relies on the assumption that reduced accessibility impairs Pol II pausing. Without direct evidence, the conclusion that meiotic arrest is caused by a failure to establish paused Pol II remains speculative. Additional experiments are needed to directly assess the impact of NFYA knockout on Pol II pausing in spermatogonia. For example, CUT&Tag for total RNA Pol II and Ser5-phosphorylated RNA Pol II on FACS-purified spermatogonia from *Nfya*-cKO and control mice would provide rigorous validation.

(2) The relatively modest reduction in chromatin accessibility was observed upon NFYA loss (Fig6C). Could this subtle change alone account for the complete absence of meiotic cells in *Nfya*-cKO? Since the scRNA-seq data from *Nfya*-cKO testes only include spermatogonial cells, the impact of NFYA depletion on the transcriptome of meiotic and post-meiotic cells remains unclear. This limitation complicates the interpretation of NFYA's transcriptional regulatory role during later stages of germ cell development.

(3) The interpretation of NFYA's role in spermatogonial cells was unclear. The authors conclude that NFYA is dispensable in spermatogonia based on the observation that the numbers of SALL4- and c-Kit-positive cells per tubule are comparable between control and *Nfya*-cKO testes. However, given that *Nfya*-cKO males exhibit reduced testis size, the constant number of spermatogonia per tubule could suggest an apparent increase in spermatogonial density relative to overall cell number. Moreover, the scRNA-seq data presented in the study show that spermatogonial cells from *Nfya*-cKO cluster separately from wild-type spermatogonia (Fig4a), suggesting transcriptional alterations in the *Nfya*-cKO spermatogonia. These findings raise the possibility that NFYA also plays an important role in spermatogonial cells, which warrants further investigation. To address this issue, the following additional analyses are recommended:

Provide data showing the number of spermatogonia normalized to the number of Sertoli cells.

Carefully analyze gene expression changes within the spermatogonial population between *Nfya*-cKO and wild-type testes.

(4) Although the authors describe NFYA binding to genes involved in meiosis and spermiogenesis, they do not provide a broader characterization of NFYA target genes or how NFYA occupancy changes during the progression of spermatogenesis. This information would be valuable for understanding the functional scope of NFYA in germ cell differentiation. It would be nice if the authors include such analyses to enhance the mechanistic depth of the study.

Minor Comments:

Lines 215-219 and Fig. 2A: Please report other enriched motifs identified in the analysis and justify the decision to focus on NFYA for further investigation.

Lines 255-256: NFYA appears to be expressed in round spermatids based on Fig. 2C and Fig. S5B. Please state the expression level of NFYA in RS explicitly.

Lines 375-395: The authors state that "these results demonstrate that NFYA-deficient germ cells developed into preleptotene spermatocytes, but stalled there to progress through meiosis." This raises an additional question: do NFYA-deficient germ cells fail to properly transition from mitosis to meiosis and instead retain mitotic characteristics?

Figure numbering: To maintain consistency with the order of appearance in the main text, it would be preferable to switch the numbering of Fig. S6 and Fig. S7.

Referee #2:

The study aims to characterize chromatin poising during spermatogenesis and proposes that NFYA functions as a pioneer factor at these regulatory sites. The most compelling strength is the strong phenotype in *Nfya* conditional knockout (cKO) mice, which supports an essential role for NFYA in spermatogenesis.

However, I do not agree with the conclusion that NFYA acts as a pioneer factor. By definition, pioneer factors convert closed chromatin into open chromatin. In this study, the evidence for such a role is insufficient and the claim is overstated:

1. NFYA's impact on chromatin accessibility appears modest, and many NFYA-bound sites remain accessible even in its absence (Figure 5).
2. The promoters examined are already accessible throughout spermatogenesis, indicating that NFYA is unlikely to initiate accessibility.

These points suggest that NFYA may help maintain accessibility rather than establish it. This conceptual framework should be revised, and all descriptions of "pioneer" activity should be removed to avoid misleading interpretations.

Although the functional data from *Nfya*-cKO mice are promising, the manuscript requires substantial revision to address both technical limitations and conceptual oversights.

Major Points:

1. Many references are properly formatted. Please ensure they follow journal guidelines.
2. The term "Pol II peaks" is misleading-PRO-seq signal does not solely represent RNA polymerase II. Please clarify. The Results section is currently difficult to interpret.
3. The findings in Figures 1 and 2 relate closely to recent work from the Cairns lab (Yi et al., NSMB 2025). This paper should be cited, and its relevance discussed.
4. The claim that NFYA is a pioneer factor should be removed. Pioneer factors establish accessibility at previously closed chromatin. Here, the promoters remain open even without NFYA (Figure 5), suggesting a maintenance role instead.
5. Yi et al. (NSMB 2025) showed that many transcription factors bind poised promoters. NFYA may be one such factor, but it is unlikely to be unique; this broader context should be discussed.
6. NFYA CUT&RUN replicates are inconsistent (replicate 1 differs greatly from replicates 2 and 3). These experiments should be repeated to ensure reproducibility.
7. The section on NFYA binding sites contains excessive speculation. Claims should be supported with functional data, ideally from *Nfya*-cKO mice. For example, Line 329's statement on mitotic transcriptional programs requires supporting evidence.
8. Line 335 ("NFYA may act as a pioneer factor...") should be deleted.
9. Please clarify the identity of *Stra8*-Cre and include an appropriate citation.
10. The evaluation of meiotic entry is insufficient. Additional STRA8 and/or MEIOSIN immunostaining is needed to determine whether preleptotene spermatocytes are present in mutant testes.
11. In Figure 4, the mutant data differ markedly from controls, but the conclusions are unclear. Eight weeks is likely too late for meaningful analysis; an earlier time point (e.g., 11 days postpartum) would be more appropriate.
12. The analysis in Figure 6 is speculative and not well supported by data. This section should be substantially revised or removed.

We would like to thank the Reviewers' insightful and constructive comments which helped us to improve our work. We are also very gladdened to have the Reviewer's endorsement for the biological significance of our study in the field of gene regulatory events during spermatogenesis. As a young and small lab with limited budget, we have done our best to address each reviewer comment. Based on the Reviewers' comments, we conducted additional experiments and analyses, and revised our manuscript. All changes made in the manuscript are marked in yellow for the Reviewers' convenience.

During revising our study, Dilay Deren Er contributed to our work by performing some of the immunostaining experiment. Therefore, she is included in the author list.

Point-by-point responses to the Reviewers' critiques

Referee #1 (Report for Author)

The dynamic shift in the transcriptome plays a critical role in spermatogenesis, particularly at the onset of meiosis. Previous studies have shown that RNA polymerase II pauses at the transcription start sites of meiotic and spermiogenic genes prior to their activation. However, the factors responsible for establishing a permissive chromatin state at these loci remain largely unknown. In this study, the authors identify NFYA as a key regulator of timely gene activation during meiosis and spermiogenesis. NFYA functions as a pioneer factor in pre-meiotic germ cells, facilitating chromatin accessibility at the promoters of meiotic genes-including those regulated by the STRA8/MEIOSIN axis-and thereby establishing a coherent feedforward loop essential for the timely execution of the meiotic gene expression program. This model is supported by the observation that conditional germline deletion of Nfya in male mice results in a failure to enter meiosis.

The findings convincingly demonstrate the essential role of NFYA in spermatogenesis and offer important insights into the molecular mechanisms governing germ cell

development. The manuscript is well written, and the data are rich and thoroughly analyzed. Nevertheless, the current version of the manuscript would benefit from careful revision and clarification in several areas. Incorporating additional data that provide mechanistic insight and/or refining the presentation of key findings would substantially enhance the impact and clarity of the study.

Thank you. We truly appreciate the Reviewer's evaluation of the significance and the quality of our study. Such comments motivate graduate students of the lab who worked hard to establish biochemical, molecular, and high-throughput sequencing techniques including histology, FACS, chromosome spread, ATAC-seq, CUT&RUN, and sc-multiome.

Major Comments:

(1) The causal relationship between reduction in chromatin accessibility and decrease in Pol II pausing at target promoters in Nfya-cKO was unclear.

The study does not experimentally demonstrate that the observed reduction in chromatin accessibility in Nfya-cKO spermatogonia leads to a corresponding decrease in Pol II pausing at target promoters. The direct link between NFYA-mediated chromatin accessibility and Pol II pausing in spermatogonia remains unproven. The mechanistic model in Figure 6E, which explains the meiotic arrest phenotype, relies on the assumption that reduced accessibility impairs Pol II pausing. Without direct evidence, the conclusion that meiotic arrest is caused by a failure to establish paused Pol II remains speculative. Additional experiments are needed to directly assess the impact of NFYA knockout on Pol II pausing in spermatogonia. For example, CUT&Tag for total RNA Pol II and Ser5-phosphorylated RNA Pol II on FACS-purified spermatogonia from Nfya-cKO and control mice would provide rigorous validation.

We thank the Reviewer for the great suggestion. We performed CUT&Tag for Pol II Ser5P from FACS-purified SpG of *Nfya*-CKO and *Cre*/+ to test whether reduced chromatin accessibility impairs paused Pol II occupancy at poised promoters in NFYA-deficient SpG. We found that Pol II Ser5P occupancy around the TSSs of NFYA-bound poised genes was reduced notably in NFYA-deficient SpG when compared with NFYA-unbound poised genes and randomly chosen genes ($n = 5,000$). In our revised manuscript, we added our findings in a new Expanded View Figure (Fig. EV15) under the subsection of the Result section with the title "*Nfya* deletion impairs chromatin accessibility and paused Pol II occupancy at the promoters of poised genes in spermatogonia". (See below, text on page 21).

*"After the unphosphorylated Pol II is recruited to accessible promoters (Chen et al., 2018a), phosphorylation of the C-terminal domain of the largest subunit of Pol II at serine-5 (Pol II Ser5P) allows Pol II to escape pre-initiation complex and synthesizes 20–60 nt of RNAs, and finally Pol II Ser5P undergoes promoter-proximal pausing by the DRB Sensitivity-Inducing Factor (DSIF) and the Negative Elongation Factor (NELF) (Core et al., 2008; Kwak et al., 2013; Chen et al., 2018a; Vihervaara et al., 2023). We therefore performed CUT&Tag (Henikoff et al., 2020) for Pol II Ser5P from FACS-purified SpG of *Nfya*-CKO and *Cre*/+ mice to test whether reduced chromatin accessibility impairs paused Pol II occupancy at poised promoters in NFYA-deficient SpG. We performed three biological replicates from *Nfya*-CKO and *Cre*/+ (Fig. EV15A; agreement between replicates; SpG from *Nfya*-CKO Spearman's $\rho > 0.92$, SpG from *Cre*/+ Spearman's $\rho > 0.95$). We first quantified the density of Pol II Ser5P around the TSS relative to the Pol II Ser5P density at gene body for poised and non-poised meiosis-I and spermiogenesis genes, and mitosis genes in SpG from *Cre*/+ (Fig. EV15B). Both poised meiosis-I and spermiogenesis genes displayed higher Pol II Ser5P accumulation at promoters compared with mitosis genes in SpG from *Cre*/+ (Fig.*

EV15B). This finding corroborates with the temporal expression of these gene categories: mitosis genes are actively transcribed in SpG, whereas poised meiosis-I and spermiogenesis genes accumulate paused Pol II that is later released into transcriptional elongation in pachytene stage of meiosis-I (Figs. EV1A, EV2B, EV2C). We next quantified the normalized Pol II Ser5P coverage around the TSSs of NFYA-bound and -unbound poised genes in SpG of *Nfya*-CKO and *Cre*/⁺ (Figs. EV15C, EV15D). *E. coli* spike-in sequences in each sample enabled us to calculate normalized coverage for each sample. We found reduced Pol II Ser5P occupancy around the TSSs of NFYA-bound poised genes in SpG from *Nfya*-CKO compared to that from *Cre*/⁺ mice (Figs. EV15C, EV15D). We also observed reduction in Pol II Ser5P occupancy around the TSSs of NFYA-unbound poised genes between *Nfya*-CKO and *Cre*/⁺ mice. Yet, such reduction appeared to be less than the reduction we observed for NFYA-bound genes (Figs. EV15C, EV15D). We thus analyzed the changes in Pol II Ser5P occupancy within ± 0.5 kb of the TSSs of NFYA-bound and-unbound poised genes, and randomly chosen 5,000 control genes in SpG of *Nfya*-CKO compared to that of *Cre*/⁺ mice (Fig. EV15E). Intriguingly, we found larger reduction in Pol II Ser5P occupancy around the TSSs of NFYA-bound poised genes in NFYA-deficient SpG compared to that of NFYA-unbound poised genes (NFYA-bound poised genes, *Nfya*-CKO \div *Cre*/⁺ = 0.63; NFYA-unbound genes, *Nfya*-CKO \div *Cre*/⁺ = 0.77) (Fig. EV15E; Two-sided wilcoxon matched-pairs signed ranked sum test; *****p* < 0.0001). However, we note that the reduction in Pol II Ser5P occupancy for NFYA-unbound poised genes was comparable to that for randomly chosen control genes (*Nfya*-CKO \div *Cre*/⁺ = 0.76). When taken together, we conclude that prior to meiotic entry, NFYA regulates chromatin accessibility—thereby contributing to the accumulation of paused Pol II—at the promoters of genes that remain poised for activation later during meiosis.”

(2) *The relatively modest reduction in chromatin accessibility was observed upon NFYA loss (Fig6C). Could this subtle change alone account for the complete absence of meiotic cells in Nfya-cKO? Since the scRNA-seq data from Nfya-cKO testes only include spermatogonial cells, the impact of NFYA depletion on the transcriptome of meiotic and post-meiotic cells remains unclear. This limitation complicates the interpretation of NFYA's transcriptional regulatory role during later stages of germ cell development.*

We thank the Reviewer for the comment. To address the Reviewer's concern and to demonstrate that the impact of NFYA deletion on chromatin accessibility is indeed notable, we performed additional analyses, which we now added into our revised manuscript. In the original submission, we measured the change in chromatin accessibility at the promoters of NFYA-bound and -unbound genes between SpG cells from *Cre/+* and *Nfya-CKO* (Fig. 5C,D). Given that we report a strong correlation between NFYA occupancy and accessible chromatin at poised gene promoters in wild-type SpG cells (Fig. 2E), in our revised manuscript, we then further examined the change in chromatin accessibility at NFYA-bound promoters between *Nfya-CKO* and *Cre/+* relative to their NFYA occupancy. After rank ordering the NFYA-bound promoters into four quartiles based on their NFYA occupancy (New figure; Fig. EV14A), we found more pronounced reduction in accessible chromatin at promoters, which displayed the highest NFYA occupancy, than those promoters with the least NFYA occupancy (New figure; Figs. EV14B–C). We now included our new results and revised the corresponding text in our revised manuscript (text on page 20).

Moreover, we kindly refer to our response to the comment #3 (below). We also performed differential gene expression analysis between the SpG of *Nfya-CKO* and *Cre/+* mice (New figure; Fig. 6). While our analysis revealed deregulated genes related

to mitotic transcriptional program, we also found that the RNA abundance of many STRA8/MEIOSIN direct target genes were reduced in NFYA-deficient SpG (New figure; Fig. 6). Taken together, reduced chromatin accessibility at poised promoters and deregulated transcriptome signature in NFYA-deficient SpG likely suffice for spermatogenic arrest at meiotic entry. We hope that the Reviewer will find our revision reasonable.

(3) The interpretation of NFYA's role in spermatogonial cells was unclear. The authors conclude that NFYA is dispensable in spermatogonia based on the observation that the numbers of SALL4- and c-Kit-positive cells per tubule are comparable between control and Nfya-cKO testes. However, given that Nfya-cKO males exhibit reduced testis size, the constant number of spermatogonia per tubule could suggest an apparent increase in spermatogonial density relative to overall cell number. Moreover, the scRNA-seq data presented in the study show that spermatogonial cells from Nfya-cKO cluster separately from wild-type spermatogonia (Fig4a), suggesting transcriptional alterations in the Nfya-cKO spermatogonia. These findings raise the possibility that NFYA also plays an important role in spermatogonial cells, which warrants further investigation. To address this issue, the following additional analyses are recommended:

We thank the Reviewer for the great suggestion.

Provide data showing the number of spermatogonia normalized to the number of Sertoli cells.

We thank the Reviewer for the fantastic suggestion. We now performed triple staining for SALL4, c-Kit, and GATA4 on the testis sections from *Nfya-CKO* and *Cre/+* mice (New figure; Fig. 3G). We note that in the revised manuscript, we quantified the number of SOX9-positive Sertoli cells in *Nfya-CKO* and *Cre/+* males (Fig. EV10F): we found comparable number of Sertoli cells between *Nfya-CKO* and *Cre/+* males confirming the

germline deletion of *Nfya*. Moreover, this finding grants Sertoli cells as a good normalization factor as suggested by the Reviewer. However, anti-SOX9 antibody did not work in our triple staining. We therefore instead performed triple staining using anti-GATA4 antibody along with anti-SALL4 and anti-c-Kit antibodies (New figure; Fig. 3G). GATA4 is an established Sertoli cell marker (Kyrönlahti et al., *Mol Cell Endocrinol*, 2011). After normalizing the number of undifferentiated SpG or differentiating SpG to that of Sertoli cells, our results revealed that while the maintenance of undifferentiated SpG occurs normally within *Nfya*-CKO testis, deletion of *Nfya* results in compromised differentiation of SpG (Fig. 3G). In our revised manuscript, we added our findings in a new subsection of the Result section with the title “*Nfya* deletion impairs spermatogonial differentiation” (See below, text on page 16).

***“Nfya* deletion impairs spermatogonial differentiation**

NFYA is expressed abundantly in SpG and pL (Figs. 2 and EV6) and the promoters of ~one-third of genes required for mitotic stage are bound by NFYA (Fig. 2F). Yet, spermatogenesis arrests at the pL stage in *Nfya*-CKO testis. We thus examined whether spermatogonial maintenance and differentiation occur normally in *Nfya*-CKO males. We performed triple staining for SALL4, c-Kit, and GATA4 on the testis sections from *Nfya*-CKO and *Cre*/⁺ mice. SALL4 is a marker for undifferentiated SpG (Hobbs et al., 2012), while c-Kit is for differentiating SpG (Yoshinaga et al., 1991). GATA4 is essential for the function of Sertoli cells (Kyrönlahti et al., 2011). Given that Sertoli cells were unaffected in *Nfya*-CKO compared to *Cre*/⁺ males (Fig. EV10F), we normalized the number of SALL4-positive undifferentiated SpG or c-Kit-positive differentiating SpG to the number of GATA4-positive Sertoli cells. We found a comparable number of SALL4-positive undifferentiated SpG normalized to Sertoli cells between *Nfya*-CKO and *Cre*/⁺ mice, whereas the number of c-Kit-positive differentiating SpG was reduced ~threefold in *Nfya*-CKO compared to *Cre*/⁺ (Fig. 3G). These findings suggests that

while the maintenance of undifferentiated SpG occurs normally within Nfya-CKO testis, deletion of Nfya results in compromised differentiation of SpG.”

Carefully analyze gene expression changes within the spermatogonial population between Nfya-cKO and wild-type testes.

We thank the Reviewer for the fantastic suggestion. We now performed differential gene expression analysis between SpG of *Nfya-CKO* and *Cre/+* using our sc-RNA-seq data. In the revised manuscript, we present our findings in a new subsection of the Result section with the title “*NFYA participates in the regulation of mitotic transcriptional program in spermatogonia*” along with the incorporation of a new figure (New figure; Fig. 6). Our analysis revealed that <5% of all genes were deregulated in SpG from *Nfya-CKO* compared to that from *Cre/+* (Fig. 6A), which likely accounts for moderate perturbation we observed in the differentiation of SpG in *Nfya-CKO* (Fig. 3G). Interestingly, we found that among the genes, whose promoters are bound by NFYA and whose transcript abundance is reduced in NFYA-deficient SpG, gene ontology related to the regulation of transcription was enriched, suggesting that many other NFYA-unbound deregulated genes are likely regulated by transcription factors that are themselves under NFYA control. In our study, we found that NFYA regulates chromatin accessibility at the promoters of *Stra8* and *Meiosin* genes, as well as the promoters of STRA8/MEIOSIN target genes (Fig. EV16). Unlike poised meiosis-I and spermiogenesis genes whose transcription burst at pachytene stage of meiosis-I, the expression of STRA8/MEIOSIN target genes starts in differentiating SpG and peaks in L/Z/eP spermatocytes consistent with their role in meiotic entry (Fig. EV16B). Intriguingly, our differential gene expression analysis revealed the RNA abundance of many STRA8/MEIOSIN target genes, whose promoters are bound by NFYA, were reduced in NFYA-deficient SpG partly accounting for the observation that spermatogenesis arrests

at meiotic entry in *Nfya*-CKO males. See the text from the revised manuscript below (text on page 23).

“NFYA participates in the regulation of mitotic transcriptional program in spermatogonia

*NFYA regulates chromatin accessibility at the promoters of poised meiosis-I and spermiogenesis genes in SpG, prior to the onset of their transcription during prophase I of meiosis (Figs. 1, 2, and 5). However, unlike poised genes, transcription of mitotic genes is switched off after entry into meiosis I (Figs. EV1A, EV2B, EV2C). We found that NFYA binds ~one-third of the mitotic gene promoters in SpG (Fig. 2F). Deletion of Nfya results in compromised differentiation of SpG (Fig. 3G). Moreover, scRNA-seq analysis revealed that NFYA-deficient SpG cluster separately from control SpG in UMAP space (Fig. 4A). These suggest a regulatory function for NFYA in mitotic transcriptional program beyond its role in regulating accessible chromatin at poised meiotic promoters. We found that accessible chromatin at mitotic promoters bound by NFYA was decreased in NFYA-deficient SpG ($Nfya\text{-CKO} \div Cre/+ = 0.75$), whereas chromatin accessibility at the promoters of NFYA-unbound mitotic genes remained nearly unchanged ($Nfya\text{-CKO} \div Cre/+ = 0.96$) (Fig. 5D; Two-sided wilcoxon matched-pairs signed ranked sum test; **** $p < 0.0001$). Using scRNA-seq, we next measured the change in the steady-state RNA abundance between SpG from *Nfya*-CKO and *Cre/+* (Fig. 6A). The RNA abundance of 1,241 genes—that account for <5% of all genes—were significantly deregulated in SpG from *Nfya*-CKO compared with that from *Cre/+* (Fig. 6A; $Nfya\text{-CKO} \div Cre/+ \leq 0.5$ or ≥ 2 ; $FDR < 0.05$). Deregulation of small fraction of all genes likely explains mild perturbation we observed in the differentiation of SpG in *Nfya*-CKO males (Fig. 3G). Of the 1,241 deregulated genes, steady-state RNA abundance of 629 was higher in NFYA-deficient SpG, while the RNA products of 612 genes were reduced (Fig. 6A). Our CUT&RUN analysis of NFYA binding from wild-type*

SpG revealed that NFYA bound near the TSSs for 201 of 1,241 genes whose transcripts were deregulated in NFYA-deficient SpG (Fig. 6B). Of the 201 NFYA-bound deregulated genes, RNA abundance of 85 were reduced in NFYA-deficient SpG (Fig. 6B). Interestingly, among these 85 genes, 11 belonged to mitotic gene category that were enriched for GO related to the regulation of transcription suggesting that many other NFYA-unbound deregulated genes are likely regulated by transcription factors that are themselves under NFYA control (Figs. 6C and 6D; Table EV5). Indeed, these 11 genes included *Foxo1* and *Sox3*, which encode transcription factors that participate in the regulation of mitotic genetic program (McAninch et al., 2020; Shen et al., 2022; Yi et al., 2025).

Finally, among the 612 genes whose RNA abundance were reduced in NFYA-deficient SpG, 43 were direct STRA8/MEIOSIN targets that are required for meiotic entry and progression (Kojima et al., 2019; Ishiguro et al., 2020) (Fig. 6A). Our NFYA CUT&RUN analysis from wild-type SpG revealed that NFYA bound near the TSSs for 22 of 43 STRA8/MEIOSIN target genes, whose transcripts were reduced in NFYA-deficient SpG, including *Meioc*, *Hspa5*, *Ybx*, *Pabpc1*, and *Pcbp3* (Fig. 6E; Table EV5). MEIOC regulates the transition from mitosis to meiosis-I (Abby et al., 2016; Soh et al., 2017). HSPA5 is required for the maintenance of differentiating SpG (Wen et al., 2023). RNA-binding proteins, YBX2, PABPC1, and PCBP3, regulate the translation of mRNAs required later during spermatogenesis (Kimura et al., 2009; Chapman et al., 2013; He et al., 2019). Together, these findings are consistent with our observation that spermatogenesis arrests at meiotic entry in *Nfya*-CKO males.”

(4) Although the authors describe NFYA binding to genes involved in meiosis and spermiogenesis, they do not provide a broader characterization of NFYA target genes or how NFYA occupancy changes during the progression of spermatogenesis. This information would be valuable for understanding the functional scope of NFYA in germ

cell differentiation. It would be nice if the authors include such analyses to enhance the mechanistic depth of the study.

We thank the Reviewer for the comment. In the first paragraph of the section '*NFYA binds the promoters of poised genes*', we indeed first analyzed genome-wide NFYA binding in all three cell stages, SpG, L/Z, and P/D (Fig. EV8A). We next examined how NFYA occupancy changes throughout these stages because NFYA is highly enriched before meiosis (Figs. EV8B and EV8C). We now revised the paragraph to make our findings more accessible to the readership (See below, text on page 11).

“In SpG, 70% of all NFYA peaks were within the ± 2 kb of annotated TSSs, whereas only 2.2% of genome-wide NFYA peaks resided in enhancer regions marked by histone modifications, H3K4me1 (Fig. EV8A). Similar observations were obtained in L/Z and P/D suggesting that NFYA primarily functions at the promoter-proximal regions of genes (Fig. EV8A). Given that NFYA is expressed abundantly in pre-meiotic cells and that its expression reduces after meiotic entry (Figs. 2C, EV6C, EV6D), we next examined the change in NFYA occupancy throughout pre-meiotic and meiotic stages. We observed that NFYA occupancy at TSSs was higher in pre-meiotic SpG than in L/Z and P/D (Fig. EV8B). Consistently, our genome-wide analysis identified 4,020 protein-coding and 360 non-coding genes with an NFYA peak within ± 2 kb of their TSSs in at least two replicates of SpG, whereas we found lesser number of genes bound by NFYA in L/Z and P/D (Fig. EV8C; Table EV3)”

Next, in the second paragraph of the section '*NFYA binds the promoters of poised genes*', we examined NFYA occupancy around TSSs of poised genes throughout three cells stages and found that the promoter-proximal regions of poised genes accumulate stronger NFYA binding, which strongly correlates with the degree of accessible chromatin, in SpG cells than in L/Z and P/D cells (Figs. 2E and EV9).

Finally, in the last paragraph of the section '*NFYA binds the promoters of poised genes*', we analyzed which of the genes from each gene category are unambiguously bound by NFYA (Fig. 2F; genes that are expressed in SpG cells, *mitotic genes*; genes that are expressed during meiosis-I, *meiosis-I genes*, and genes related to spermiogenesis, *spermiogenesis genes*). The genes that are unambiguously bound by NFYA is described in the manuscript text See below, text on page 13).

“To determine the number of unambiguous NFYA-bound genes for each gene category, we computed the distance from the TSSs of poised and non-poised genes to the nearest NFYA peak in three biological replicates. Those genes with NFYA peak within the ± 2 kb of their TSSs in at least two replicates were considered NFYA-bound genes.”

We hope that the Reviewer will find our revision reasonable.

Minor Comments:

Lines 215-219 and Fig. 2A: Please report other enriched motifs identified in the analysis and justify the decision to focus on NFYA for further investigation.

We thank the Reviewer for the suggestion. We now report other top-five scoring motifs that are beneath the ATAC-seq peaks around the promoters of poised genes in SpG cells (Fig. EV5). The motivation as to why we focused on NFYA among other top-five scoring motifs is that accumulated line of biochemical, genetic, and molecular evidence from other cellular contexts has proposed the regulatory role for NFYA in chromatin accessibility, mostly as a 'pioneer factor'. However, upon the Reviewer #2's comments, we discussed potential 'pioneer role' of NFYA during spermatogenesis only in discussion section (See discussion section, text on page 25). Nevertheless, we revised the related section according to the Reviewer's suggestion (See below, text on page 9).

“Because accumulated line of evidence has revealed the regulatory role of NFYA in accessible chromatin in other cellular contexts (Coustry et al., 2001; Fleming et al., 2013; Nardini et al., 2013; Oldfield et al., 2014; Sherwood et al., 2014; Oldfield et al., 2019), we sought to examine the molecular function of NFYA during spermatogenesis. Notably our analysis identified other motifs that are bound by KLF1/3/5, SP1/2, DMRT1/6, and ELK3/4 (Fig. EV5). Resolving the possible regulatory role of these TFs in chromatin accessibility and their relation with NFYA during spermatogenesis will be of a promising research direction.”

Lines 255-256: NFYA appears to be expressed in round spermatids based on Fig. 2C and Fig. S5B. Please state the expression level of NFYA in RS explicitly.

We thank the Reviewer for the suggestion. We now stated that we observed moderate to strong nuclear NFYA signal in RS cells (Fig. 2C). See the revised manuscript text below (text on page 10).

“Notably, even though we observed reduced expression of NFYA in primary meiocytes, RS showed moderate to strong nuclear NFYA in the stages II, III, and VI tubules (Fig. 2C). Consistently, immunofluorescence staining revealed that those germ cells residing near basal membrane—intermediate and type B SpG, as well as pL—had the highest NFYA signal in their nuclei when compared to those localized more towards the lumen—L, Z, P, and D (Fig. EV6C). Moreover, immunostaining of NFYA in the testes sections from staged mice corroborated these findings: we detected a strong NFYA signal localized in the nuclei of pre-meiotic cells and spermatids, whilst the signal from meiocytes was either low or below the limit of detection (Fig. EV6D).”

Lines 375-395: The authors state that “these results demonstrate that NFYA-deficient germ cells developed into preleptotene spermatocytes, but stalled there to progress

through meiosis." This raises an additional question: do NFYA-deficient germ cells fail to properly transition from mitosis to meiosis and instead retain mitotic characteristics?

We thank the Reviewer for the fantastic question. In order to answer the Reviewer's question, we performed immunofluorescence staining for H3S10P—a marker for mitotic chromosome condensation (Hendzel et al., *Chromosoma*, 1997)—along with γ H2AX on the testis sections from *Cre/+* and *Nfya-CKO* males. We found that spermatocytes resembling preleptotene cells reveal precocious mitotic status (Fig. 3F). We now added our findings in Fig. 3F. See the text from the revised manuscript below (text on page 16).

*"Both *Stra8* KO and *Meiosin* KO spermatocytes revealed precocious mitotic status (Mark et al., 2008; Ishiguro et al., 2020). To test whether NFYA-deficient spermatocytes resembling preleptotene cells underwent mitotic chromosome condensation, we performed double staining for H3S10P—H3Ser10 phosphorylation, a marker for mitotic chromosome condensation (Hendzel et al., 1997)—and γ H2AX. Notably, some γ H2AX-positive spermatocytes from *Nfya-CKO* testis revealed H3S10P staining, whereas strong H3S10P signal on centromeres was confined only in spermatocytes at Metaphase I in *Cre/+* mice, suggesting that NFYA-deficient spermatocytes resembling pL fail to properly transit from mitosis to meiosis-I, but rather retain mitotic characteristic (Fig. 3F). Together, we conclude that NFYA-deficient germ cells developed into pL, but stalled there to progress through meiosis."*

Figure numbering: To maintain consistency with the order of appearance in the main text, it would be preferable to switch the numbering of Fig. S6 and Fig. S7.

We think that the order of Figs. S6 (now Figs. EV8 and EV9) and S7 (now Fig. EV10) should remain unchanged. Because Figs. EV8 & EV9 reveal the characterization of genome-wide NFYA occupancy across wild-type FACS-purified germ cells, while Fig.

EV10 shows the strategy of male germline-specific deletion of *Nfya* and the validation of the deletion.

Referee #2 (Report for Author)

The study aims to characterize chromatin poisoning during spermatogenesis and proposes that NFYA functions as a pioneer factor at these regulatory sites. The most compelling strength is the strong phenotype in Nfya conditional knockout (cKO) mice, which supports an essential role for NFYA in spermatogenesis.

We truly thank the Reviewer for the evaluation of the significance of our study. Such comments motivate graduate students of the lab who worked hard to establish biochemical, molecular, and high-throughput sequencing techniques including histology, FACS, chromosome spread, ATAC-seq, CUT&RUN, and sc-multiome.

However, I do not agree with the conclusion that NFYA acts as a pioneer factor. By definition, pioneer factors convert closed chromatin into open chromatin. In this study, the evidence for such a role is insufficient and the claim is overstated:

The Reviewer raises a valid point and we appreciate the perspective of the Reviewer.

Even though the central notion is that pioneer factors bind closed chromatin, they can also bind to sites that are primed by prior pioneer factors or accessible chromatin features (Cernilogar et al., *NAR*, 2019). Moreover, accumulated line of biochemical, genetic, and molecular studies in other cellular contexts have proposed NFYA as a 'pioneer factor': (i) NFYA can bind to its DNA motif in polycomb-repressed chromatin (Fleming JD et al., *Genome Res*, 2013); (ii) NFYA binding can induce 80° bend in DNA thereby promoting chromatin accessibility and binding of other TFs (Nardini et al., *Cell*, 2013; Oldfield et al., *Mol. Cell*, 2014); (iii) computational modeling that analyzes

genome-wide DNase I hypersensitivity profiles classified NFYA among the strongest *pioneer activity indices* (Sherwood *et al.*, *Nature Biotechnology*, 2014).

Given the NFYA's maintenance role in chromatin accessibility, the notable reduction in chromatin accessibility in the absence of NFYA, and the established literature suggesting NFYA as a 'pioneer factor', we toned down throughout manuscript and discussed the potential 'pioneer role' of NFYA only in the discussion section (See below, text on page 25). We think that it is important to discuss potential pioneer role of NFYA during spermatogenesis in the light of our findings and current literature in the discussion section because it may provide perspective and insight for further studies. We respectfully hope that the Reviewer will agree with us.

*“During step-wise developmental processes, pioneer factors can elicit chromatin opening at gene regulatory sequences to prime following developmental events as opposed to immediate induction of gene expression (Bevington *et al.*, 2016; Zaret, 2020; Yi *et al.*, 2025). Consistently, we reveal that*

*in mitotic stage, NFYA regulates the chromatin accessibility at the promoters of genes that are indeed transcribed during meiosis-I. Our findings thus raise a possibility that NFYA may function as a pioneer factor in pre-meiotic SpG. Supporting this view, accumulated line of biochemical, genetic, and molecular evidence has proposed NFYA as a 'pioneer factor' in other cellular contexts (Coustry *et al.*, 2001; Fleming *et al.*, 2013; Nardini *et al.*, 2013; Oldfield *et al.*, 2014; Sherwood *et al.*, 2014; Oldfield *et al.*, 2019). NFYA can bind to its DNA motif in polycomb-repressed chromatin (Fleming *et al.*, 2013). In mouse ESCs, NFYA promotes chromatin accessibility for the binding of master TFs (Oldfield *et al.*, 2014). In fact, the model proposes that NFYA binding induces ~80° bend in DNA, ultimately resulting in nucleosome repositioning (Oldfield *et al.*, 2014). NFYA forms a complex with NFYB and NFYC (Dolfini *et al.*, 2009). The crystal structure of NFY complex bound to its DNA motif demonstrated that NFY*

complex induces a α helix into the DNA minor groove, thereby establishing permissive chromatin modifications at NFYA-bound promoters (Nardini et al., 2013). Moreover, computational modeling that analyzes genome-wide DNase I hypersensitivity profiles classified NFYA among the strongest pioneer activity indices (Sherwood et al., 2014).

An *in vitro* study demonstrated that NFYA displaces nucleosomes (Cousty et al., 2001). One of the hallmarks of pioneer factors is that they recruit nucleosome-remodeling complexes to promote nucleosome reorganization (Barral and Zaret, 2024). We therefore cannot rule out the possibility that NFYA recruits a chromatin remodeler to regulate permissive chromatin during spermatogenesis. This will be an important subject of investigation in our future studies.

Recent study demonstrated that TFs, ZBTB16, SALL4, and SOX3—which are expressed in undifferentiated SpG (Hobbs et al., 2012; McAninch et al., 2020)—, cooperatively establish accessible chromatin at the promoters of meiotic genes in juvenile SpG (Yi et al., 2025). The promoters bound by ZBTB16, SALL4, and SOX3 retain open chromatin features, such as H3K4me3, H3K27Ac, and hypomethylation (Yi et al., 2025). Although widely accepted notion is that pioneer factors bind closed chromatin (Barral and Zaret, 2024), they can also bind to sites that are primed by accessible chromatin features or prior pioneer factors (Cernilogar et al., 2019). Given that NFYA protein abundance is lower in undifferentiated type A SpG than differentiating SpG (Fig. 2C), cooperative activity of ZBTB16, SALL4, and SOX3 in undifferentiated SpG may prime the binding of NFYA at meiotic gene promoters in differentiating SpG. Congruently, we observed that ~one-third of the promoters bound by ZBTB16, SALL4, and SOX3 were occupied by NFYA in differentiating SpG (Table EV6; 5,519 of the 16,151 promoters) (Yi et al., 2025). Together with ours and recent studies, it may be possible that sequential and collective activity of factors conducts establishment and maintenance of accessible chromatin at meiotic promoters, thereby promoting orderly entry and progression of meiosis-I. Therefore, investigating the interplay between NFYA

and other key factors regulating open chromatin before entry into meiosis-I will be of great interest to study gene regulatory events of spermatogenesis.”

1. NFYA's impact on chromatin accessibility appears modest, and many NFYA-bound sites remain accessible even in its absence (Figure 5).

We thank the Reviewer for the comment. To address the Reviewer's concern and to demonstrate that the impact of NFYA deletion on chromatin accessibility is indeed notable, we performed additional analyses, which we now added into our revised manuscript. In the original submission, we measured the change in chromatin accessibility at the promoters of NFYA-bound and -unbound genes between SpG cells from *Cre/+* and *Nfya-CKO* (Fig. 5C,D). Given that we report a strong correlation between NFYA occupancy and accessible chromatin at poised gene promoters in wild-type SpG cells (Fig. 2E), in our revised manuscript, we then further examined the change in chromatin accessibility at NFYA-bound promoters between *Nfya-CKO* and *Cre/+* relative to their NFYA occupancy. After rank ordering the NFYA-bound promoters into four quartiles based on their NFYA occupancy (New figure; Fig. EV14A), we found more pronounced reduction in accessible chromatin at promoters, which displayed the highest NFYA occupancy, than those promoters with the least NFYA occupancy (New figure; Figs. EV14B–C).

We now included our new results and revised the corresponding text in our revised manuscript (See below, text on page 20)

“Prior to the initiation of meiosis, in pre-meiotic cells, NFYA occupies the promoters of genes that remain poised for activation later during meiosis. To directly test the idea that NFYA regulates chromatin accessibility in pre-meiotic cells, we examined whether Nfya deletion impairs chromatin accessibility at meiotic promoters in SpG. We analyzed changes in chromatin accessibility around the promoters of NFYA-bound and -unbound gene categories in undifferentiated SpG from Cre/+ and Nfya-CKO

mice (Fig. 5C). Remarkably, we found that relative scATAC-seq signal at the promoters of NFYA-bound poised meiosis-I and spermiogenesis genes were reduced in NFYA-deficient undifferentiated SpG compared to Cre/+ mice (Fig. 5C; Table EV4D). In contrast, ATAC-seq signal at the promoters of NFYA-unbound poised genes remained almost unchanged between undifferentiated SpG from Cre/+ and Nfya-CKO mice (Fig. 5C; Table EV4D). In fact, chromatin accessibility at the promoters of NFYA-bound poised genes was significantly reduced in undifferentiated SpG from Nfya-CKO (Nfya-CKO ÷ Cre/+ = ~0.75) (Fig. 5D; Two-sided wilcoxon matched-pairs signed ranked sum test; ****p < 0.0001), whereas we observed a minor change in chromatin accessibility at the promoters of NFYA-unbound poised genes (Nfya-CKO ÷ Cre/+ = ~0.93). We found a strong correlation between NFYA occupancy and the degree of chromatin accessibility at poised promoters in wild-type SpG (Fig. 2E). We thus further examined the change in chromatin accessibility at NFYA-bound promoters between Nfya-CKO and Cre/+ relative to their NFYA occupancy. After rank ordering the NFYA-bound promoters into four quartiles based on their NFYA occupancy (Fig. EV14A), we found more pronounced reduction in accessible chromatin at promoters, which displayed the highest NFYA occupancy, than those promoters with the least NFYA occupancy (Fig. EV14B–D; fourth quartile, Nfya-CKO ÷ Cre/+ = 0.66-0.69; first quartile, Nfya-CKO ÷ Cre/+ = 0.81-0.85; Two-sided wilcoxon matched-pairs signed ranked sum test; ****p < 0.0001).”

2. The promoters examined are already accessible throughout spermatogenesis, indicating that NFYA is unlikely to initiate accessibility.

We thank the Reviewer for the perspective. Given the NFYA’s maintenance role in chromatin accessibility, the notable reduction in chromatin accessibility in the absence of NFYA, and the established literature suggesting NFYA as a pioneer factor, we toned

down throughout manuscript and discussed the potential 'pioneer role' of NFYA only in the discussion section (text on page 25).

Moreover, we note that we replaced the terms 'initiate' or 'facilitate' or 'establish' throughout manuscript with 'regulate' instead. Note that our manuscript title now became "Transcription factor NFYA directs male meiotic entry by regulating accessible chromatin at meiotic promoters in mice" instead of "Transcription factor NFYA directs male meiotic entry by facilitating accessible chromatin at meiotic promoters in mice"

These points suggest that NFYA may help maintain accessibility rather than establish it. This conceptual framework should be revised, and all descriptions of "pioneer" activity should be removed to avoid misleading interpretations.

Although the functional data from Nfya-cKO mice are promising, the manuscript requires substantial revision to address both technical limitations and conceptual oversights.

We kindly refer to our responses above and point-by-point responses for the related comments below.

Major Points:

1. *Many references are properly formatted. Please ensure they follow journal guidelines.*

We now followed the EMBO Journal's guideline for references and formatted our citations accordingly.

2. The term "Pol II peaks" is misleading-

Promoter-proximal regions of genes accumulate peaks of engaged RNA polymerase II (Pol II) representing paused Pol II. Supporting this view, we here provide the following statement from the highly cited review article by Core L. and Adelman K. (Core L. and Adelman K., *Genes Dev*, 2019): "Upon the development of global run-on sequencing

(GRO-seq), it became clear that the peaks of Pol II near promoters represented elongation complexes that were paused but competent to resume transcription elongation”.

Thus, in our work, ‘Pol II peaks’ refers to engaged Pol II peaks within the ± 2 kb of the transcription start sites (TSSs) of genes we analyzed. See the text below from the related result section (text on page 6)

“We thus first assessed whether the TSSs of meiosis-I and spermiogenesis genes retain significant Pol II peaks in SpG cells prior to their expression in Spl or RS cells. Using MACS3 (FDR < 0.01), we found that ~79% of meiosis-I genes exhibited significant Pol II peaks within the ± 2 kb of their TSSs in SpG cells (2,415 of 3,062 genes; median distance from the TSS to nearest Pol II peak = 158 bp) (Fig. EV2A; Table EV1B). Likewise, the promoters of ~63% of spermiogenesis genes retained Pol II peak in SpG cells (1,488 of 2,345 genes; median distance from the TSS to nearest Pol II peak = 166 bp) (Fig. EV2A; Table EV1B).”

PRO-seq signal does not solely represent RNA polymerase II. Please clarify. The Results section is currently difficult to interpret.

We here provide the statement from the abstract of the original paper by Kwak et al. (Kwak H. et al., *Science*, 2014): *“We developed a Precision nuclear Run-On and sequencing assay (PRO-seq) to map the genome-wide distribution of transcriptionally-engaged Pol II at base-pair resolution.”* Nevertheless, the Reviewer perhaps refers to that PRO-seq measures the activity of all three engaged RNA polymerases (Pol I, Pol II, and Pol III) via nuclear run-on reaction. However, the gene sets we work in our study are Pol II-driven genes. We thus do not know how to address Reviewer’s comment to complete satisfaction.

3. *The findings in Figures 1 and 2 relate closely to recent work from the Cairns lab (Yi et al., NSMB 2025). This paper should be cited, and its relevance discussed.*

We thank the Reviewer for the great suggestion. We now cited the work from Yi et al. (Yi et al., Nat Struct Mol Biol, 2025) and discussed the relevance of its findings with our study in the discussion section (See below, text on page 27).

“Recent study demonstrated that TFs, ZBTB16, SALL4, and SOX3—which are expressed in undifferentiated SpG (Hobbs et al., 2012; McAninch et al., 2020)—, cooperatively establish accessible chromatin at the promoters of meiotic genes in juvenile SpG (Yi et al., 2025). The promoters bound by ZBTB16, SALL4, and SOX3 retain open chromatin features, such as H3K4me3, H3K27Ac, and hypomethylation (Yi et al., 2025). Although widely accepted notion is that pioneer factors bind closed chromatin (Barral and Zaret, 2024), they can also bind to sites that are primed by accessible chromatin features or prior pioneer factors (Cernilogar et al., 2019). Given that NFYA protein abundance is lower in undifferentiated type A SpG than differentiating SpG (Fig. 2C), cooperative activity of ZBTB16, SALL4, and SOX3 in undifferentiated SpG may prime the binding of NFYA at meiotic gene promoters in differentiating SpG. Congruently, we observed that ~one-third of the promoters bound by ZBTB16, SALL4, and SOX3 were occupied by NFYA in differentiating SpG (Table EV6; 5,519 of the 16,151 promoters) (Yi et al., 2025). Together with ours and recent studies, it may be possible that sequential and collective activity of factors conducts establishment and maintenance of accessible chromatin at meiotic promoters, thereby promoting orderly entry and progression of meiosis-I. Therefore, investigating the interplay between NFYA

and other key factors regulating open chromatin before entry into meiosis-I will be of great interest to biologists studying gene regulatory events of spermatogenesis.”

4. The claim that NFYA is a pioneer factor should be removed. Pioneer factors establish accessibility at previously closed chromatin. Here, the promoters remain open even without NFYA (Figure 5), suggesting a maintenance role instead.

We thank the Reviewer for the perspective. Given the NFYA's maintenance role in chromatin accessibility, the notable reduction in chromatin accessibility in the absence of NFYA, and the established literature suggesting NFYA as a pioneer factor, we toned down throughout manuscript and discussed the potential 'pioneer role' of NFYA only in the discussion section (text on page 25).

5. Yi et al. (NSMB 2025) showed that many transcription factors bind poised promoters. NFYA may be one such factor, but it is unlikely to be unique; this broader context should be discussed.

We thank the Reviewer for this great suggestion. We kindly refer to our response to the comment #3.

6. NFYA CUT&RUN replicates are inconsistent (replicate 1 differs greatly from replicates 2 and 3). These experiments should be repeated to ensure reproducibility.

In the original submission, throughout manuscript, we assessed the agreement between the replicates of ATAC-seq or CUT&RUN by Spearman's correlation (ρ) using genome-wide read counts. However, we thank the Reviewer for the comment which led us to examine how our both ATAC-seq and CUT&RUN data are distributed. To understand how our data is distributed, we first calculated the log₂-transformed read counts within ± 2 kb of their transcription start sites (TSSs) of all genes in each replicate for ATAC-seq or CUT&RUN as in our previous publication (Yu T et al., *Nat Commun*, 2021). We then

calculated the frequency of ATAC-seq or CUT&RUN signal at the promoter of each gene (**Fig. R1**; below)

Figure R1. Distribution of data from each replicate of **(A)** ATAC-seq or **(B)** NFYA CUT&RUN. The log₂-transformed read counts ± 2 kb of TSSs of all genes in each replicate for ATAC-seq or NFYA CUT&RUN is calculated.

We in fact observed that while ATAC-seq replicates non-normally distribute, the replicates of NFYA CUT&RUN tend to be normally distributed (**Fig. R1**). Spearman's correlation (ρ) measures the relation between two data sets that are non-normally distributed, while Pearson correlation (r) measures the relation between two normally distributed data sets. Hence, in the revised manuscript, we performed Spearman's correlation (ρ) for the agreement of ATAC-seq replicates using the log₂-transformed read counts within ± 2 kb of their transcription start sites (TSSs) of all genes (Fig. EV4B), whereas we performed Pearson correlation (r) to measure the agreement between the replicates of NFYA CUT&RUN using the log₂-transformed read counts within ± 2 kb of their transcription start sites (TSSs) of all genes. Note that similar approach was applied in one of the studies from Steven Henikoff's lab who invented CUT&RUN technique (Janssens D et al., *Methodology*, 2018). Upon the Reviewer's comment, we now provide a new Expanded View Figure showing the agreement between NFYA

CUT&RUN replicates (New figure; Fig. EV7A). Moreover, we present few exemplified genes with NFYA peaks around their promoters throughout spermatogenesis which will allow the readership to observe the peak structure and intensity in each replicate (Fig. EV7B). After performing new analysis, we now found strong correlation between the replicates of NFYA CUT&RUN. We revised the corresponding text in the manuscript (See below, text on page 11)

“We sequenced three biological replicates for each cell population (Fig. EV7; agreement between replicates; Pearson correlation coefficient (r); SpG $r > 0.95$, L/Z Spearman’s $r > 0.93$, P/D $r > 0.89$).”

We also like to highlight that unlike many other studies, we did not merge the replicates and called the significant genome-wide peaks from the merged data. In fact, as reported in *Result* and *Methods* sections, we separately analyzed our independent replicates as we and others previously published (Cechini K. et al., *Reproduction*, 2023; Yu T. et al., *RNA*, 2023; Kojima M. et al., *eLife*, 2019).

Method section (text on page 52): *“NFYA peaks were identified using MACS3 (v3.0.2 (Zhang et al., 2008); FDR < 0.1). CUT&RUN for IgG antibody was used as the background control to call significant NFYA peaks. Genes with NFYA peak within ± 2 kb of their transcription start sites in at least two replicates of CUT&RUN experiments were considered as NFYA-bound genes.”*

Result section (text on page 13): *“To determine the number of unambiguous NFYA-bound genes for each gene category, we computed the distance from the TSSs of poised and non-poised genes to the nearest NFYA peak in three biological replicates. Those genes with NFYA peak within the ± 2 kb of their TSSs in at least two replicates were considered NFYA-bound genes.”*

Considering the strong correlation using Pearson correlation analysis (above motivation as to why Pearson correlation is more suitable for our NFYA CUT&RUN data set) and defining unambiguous NFYA-bound genes by analyzing independent replicates

individually, we believe that repeating all NFYA CUT&RUN experiments for all cell stages would not alter the main conclusion of our study. Moreover, the additional time and resources for repeating all NFYA CUT&RUN experiments for all cell stages pose a burden on our small lab. We hope that the Reviewer will find our revision and justification reasonable.

7. The section on NFYA binding sites contains excessive speculation. Claims should be supported with functional data, ideally from Nfya-cKO mice. For example, Line 329's statement on mitotic transcriptional programs requires supporting evidence.

We thank the Reviewer for the fantastic suggestion. The same point was brought up by the Reviewer #1, comment #3. We now performed differential gene expression analysis between SpG of *Nfya-CKO* and *Cre/+* using our sc-RNA-seq data. In the revised manuscript, we present our findings in a new subsection of the Result section with the title "*NFYA participates in the regulation of mitotic transcriptional program in spermatogonia*" along with the incorporation of a new figure (Figure 6). Our analysis revealed that <5% of all genes were deregulated in SpG from *Nfya-CKO* compared to that from *Cre/+* (Fig. 6A), which likely accounts for moderate perturbation we observed in the differentiation of SpG in *Nfya-CKO* (Fig. 3G). Interestingly, we found that among the genes, whose promoters are bound by NFYA and whose transcript abundance is reduced in NFYA-deficient SpG, gene ontology related to the regulation of transcription was enriched, suggesting that many other NFYA-unbound deregulated genes are likely regulated by transcription factors that are themselves under NFYA control. In our study, we found that NFYA regulates chromatin accessibility at the promoters of *Stra8* and *Meiosin* genes, as well as the promoters of STRA8/MEIOSIN target genes (Fig. EV16). Unlike poised meiosis-I and spermiogenesis genes whose transcription bursts at pachytene stage of meiosis-I, the expression of STRA8/MEIOSIN target genes starts in differentiating SpG and peaks in L/Z/eP spermatocytes consistent with their role in

meiotic entry (Fig. EV16B). Intriguingly, our differential gene expression analysis revealed that RNA abundance of many STRA8/MEIOSIN target genes, whose promoters are bound by NFYA, were reduced in NFYA-deficient SpG partly accounting for the observation that spermatogenesis arrests in *Nfya*-CKO males. See the text from the revised manuscript below (text on page 23).

“NFYA participates in the regulation of mitotic transcriptional program in spermatogonia

*NFYA regulates chromatin accessibility at the promoters of poised meiosis-I and spermiogenesis genes in SpG, prior to the onset of their transcription during prophase I of meiosis (Figs. 1,2, and 5). However, unlike poised genes, transcription of mitotic genes is switched off after entry into meiosis I (Figs. EV1A, EV2B, EV2C). We found that NFYA binds ~one-third of the mitotic gene promoters in SpG (Fig. 2F). Deletion of Nfya results in compromised differentiation of SpG (Fig. 3G). Moreover, scRNA-seq analysis revealed that NFYA-deficient SpG cluster separately from control SpG in UMAP space (Fig. 4A). These suggest a regulatory function for NFYA in mitotic transcriptional program beyond its role in regulating accessible chromatin at poised meiotic promoters. We found that accessible chromatin at mitotic promoters bound by NFYA was decreased in NFYA-deficient SpG ($Nfya\text{-CKO} \div Cre/+ = 0.75$), whereas chromatin accessibility at the promoters of NFYA-unbound mitotic genes remained nearly unchanged ($Nfya\text{-CKO} \div Cre/+ = 0.96$) (Fig. 5D; Two-sided wilcoxon matched-pairs signed ranked sum test; **** $p < 0.0001$). Using scRNA-seq, we next measured the change in the steady-state RNA abundance between SpG from *Nfya*-CKO and *Cre*/*+* (Fig. 6A). The RNA abundance of 1,241 genes—that account for <5% of all genes—were significantly deregulated in SpG from *Nfya*-CKO compared with that from *Cre*/*+* (Fig. 6A; $Nfya\text{-CKO} \div Cre/+ \leq 0.5$ or ≥ 2 ; $FDR < 0.05$). Deregulation of small fraction of all genes likely explains mild perturbation we observed in the differentiation of SpG in*

Nfya-CKO males (Fig. 3G). Of the 1,241 deregulated genes, steady-state RNA abundance of 629 was higher in *NFYA*-deficient SpG, while the RNA products of 612 genes were reduced (Fig. 6A). Our CUT&RUN analysis of *NFYA* binding from wild-type SpG revealed that *NFYA* bound near the TSSs for 201 of 1,241 genes whose transcripts were deregulated in *NFYA*-deficient SpG (Fig. 6B). Of the 201 *NFYA*-bound deregulated genes, RNA abundance of 85 were reduced in *NFYA*-deficient SpG (Fig. 6B). Interestingly, among these 85 genes, 11 belonged to mitotic gene category that were enriched for GO related to the regulation of transcription suggesting that many other *NFYA*-unbound deregulated genes are likely regulated by transcription factors that are themselves under *NFYA* control (Figs. 6C and 6D; Table EV5). Indeed, these 11 genes included *Foxo1* and *Sox3*, which encode transcription factors that participate in the regulation of mitotic genetic program (McAninch et al., 2020; Shen et al., 2022; Yi et al., 2025).

Finally, among the 612 genes whose RNA abundance were reduced in *NFYA*-deficient SpG, 43 were direct *STRA8/MEIOSIN* targets that are required for meiotic entry and progression (Kojima et al., 2019; Ishiguro et al., 2020) (Fig. 6A). Our *NFYA* CUT&RUN analysis from wild-type SpG revealed that *NFYA* bound near the TSSs for 22 of 43 *STRA8/MEIOSIN* target genes, whose transcripts were reduced in *NFYA*-deficient SpG, including *Meioc*, *Hspa5*, *Ybx*, *Pabpc1*, and *Pcbp3* (Fig. 6E; Table EV5). *MEIOC* regulates the transition from mitosis to meiosis-I (Abby et al., 2016; Soh et al., 2017). *HSPA5* is required for the maintenance of differentiating SpG (Wen et al., 2023). RNA-binding proteins, *YBX2*, *PABPC1*, and *PCBP3*, regulate the translation of mRNAs required later during spermatogenesis (Kimura et al., 2009; Chapman et al., 2013; He et al., 2019). Together, these findings are consistent with our observation that spermatogenesis arrests at meiotic entry in *Nfya*-CKO males.”

8. Line 335 (“*NFYA* may act as a pioneer factor...”) should be deleted.

We now deleted the statement “*NFYA may act as a pioneer factor*” and instead revised it as follows: “*Together, our data suggests that prior to meiosis, NFYA may regulate accessible chromatin at the promoters of poised genes that are expressed after meiotic entry.*” We discussed the potential ‘pioneer role’ of NFYA only in the discussion section (text on page 25).

9. Please clarify the identity of *Stra8-Cre* and include an appropriate citation.

We revised the related result section describing the identity of *Stra8-Cre* mice to make the section more accessible for the readership (See below, text on page 13)

*“To understand the biological role of NFYA in meiotic entry and progression in a C57BL/6J background, we generated *Nfyaf1/fl*; *Stra8-Cre* mice in which exons 5–6 of *Nfya* is deleted specifically in germ cells by Cre recombinase (Figs. EV10A–EV10C; henceforth *Nfya-CKO*). *Stra8* expression is active in undifferentiated and differentiating SpG, and pL (Zhou et al., 2008), *Stra8-Cre* mice generated in C57BL/6J background by Cyagen thus express P2A-ZsGreen1-T2A-Cre cassette upstream of the stop codon of *Stra8* in pre-meiotic germ cells (Figs. EV10A–EV10C; Method section).”*

The *Stra8-Cre* line is generated by the animal modeling company, Cyagen Biosciences. We now also updated the Method section by adding the following sentence: “*Stra8-P2A-ZsGreen1-T2A-Cre* mice were purchased from Cyagen (Cyagen Biosciences; C001536).”

10. The evaluation of meiotic entry is insufficient. Additional STRA8 and/or MEIOSIN immunostaining is needed to determine whether preleptotene spermatocytes are present in mutant testes.

We already had double immunofluorescence staining for STRA8 and γ H2AX on the testis sections from *Cre/+* and *Nfya-CKO* mice in the original submission. In the revised

manuscript, Fig. 3E now presents the result for the staining. Below, we provide the corresponding text for Fig. 3E from the revised manuscript (text on page 16).

“We next performed immunostaining for STRA8—expressed highly in pL (Zhou et al., 2008)—along with γ H2AX to evaluate whether NFYA is required for meiotic entry. Intriguingly, some γ H2AX-positive spermatocytes from Nfya-CKO testis showed nuclear STRA8 signal even though those cells were low in number (Fig. 3E).”

11. In Figure 4, the mutant data differ markedly from controls, but the conclusions are unclear. Eight weeks is likely too late for meaningful analysis; an earlier time point (e.g., 11 days postpartum) would be more appropriate.

We thank the Reviewer for pointing out that the mutant data differ markedly from controls (Fig. 4A). Following the suggestion from the Reviewer #1 (Comment #3), in the revised manuscript, we normalized the number of undiff. SpG or diff. SpG to the number of Sertoli cells (Fig. 3G). We found that maintenance of undiff. SpG occurs normally, whereas differentiation of SpG is modestly impaired in Nfya-CKO males (Fig. 3G). This finding explains why the mutant data differ from the control in our scRNA-seq experiment. Along the parallel line, we now performed differential gene expression analysis between SpG of Nfya-CKO and Cre/+ using our sc-RNA-seq data and reported differential transcriptome signature between the mutant and control in our revised manuscript (Subsection in the Result section: “NFYA participates in the regulation of mitotic transcriptional program in spermatogonia”; New figure; Figure 6).

Considering the depth of resolution obtained by sc-multiomics (Fig. EV13), the additional time and resources required for sc-multiome from 11 dpp males pose a burden on small labs like ours. We hope that the Reviewer agrees that sc-multiome experiment is very expensive. Re-running BD Rhapsody Single-Cell Analysis System on the testes of Cre/+ and Nfya-CKO at 11 dpp and sequencing the libraries using

NovaSeq X Plus Platform (Illumina) will cost us additional >\$50,000. We hope that the Reviewer finds our justification reasonable.

12. The analysis in Figure 6 is speculative and not well supported by data. This section should be substantially revised or removed.

Feedforward loops are defined by their genetic architecture. By definition, a coherent feedforward loop has the architecture shown at right (Alon, *Nat Rev Genet*, 2008). In this architecture X and Y are transcription factors, and Z is the target of both. For example, X = Nfya, Y = Stra8 and/or Meiosin, Z = STRA8/MEIOSIN target genes. Nevertheless, we revised the section and dropped the model 'coherent feedforward loop' throughout the manuscript to make the section more accessible to the readership. In the revised manuscript, now the title for the related subsection became "NFYA regulates accessible chromatin at the promoters of genes activated by STRA8/MEIOSIN axis". Figure 6 now became Fig. EV16. Below, we provide the corresponding text for Fig. EV16 from the revised manuscript (text on page 22).

a
Coherent FFL

Coherent
type 1

"NFYA regulates accessible chromatin at the promoters of genes activated by STRA8/MEIOSIN axis

TFs STRA8 and MEIOSIN induce the transition from mitosis to meiosis (Anderson et al., 2008; Kojima et al., 2019; Ishiguro et al., 2020). Similar to the defective phenotypes reported in Stra8 (Anderson et al., 2008) and Meiosin (Ishiguro et al., 2020) mutant mice, spermatogenesis arrests at meiotic entry in Nfya-CKO mice. Interestingly, we observed reduced chromatin accessibility at Stra8 and Meiosin promoters that are in

fact bound by NFYA (Fig. EV16A; Table EV3; *Stra8*, *Nfya*-CKO ÷ *Cre*/⁺ = 0.90; *Meiosin*, *Nfya*-CKO ÷ *Cre*/⁺ = 0.70). We next sought to examine the change in chromatin accessibility at the promoters of 1,057 genes whose transcription is directly regulated by both STRA8 and MEIOSIN or STRA8 alone, or MEIOSIN alone (Kojima et al., 2019; Ishiguro et al., 2020). The RNAs of STRA8/MEIOSIN target genes first appeared in diff. SpG and peaked in early spermatocytes of prophase I, L/Z/eP, corroborating with their role in meiotic initiation and progression (Kojima et al., 2019; Ishiguro et al., 2020) (Fig. EV16B). Intriguingly, of the 1,057 genes, the promoters of 471 were bound by NFYA in SpG (Fig. EV16C; left panel; median distance from TSS to the nearest NFYA peak = 257 bp). We found that chromatin accessibility at the promoters of NFYA-bound STRA8/MEIOSIN target genes was reduced ~37% in undifferentiated SpG from *Nfya*-CKO mice compared to that from *Cre*/⁺ mice (Fig. EV16C; right panel; Two-sided wilcoxon matched-pairs signed ranked sum test; *****p* < 0.0001). Among the 471 NFYA-bound STRA8/MEIOSIN target genes whose promoter accessibility was reduced in NFYA-deficient SpG, many function in meiotic recombination (e.g., *Sycp2*, *Syce3*, *Prdm9*, *Rad51*, *Msh5*, *Stag3* (Handel and Schimenti, 2010; Baudat et al., 2013)), cell cycle (e.g., *Meioc* (Soh et al., 2017)), and meiotic transcriptional program (e.g., *Taf4b* (Falender et al., 2005)) (Fig. EV16D). Together, our data suggest that NFYA regulates accessible chromatin at the promoters of *Stra8* and *Meiosin* genes as well as genes that are activated by STRA8 and MEIOSIN”

Moreover, our differential gene expression analysis revealed the RNA abundance of many STRA8/MEIOSIN target genes, whose promoters are bound by NFYA, were reduced in NFYA-deficient SpG partly substantiating the regulatory role of NFYA on STRA8/MEIOSIN axis (See the subsection in the Result section: “NFYA participates in the regulation of mitotic transcriptional program in spermatogonia”; New figure; Figure 6). We hope that the Reviewer finds our revision reasonable.

Dear Prof. Ozata,

Thank you for submitting a revised version of your manuscript. Your study has now been seen by all original referees, who find that their previous concerns have been addressed and now only request minor textual edits and recommend publication of the manuscript. In addition to these requested textual edits there remain only a few mainly editorial points that have to be addressed before I can extend formal acceptance of the manuscript:

- AFFILIATIONS (research institution or university vs. biotech company): employment in a biotech company should be stated in DCIS
- Please adjust the format of the reference list and of the in-text citations according to EMBO Journal format (alphabetical order, author name et al + year.../up to 10 author names in the reference list before et al / please refer to our Guide to Authors for additional information on EMBO J reference format).
- Please rename the "Data and code availability" section to "Data availability"
- PROTOCOL: please check the example of protocol.io on page 48
- Please adjust the in-text callouts for individual figures and figure panels: e.g. there is a callout for the missing Table S2; missing callout for Fig. 4D
- DATASET EV LEGENDS: source file names, titles, legends and manuscript callouts all need to be updated to Dataset EV1-EV# instead of Tables EV1-EV7, legends should be removed from ms and uploaded as a separate tab/sheet in each Excel file APPENDIX 1 FILE WITH ToC: n/a
- Please remove the R&T TABLE from the manuscript file and upload as an individual file using the template from our guide to the authors
- Please upload the synopsis in jpeg/tif format and make sure that it is exactly 550 pixels wide and between 300-600 pixels high.
- "Structured Methods" should be renamed to "Methods"
- EV figures should be renamed to Figure EV1-EV16 instead of Expanded View Figure 1 in figure legends and labels (please consider moving some of the EV figures to the Appendix PDF with the appropriate nomenclature and callouts)
- Sections need to be named and the order should be corrected: Title page - Abstract - Keywords - Introduction - Results - Discussion - Methods - Data Availability - Acknowledgements - Disclosure and Competing Interests Statement - References - Figure Legends - Table(s) - Expanded View Figure Legends.
- Please remove the statistical analysis from all figures with n=2 (2b and EV11b) and show data points only.
- Figure Legends (main + EV):
 1. Please define the annotated p values ****/***/**/* as well as provide the exact p-values for the same in the legend of figure 5d; EV 14b; EV 15b, e as appropriate.
 2. Please note that the exact p values are not provided in the legends of figures 1b; 3b, c, e-g; EV 9b; EV 10e; EV12 a-e; EV 16c
 3. Please indicate the statistical test used for data analysis in the legends of figures 5d; 6a; figure 5d; EV 14b; EV 15b, e
 4. Please note that the MI, D arrows are not defined in the legend of figures 3f; EV 6C.

With best regards,
Cornelius Schneider

Cornelius Schneider, PhD
Editor | The EMBO Journal
c.schneider@embojournal.org

Please refer to our figure preparation guideline in order to ensure proper formatting and readability in print as well as on screen:

<https://link.springer.com/journal/44318/submission-guidelines#cms-Figure-and-data-presentation>

Read our guidance for manuscript revisions and related editorial policies: <https://link.springer.com/journal/44318/submission-guidelines#cms-Revised-submissions>

<https://media.springernature.com/original/springer-cms/rest/v1/content/27825798/data/v1>

- a point-by-point response to the referees' comments, with a detailed description of the changes made (as a word file).
- a word file of the manuscript text.
- individual production quality figure files (one file per figure)
- a complete author checklist
- Expanded View files (replacing Supplementary Information)
- a Reagents and Tools Table as part of the Methods section

Please remember: Digital image enhancement is acceptable practice, as long as it accurately represents the original data and conforms to community standards. If a figure has been subjected to significant electronic manipulation, this must be noted in the figure legend or in the 'Methods' section. The editors reserve the right to request original versions of figures and the original images that were used to assemble the figure.

We realize that it is difficult to revise to a specific deadline. In the interest of protecting the conceptual advance provided by the work, we recommend a revision within 3 months (19th Apr 2026). Please discuss the revision progress ahead of this time with the editor if you require more time to complete the revisions. Use the link below to submit your revision:

Referee #1:

The authors have adequately revised their manuscript, and the revisions satisfactorily address my previous concerns.

Referee #2:

I found that the authors have made substantial efforts to improve the manuscript, and I sincerely commend them for these improvements. Many of my previous comments have been adequately addressed. At this stage, I have only several minor suggestions.

1. The newly added discussion section addressing the potential pioneer function of NFYA is appropriate and well aligned with the data. However, I do not agree with the citation of the previous study stating that "pioneer factors can also bind to sites that

are primed by prior pioneer factors or accessible chromatin features (Cernilogar et al., NAR, 2019)," as this point is not directly related to pioneer activity per se. Overall, the discussion is scientifically well reasoned, but the manuscript would benefit from English language editing to improve clarity, grammar, and flow.

2. Regarding previous major point #1, I still observe multiple incomplete citations throughout the manuscript. Please ensure that all citations are complete and correctly formatted.
3. Regarding previous major point #2, please clearly explain what PRO-seq detects and consistently use the term "engaged Pol II peaks" rather than "Pol II peaks," as the latter is potentially confusing. The term "Pol II peaks" is ambiguous, as it may be interpreted as representing all forms of RNA polymerase II.
4. Regarding previous major point #11, I understand the budget limitations. I suggest simplifying the presentation of the results in Figure 4 and explicitly noting the limitations of the current dataset so that readers are aware of these constraints when interpreting the data.
5. For the new Figure 6, please clarify the age of the mice analyzed.

We thank the Reviewers for their endorsement for the improvements made in our first round of revision and we are very glad that the Reviewers suggest our manuscript for publication in *The EMBO Journal*.

We have carefully reviewed the remaining comments from the Reviewers and the Editors and have addressed them point-by-point below. We believe that these final changes have further strengthened the clarity and accuracy of the manuscript. All changes made in the manuscript are marked in yellow for your convenience.

Point-by-point responses to the Editors' and Reviewers' critiques

Editorial comments:

- *AFFILIATIONS (research institution or university vs. biotech company): employment in a biotech company should be stated in DCIS.*

There are no authors listed in our manuscript who are employed in a biotech company.

- *Please adjust the format of the reference list and of the in-text citations according to EMBO Journal format (alphabetical order, author name et al + year.../up to 10 author names in the reference list before et al / please refer to our Guide to Authors for additional information on EMBO J reference format).*

We now formatted according to the *EMBO Journal* guidelines.

- *Please rename the "Data and code availability" section to "Data availability".*

We now renamed the section accordingly.

- *PROTOCOL: please check the example of protocol.io on page 48*

We did not invent a new protocol/method in this study; we used publicly available protocols/methods. We therefore do not have any protocol to submit to protocol.io.

- Please adjust the in-text callouts for individual figures and figure panels: e.g. there is a callout for the missing Table S2; missing callout for Fig. 4D

We now adjusted the callouts.

- DATASET EV LEGENDS: source file names, titles, legends and manuscript callouts all need to be updated to Dataset EV1-EV# instead of Tables EV1-EV7, legends should be removed from ms and uploaded as a separate tab/sheet in each Excel file

APPENDIX 1 FILE WITH ToC: n/a.

We now renamed the Tables EV to Dataset EV throughout the manuscript. Moreover, we removed the legends of Dataset EVs from the manuscript and uploaded as a separate tab in each excel file.

- Please remove the R&T TABLE from the manuscript file and upload as an individual file using the template from our guide to the authors.

We submitted a separate word file containing the R&T table using the template provided on *The EMBO Journal* website.

- Please upload the synopsis in jpeg/tif format and make sure that it is exactly 550 pixels wide and between 300-600 pixels high.

We uploaded a TIFF image of our synopsis with the size corresponding to 550 pixels wide and 300-600 pixels high.

- "Structured Methods" should be renamed to "Methods"

We renamed the section accordingly

- EV figures should be renamed to Figure EV1-EV16 instead of Expanded View Figure 1 in figure legends and labels (please consider moving some of the EV figures to the Appendix PDF with the appropriate nomenclature and callouts).

We renamed the Expanded View Figures as Figure EV in figure legends and labels accordingly. We now selected five EV figures from the original submission (EV2 now become EV1; EV9 now become EV2; EV14 now become EV3; EV15 now become EV4; EV16 now become EV5). The remaining Figure EVs are now moved to Appendix according to *The EMBO Journal* guidelines. In our revised manuscript, we now have Appendix Figures S1–S11.

- Sections need to be named and the order should be corrected: Title page - Abstract - Keywords - Introduction - Results - Discussion - Methods - Data Availability - Acknowledgements - Disclosure and Competing Interests Statement - References - Figure Legends - Table(s) - Expanded View Figure Legends.

We renamed the sections accordingly.

- Please remove the statistical analysis from all figures with $n=2$ (2b and EV11b) and show data points only.

We did not include any statistical analysis for 9 dpp ($n = 2$) in Appendix Figure S9B (previously Fig. EV11B). In Figure 2B, we have included the data points from the western blots of independent replicates, and have not performed any statistical test.

- Figure Legends (main + EV):

1. Please define the annotated p values ****/**/*/* as well as provide the exact p -values for the same in the legend of figure 5d; EV 14b; EV 15b, e as appropriate.

We now defined annotated p values ****/**/*/* in the figure legends.

2. Please note that the exact p values are not provided in the legends of figures 1b; 3b, c, e-g; EV 9b; EV 10e; EV12 a-e; EV 16c

We now provide the exact p values in figure legends.

3. Please indicate the statistical test used for data analysis in the legends of figures 5d; 6a; figure 5d; EV 14b; EV 15b, e

We updated the legends with the statistical tests used.

4. Please note that the MI, D arrows are not defined in the legend of figures 3f; EV 6C.

We now included the definition of the arrows for MI and D in the legend.

Reviewers' comments:

Referee #1:

The authors have adequately revised their manuscript, and the revisions satisfactorily address my previous concerns.

We are very gladdened to have the Reviewer's endorsement for that we adequately revised our manuscript.

Referee #2:

I found that the authors have made substantial efforts to improve the manuscript, and I sincerely commend them for these improvements. Many of my previous comments have been adequately addressed. At this stage, I have only several minor suggestions.

We thank the Reviewer for acknowledging our improvements in our manuscript.

1. *The newly added discussion section addressing the potential pioneer function of NFYA is appropriate and well aligned with the data. However, I do not agree with the citation of the previous study stating that "pioneer factors can also bind to sites that are primed by prior pioneer factors or accessible chromatin features (Cernilogar et al., NAR, 2019)," as this point is not directly related to pioneer activity per se.*

We thank the Reviewer for endorsing the improvements in our revised discussion section. We also thank for the Reviewer's perspective; we now revised the related paragraph by dropping the following sentence: "*Although widely accepted notion is that pioneer factors bind closed chromatin (Barral and Zaret, 2024), they can also bind to sites that are primed by accessible chromatin features or prior pioneer factors (Cernilogar et al., 2019).*".

Overall, the discussion is scientifically well reasoned, but the manuscript would benefit from English language editing to improve clarity, grammar, and flow.

We would like to note that we asked help from Dr. Matthew Hunt (PhD, Karolinska Institute, Sweden), who is a native English speaker, to proofread our manuscript in terms of English clarity and grammar before we submit our manuscript. We thus now acknowledge his name in the Acknowledgement section.

2. Regarding previous major point #1, I still observe multiple incomplete citations throughout the manuscript. Please ensure that all citations are complete and correctly formatted.

We carefully went through all citations and reformatted them in the manuscript according to *The EMBO Journal* guidelines.

3. Regarding previous major point #2, please clearly explain what PRO-seq detects and consistently use the term "engaged Pol II peaks" rather than "Pol II peaks," as the latter is potentially confusing. The term "Pol II peaks" is ambiguous, as it may be interpreted as representing all forms of RNA polymerase II.

We now clarified what PRO-seq detects in lines between 141–146, page 6. "*Using publicly available Precision Run-On sequencing data (PRO-seq (Kaye et al., 2024)) from purified SpG, Spl, and RS, we quantified paused and elongating Pol II across*

meiosis-I and spermiogenesis genes. Spike-in sequences in each sample enabled us to quantify absolute PRO-seq signal. Note that PRO-seq provides the genome-wide mapping of transcriptionally engaged RNA polymerases (Core et al., 2008). However, transcription units we study are transcribed by Pol II.”.

Moreover, we thank the Reviewer’s suggestion; we now used the term “engaged Pol II peaks” rather than “Pol II peaks” throughout the manuscript.

4. Regarding previous major point #11, I understand the budget limitations. I suggest simplifying the presentation of the results in Figure 4 and explicitly noting the limitations of the current dataset so that readers are aware of these constraints when interpreting the data.

We now simplified the Figure 4; We moved the previous panels, Figs. 4D & 4F, to new Appendix Figure S11. In our revised manuscript, Fig. 4D become Appendix Figure S11F and Fig. 4F become Appendix Figure S11E. Following the Reviewer’s suggestion, we now noted the limitation of using 8-weeks-old mice as follows (lines between 531–535, page 19): “*Even though 8 weeks possibly limits the depth for detection of spermatogonia in Cre/+ mice, our systematic analysis of simultaneous scRNA-seq and scATAC-seq demonstrated continuous development of germ cells in wild-type and Cre/+ testes, and provided further supporting evidence that spermatogenesis arrests at meiotic entry in Nfya-CKO mice.*”

5. For the new Figure 6, please clarify the age of the mice analyzed.

We now specified the age of mice used, which is 8-week-old, in the legend for Fig. 6A.

Dear Prof. Ozata,

I am pleased to inform you that your manuscript has been accepted for publication in the EMBO Journal.

You may qualify for financial assistance for your publication charges - either via a Springer Nature fully open access agreement or an EMBO initiative. Check your eligibility: <https://link.springer.com/journal/44318/how-to-publish-with-us>

Yours sincerely,

Cornelius Schneider, PhD
Editor
The EMBO Journal
c.schneider@embojournal.org

Please note that it is The EMBO Journal policy for the transcript of the editorial process (containing referee reports and your response letters) to be published as an online supplement to each paper. If you should prefer removal of any referee-only figures included in the point-by-point response(s), e.g. because they may still be used for future publication or because they have been reproduced from published work by others, please do let us know immediately via response email.

More information is available here: <https://link.springer.com/partners/embo-press/editorial-policies#Peer%20review>